# Pushing the boundary of quantum advantage in hard combinatorial optimization with probabilistic computers

Shuvro Chowdhury [1] ✉, Navid Anjum Aadit [1], Andrea Grimaldi [2,3], Eleonora Raimondo [2,4], Atharva Raut [5], P. Aaron Lott[6,7], Johan H. Mentink [8], Marek M. Rams [9], Federico Ricci-Tersenghi [10], Massimo Chiappini [4], Luke S. Theogarajan [1], Tathagata Srimani[5], Giovanni Finocchio [2], Masoud Mohseni[11] & Kerem Y. Camsari [1] ✉

Recent demonstrations on specialized benchmarks have reignited excitement for quantum computers, yet their advantage for real-world problems remains an open question. Here, we show that probabilistic computers, co-designed with hardware to implement Monte Carlo algorithms, provide a scalable classical pathway for solving hard optimization problems. We focus on two algorithms applied to three-dimensional spin glasses: discrete-time simulated quantum annealing and adaptive parallel tempering. We benchmark these methods against a leading quantum annealer. For simulated quantum annealing, increasing replicas improves residual energy scaling, consistent with extreme value theory. Adaptive parallel tempering, supported by non-local isoenergetic cluster moves, scales more favorably and outperforms simulated quantum annealing. Field Programmable Gate Arrays or specialized chips can implement these algorithms in modern hardware, leveraging massive parallelism to accelerate them while improving energy efficiency. Our results establish a rigorous classical baseline for assessing practical quantum advantage and present probabilistic computers as a scalable platform for real-world optimization challenges.

Richard Feynman is widely credited with starting the field of quantum computing in a 1982 lecture[1]. In the same lecture, Feynman also introduced the notion of a probabilistic computer, one that naturally simulates probabilistic processes. Feynman's broader vision of physical computers, or programmable physical devices that solve a problem of interest through their natural evolution, has recently inspired a growing array of physical and physics-inspired classical computing paradigms, including systems to train deep neural networks[2], solve linear algebra problems[3], and tackle combinatorial optimization problems[4].

[1]Department of Electrical and Computer Engineering, University of California, Santa Barbara, Santa Barbara CA 93106, USA. [2]Department of Mathematical and Computer Sciences, Physical Sciences and Earth Sciences, University of Messina, 98166 Messina, Italy. [3]Department of Electrical and Information Engineering, Politecnico di Bari, 70126 Bari, Italy. [4]Istituto Nazionale di Geofisica e Vulcanologia, Via di Vigna Murata 605, 00143 Roma, Italy. [5]Department of Electrical and Computer Engineering, Carnegie Mellon University, Pittsburgh, PA, USA. [6]USRA Research Institute for Advanced Computer Science (RIACS), Mountain View, CA, USA. [7]Quantum Artificial Intelligence Laboratory (QuAIL), NASA Ames Research Center, Moffett Field, CA, USA. [8]Radboud University, Institute for Molecules and Materials, Heyendaalseweg 135, Nijmegen, The Netherlands. [9]Institute of Theoretical Physics, Jagiellonian University, Lojasiewicza 11, PL-30348 Kraków, Poland. [10]Dipartimento di Fisica, Sapienza Università di Roma, and CNR-Nanotec, Rome unit and INFN, Sezione di Roma 1, 00185 Roma, Italy. [11]Emergent Machine Intelligence, Hewlett Packard Labs, Palo Alto, CA, USA. ✉e-mail: schowdhury@ucsb.edu; camsari@ucsb.edu

Building on this vision, a key challenge is identifying scenarios where scalable and error-corrected quantum computers[5] could outperform probabilistic or classical approaches, particularly in optimization and sampling tasks. Prominent examples include Shor's algorithm for factoring large integers[6], sampling random quantum circuits[7], and learning quantum data on quantum processors[8,9], each offering potential exponential speedups over all known probabilistic alternatives, typically due to the interference of probability amplitudes in a high-dimensional Hilbert space. However, the scaling challenges and quantum error correction overhead might diminish or eliminate such quantum advantages[5]. Notably, while probabilistic computers can emulate quantum interference with polynomial resources, their convergence is in general believed to require exponential time[10]. This challenge is known as the sign-problem in Monte Carlo algorithms[11].

On the other hand, establishing a quantum advantage becomes much harder in cases where quantum fluctuations or quantum interference may be present, but not known to play a significant role. For example, even though quantum annealers by D-Wave operate on the transverse field Ising Hamiltonian, which does not suffer from the sign problem, empirical performance advantages have been sought to be demonstrated over the years[12–15]. In a similar attempt, Bernaschi et al.[16]

clarified that for a 2D quantum spin glass, quantum annealing could still provide a speedup in entering the spin-glass phase under certain conditions. However, it is unclear whether these advantages extend to solving optimization problems and represent a fundamental improvement over classical algorithms, such as simulated quantum annealing (SQA) and adaptive parallel tempering (APT), or if they are primarily due to hardware acceleration. Speedups of this second type are also a feature of dedicated probabilistic computers when the hardware architecture is tailored for probabilistic algorithms[17].

Recently, King et al.[18] demonstrated another empirical scaling advantage over continuous-time simulated quantum annealing (CT-SQA) and simulated annealing (SA) in solving classical 3D cubic Ising spin glass problems (Fig. 1a). Due to their hardness, 3D spin glasses have long served as canonical benchmarks for evaluating scaling behavior of various algorithms[19–21].

The performance reported in ref. [18] provides a timely and valuable benchmark for the field. In this work, we use this benchmark to evaluate a powerful classical alternative: probabilistic computers co-designed with domain-specific hardware. While the quantum critical dynamics observed in that study are a significant physical finding, it is crucial to assess whether the resulting optimization performance is

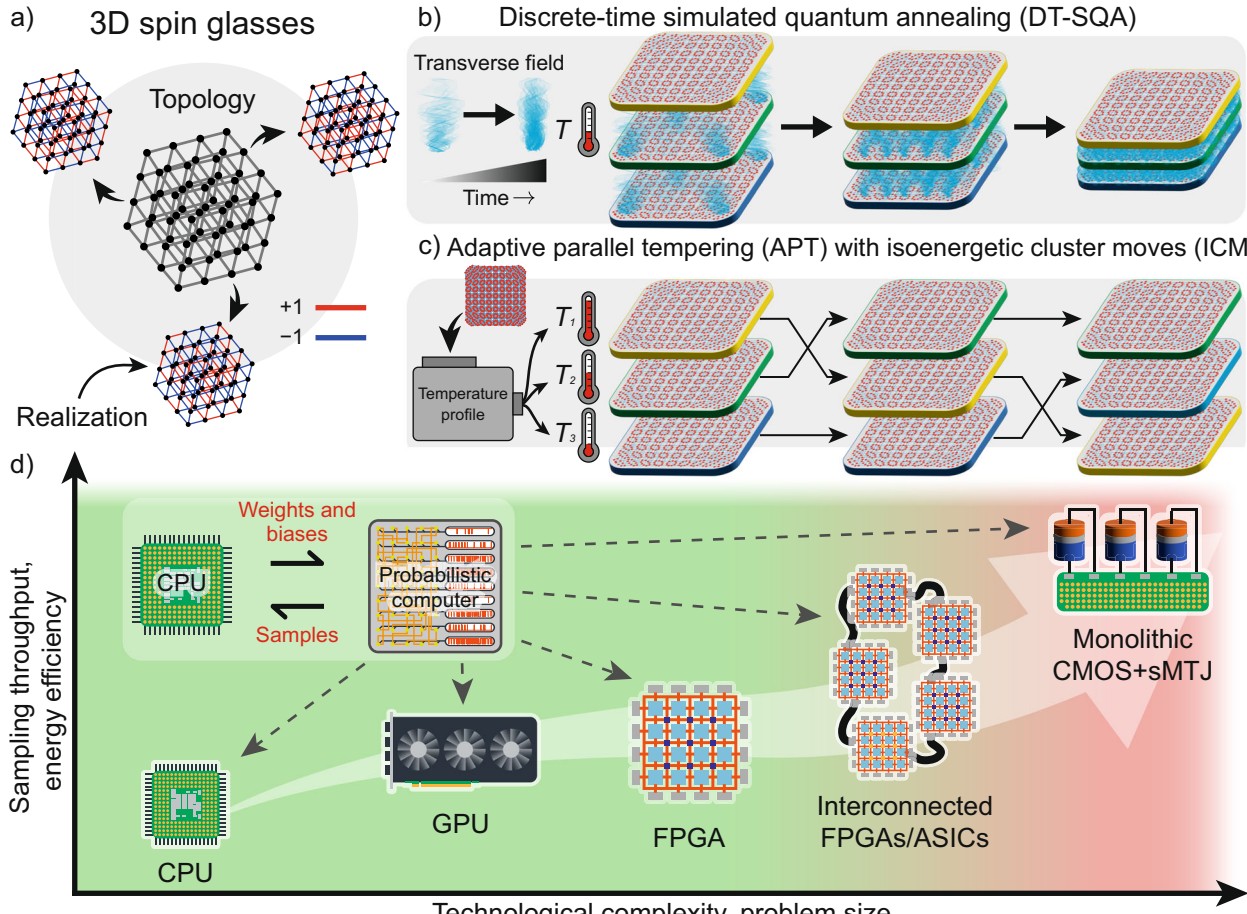

**Fig. 1 | Schematic overview of the probabilistic algorithms and technological platforms. a** Representation of a 3D Ising spin glass, where the weights connecting spins can be +1 (ferromagnetic, red) or −1 (antiferromagnetic, blue). Many real-world hard optimization problems can be mapped onto such spin-glass systems. **b, c** Two replica-based algorithms investigated in this work: discrete-time simulated quantum annealing (DT-SQA), in (b), and adaptive parallel tempering (APT) with isoenergetic cluster moves (ICM), in (c). In SQA, replicas are interconnected, and the strength of these interconnections is annealed according to a schedule. APT uses multiple replicas of the problem, each running at different temperatures, and

periodically exchanges states between replicas to escape local minima. **d** The p-computing scheme, where a general-purpose CPU interfaces with a specialized probabilistic computer to efficiently implement Monte Carlo algorithms. Various implementations of probabilistic computers are shown, including CPU[4,17], GPU[22–26], Field Programmable Gate Arrays (FPGA)[4,27–31], interconnected FPGAs, and monolithic CMOS + sMTJ (stochastic magnetic tunnel junction) chips[32,58–60]. Each platform offers trade-offs in sampling throughput, energy efficiency, problem size, and technological complexity.

competitive with the most advanced classical techniques. Our study addresses this by demonstrating that p-computers, implementing state-of-the-art replica-based Monte Carlo algorithms (Fig. 1(b, c)), can achieve a comparable, and in some cases more favorable, performance scaling on the same 3D spin glass problems. Specifically, we investigate discrete-time simulated quantum annealing (DT-SQA) and adaptive parallel tempering (APT) on various p-computer realizations (Fig. 1d).

Our hybrid computing platform combines a general-purpose computer with a p-computer specializing in fast Monte Carlo sampling (Fig. 1d). p-computers have been implemented in CPUs[4,17], GPUs[22–26], Field Programmable Gate Arrays (FPGAs)[4,27–31], and interconnected FPGAs. Specialized accelerators using single and distributed FPGAs already provide orders of magnitude performance improvements over CPUs[4]. Although small-scale p-computers using CMOS + stochastic magnetic tunnel junction technology (sMTJ) have been developed[32], monolithically integrated CMOS + sMTJ chips hold the greatest promise in terms of energy efficiency and performance. However, the large-scale monolithic integration of CMOS + sMTJ remains to be seen.

For our experiments, we use CPU and FPGA implementations of p-computers. For scaling studies, we use CPUs when prefactors of solution times are not critical and FPGAs when they are a priority. Specifically, we use DT-SQA with a large number of physical replicas and select the best replica at the end of the annealing. Using extreme value theory, we relate scaling exponents to the number of replicas, achieving good agreement with our experiments. In addition, a powerful variant of PT, equipped with isoenergetic cluster moves (ICM)[33–35], exhibits a transition from an initial gentler slope to a steeper one due to the non-local moves, providing superior scaling to DT-SQA. Finite-size scaling analysis reveals a collapse of residual energy curves, our primary metric of solution quality, for APT with the steeper slope emerging as a universal feature that delivers better performance in large-scale optimization problems, where minimizing the time-to-solution for a target residual energy is the key objective. Projections based on open-source process design kits show that modern digital chip technology can accommodate a large number of on-chip replicas, making all of our algorithms readily manufacturable in single chips. We also analyze the prefactor and architectural improvements achievable through dedicated FPGA and ASIC implementations. The projections further extend to modern digital chips and CMOS + X-type architectures incorporating nanodevices.

## Results and discussion
### Residual energy of 3D spin glasses
The problem setting is the Edwards-Anderson spin glass on the 3D cubic lattice:

$$H = -\sum_{i<j} J_{ij}\sigma_i\sigma_j, \tag{1}$$

where $\sigma_i$ are Ising spins, $\sigma_i \in \{-1, +1\}$. The coupling weights $J_{ij}$ are nonzero exclusively for nearest-neighbor pairs and, for those pairs, each $J_{ij}$ is randomly selected from $\{-1, +1\}$ with equal probability. One quantity of interest is the residual energy $\rho_E^f$ defined as a function of the annealing time $t_a$:

$$\rho_E^f(t_a) = \frac{\langle E(t_a) - E_0\rangle}{n}, \tag{2}$$

where $E_0$ is the ground energy of the Hamiltonian $H$, $E(t_a)$ is the energy measured at the end of the annealing time $t_a$ and $n$ is the number of spins in the system. The averaging is performed over different problem instances and multiple independent runs.

Experimental observations from probabilistic Monte Carlo algorithms and quantum annealers show that the residual energy scales as a power-law in $t_a$:

$$\rho_E^f \propto t_a^{-\kappa_f}. \tag{3}$$

where $\kappa_f$ is the fitted scaling exponent describing the power-law decay of the residual energy.

While the performance scaling in ref. 18 is analyzed in the context of the Kibble-Zurek mechanism (KZM), it is noted there that the residual energy does not follow a simple prediction from critical dynamics alone, as also noted in ref. 16. In our work, we are therefore primarily focused on the quality of solutions, using the residual energy scaling (Eq. (3)) simply as an empirical benchmark for different optimizers. We do not attempt to map our data onto specific KZM exponents, nor do we assume that near-critical power laws fully govern the eventual solution quality for these optimization problems.

### Analysis of residual energy scaling with DT-SQA
The DT-SQA is an annealing-based algorithm inspired by the principle of adiabatically reducing the transverse field in a quantum system. Using the well-known Suzuki-Trotter transformation[36], a $d$-dimensional quantum system is mapped onto a $(d+1)$-dimensional classical system. The additional imaginary-time dimension is composed of $R$ interconnected Trotter replicas of the original quantum system, where qubits are replaced by Ising spins. As proposed in ref. 37, our strategy is to implement the DT-SQA algorithm directly on probabilistic hardware, using distinct physical replicas.

In Fig. 2a, we evaluate the scaling performance of DT-SQA by plotting $\rho_E^f$ as a function of the annealing time, $t_a$, with varying $R$. The inverse temperature is set to $\beta = 0.5R$ and the simulations were performed on CPUs using logical problem instances defined on a 3D cubic lattice of Ising spins with dimensions $15 \times 15 \times 12$, obtained directly from ref. 18. The results show that the absolute value of the slope, $\kappa_f$, increases with $R$ when the minimum energy among the $R$ replicas is selected. A comparison in Fig. 2b reveals that the scaling exponent of DT-SQA becomes comparable to that of the quantum annealer around $R = 2850$ replicas (with $\kappa_f = 0.805$) and above. It is important to note that this is a comparison of the dimensionless scaling exponent, $\kappa_f$, which is independent of the units on the time axis (MCS for p-computers, nanoseconds for the QA). The QA residual energy data is multiplied by a factor of 2 to align with logical instances, based on the observation that broken dimers (when physical spins representing the same logical spin do not agree after annealing) are rare[18]. In ref. 18, $\kappa_f = 0.785$ and $\kappa_f = 0.51$ are quoted for the quantum annealer and CT-SQA algorithm, respectively.

The slopes quoted above are based on embedded instances (logical problem instances mapped onto the quantum annealer's physical qubit connectivity graph). Quantum annealers (QAs) typically require complex embedding schemes for combinatorial optimization problems, in which a single logical spin is represented by multiple physical spins grouped into structures called dimers, due to their fixed hardware topology (such as the Chimera or Pegasus graphs), even for relatively sparse problems like 3D spin glasses. By contrast, probabilistic computers implemented on flexible classical hardware, such as FPGAs or ASICs, can directly represent and solve the logical problem graph without embedding overhead. Since our goal is to evaluate the intrinsic performance of algorithms solving practical combinatorial optimization problems, our main results do not include embedding overheads that are specific to current quantum annealer architectures. Nevertheless, we essentially obtain similar results also on embedded graphs (see Supplementary Fig. S2) noting that the DT-SQA algorithm can match the scaling of quantum annealers in both cases.

Although both our work and ref. 18 employ SQA, our approach and goals are different. Reference 18 uses the continuous-time variant (CT-SQA) as a theoretical baseline, whereas we deliberately use the

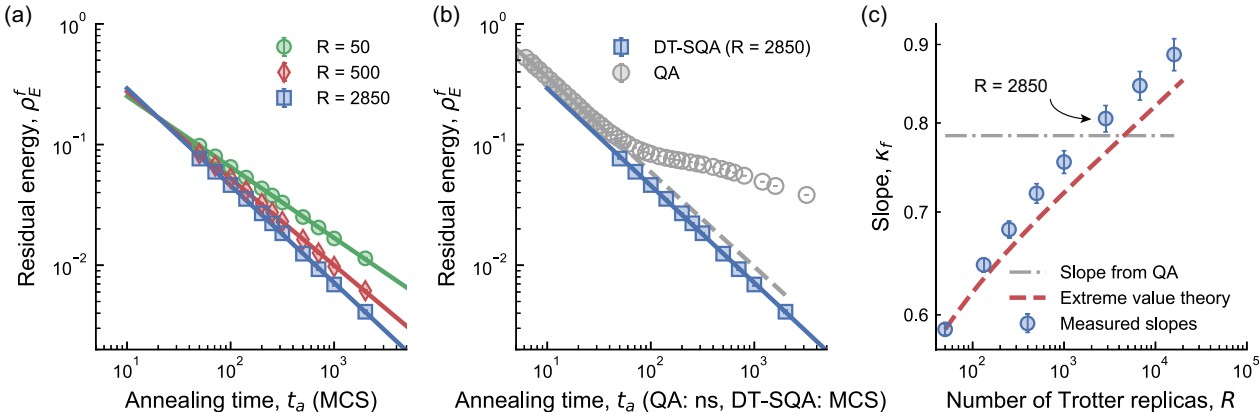

**Fig. 2 | Scaling improvement of the discrete-time simulated quantum annealing (DT-SQA) algorithm. a** The final residual energy, $\rho_E^f$, as a function of annealing time, $t_a$ (in Monte Carlo sweeps, (MCS)), for DT-SQA simulations with varying Trotter replicas, $R$, is shown for 3D Ising spin glass problems ($15 \times 15 \times 12$ with 2687 spins) using CPUs. The slope increases in absolute value with $R$. Each data point is averaged over 300 instances and 50 independent runs. The best energy among all Trotter replicas is selected at the end of $t_a$. Error bars denote the 95% bootstrap confidence interval of the mean across spin-glass instances. An MCS is an algorithmic unit representing one update attempt per spin; hardware differences alter the wall-clock time per sweep but not the scaling exponent $\kappa_f$. The comparison with the quantum annealer's physical time in **b** is therefore a comparison of this dimensionless exponent, not a direct conversion of time units. A full analysis of wall-clock time, which incorporates these hardware-specific prefactors, is presented in the architecture-based scaling improvement subsection and Fig. 4. **b** Comparison of the DT-SQA residual energy scaling (blue squares) with that of the quantum annealer (QA; gray circles) from ref. 18. With a sufficient number of replicas ($R = 2850$), DT-SQA achieves a more favorable scaling exponent. See Supplementary Fig. S2 for similar results on embedded instances. **c** The slope improvement is plotted against $R$ (blue circles), showing alignment with extreme value theory predictions (red dashed line; see Supplementary Section II). The DT-SQA scaling exponent becomes comparable to the QA's at $R \approx 2800$ and exceeds it for larger values. Error bars denote 95% confidence interval of fitting.

discrete-time SQA (DT-SQA) with $R$ explicit Trotter replicas. We chose DT-SQA as it maps naturally onto parallel hardware architectures, making it a more relevant algorithm for assessing the performance of physically realizable classical systems. Therefore, we benchmark our hardware-amenable algorithm directly against the quantum annealer's performance, rather than reproducing the CT-SQA baseline. In Monte Carlo simulations intended to accurately emulate equilibrium quantum physics, selecting the best-performing replica among multiple Trotter replicas is usually avoided, as this could bias equilibrium observables[38,39]. However, such concerns are not relevant in our context, because our goal is not quantum emulation but rather practical combinatorial optimization. Indeed, since our replicas represent independent physical entities realized by separate physical spins in hardware, identifying and selecting the replica with the lowest residual energy is both natural and appropriate.

Next, we show that the observed increase in slope $\kappa_f$ with respect to $R$ can be explained using extreme value theory (EVT; see Supplementary Section II for details) with modifications to account for correlations among replicas. In conventional EVT, the minimum energy is selected from $P$ independent runs of an algorithm, shifting the expected value of the minimum energy by $\mathcal{O}(\sqrt{\ln P})$ from the mean of the original distribution ($P = 1$). In DT-SQA, the Trotter replicas are interconnected and correlated, complicating a direct application of EVT. We observe however that the replica correlations decay over a distance, allowing us to extract effectively independent block sizes. Another complication is the dependence of the correlations with $t_a$, as the transverse coupling among the replicas ($J_\perp$) strengthens (see Supplementary Section I and Supplementary Fig. S6). To apply an EVT theory despite these complications, we partition the $R$ Trotter replicas into $P$ effective blocks, where replicas within a block are correlated but largely uncorrelated with other blocks. We then treat these blocks as separate runs and observe that the predicted scaling behavior that aligns closely with the slopes observed in Fig. 2c. The sizes of the extracted effective blocks correspond well to the measured replica-to-replica correlations (Supplementary Fig. S7b), providing further support for our modified EVT analysis.

The modified EVT approach explains how increasing replicas within a single run improves the scaling. An alternative strategy, also based on EVT, is to leverage multiple independent runs: use a fixed number of interconnected Trotter replicas, run the algorithm $P$ times independently and then select the best energy from all runs. We find that by setting $R = 32$ and running $P = 50$ independent iterations (a total of 1600 replicas), followed by selecting the best solution, DT-SQA also achieves slopes comparable to those of the quantum annealer (see Supplementary Fig. S5). However, this approach remains valid over a shorter range of $t_a$ before the power law breaks down and transitions to a flat plateau region (see Supplementary Fig. S8), showing that the two approaches are not equivalent. Nevertheless, both DT-SQA approaches—$R = 2850$, $P = 1$ (shown in Fig. 2) and $R = 32$, $P = 50$ —are feasible for implementation on a single classical chip where large groups of spins on the chip can be updated simultaneously with massive parallelism, as we discuss in the Physical Design Feasibility subsection.

It is important to note that there is a fundamental difference between the two approaches we demonstrated. When $P = 1$, all the DT-SQA replicas are connected and we perform a single run of the algorithm to match (and exceed as needed) QA slopes. The large number of replicas necessary to match the QA slopes is strong evidence of the efficiency of the quantum annealer, nonetheless, our results show that DT-SQA, when equipped with sufficient replicas, can achieve a comparable or more favorable scaling exponent. Naturally, the second approach where we pick the best results out of $P > 1$ runs can also be applied to the quantum annealer to increase residual energy slopes, however, this does not change our main findings for $P = 1$.

Finally, we note that while $\kappa_f$ is a useful metric, it may not fully capture the practical relevance of algorithms for large-scale optimization. As discussed in the next subsection, DT-SQA, despite exhibiting a steep power-law decay in residual energy at early times in instances defined on a fixed 3D spin glass lattice of size $15 \times 15 \times 12$, transitions into a plateau at longer annealing times, where the residual energy stagnates and shows little further improvement. It is ultimately outperformed by the APT algorithm, which is easier to implement and

(a)
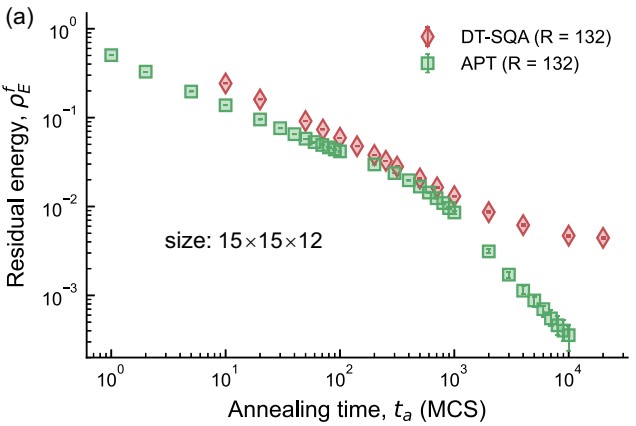

(b)

size: 15×15×12

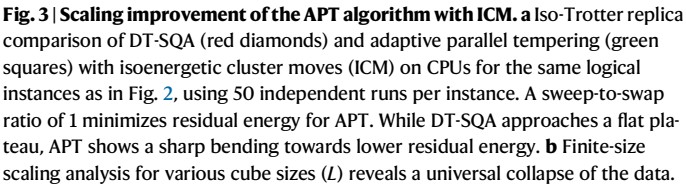

**Fig. 3 | Scaling improvement of the APT algorithm with ICM. a** Iso-Trotter replica comparison of DT-SQA (red diamonds) and adaptive parallel tempering (green squares) with isoenergetic cluster moves (ICM) on CPUs for the same logical instances as in Fig. 2, using 50 independent runs per instance. A sweep-to-swap ratio of 1 minimizes residual energy for APT. While DT-SQA approaches a flat plateau, APT shows a sharp bending towards lower residual energy. **b** Finite-size scaling analysis for various cube sizes (L) reveals a universal collapse of the data.

The later bending appears to be universal for APT, as shown by averaging over 200 independent runs per instance. For each size, 4 ICM replicas are used, with the total number of replicas varying due to the adaptive nature of the pre-processing algorithm. A similar collapse for APT without ICM is provided in the Supplementary Fig. S19. Error bars represent the 95% bootstrap confidence interval of the mean across spin-glass instances.

parallelize in hardware. APT achieves significantly lower residual energies with identical computational resources, showing that relying solely on $\kappa_f$ as a performance metric can be misleading, as different algorithms may exhibit distinct scaling behaviors at different stages of optimization, which are crucial for real-world applications.

## Comparison with adaptive parallel tempering

We now compare DT-SQA with the powerful replica-based adaptive parallel tempering (APT) algorithm, widely considered as the state-of-the-art for solving spin-glass problems[40–46]. APT also utilizes replicas of the problem graph, but these run in parallel at different temperatures, with adjacent replicas periodically swapped based on the Metropolis criterion:

$$p_{swap} = \min(1, \exp(\Delta\beta\Delta E)), \tag{4}$$

where $\Delta E = E_{i+1} - E_i$ is the energy difference, and $\Delta\beta = \beta_{i+1} - \beta_i$ is the difference in inverse temperatures between replicas $i$ and $i + 1$ (with $\beta_i < \beta_{i+1}$). This mechanism enables high-temperature replicas to explore the energy landscape broadly while low-temperature replicas preserve optimal states. The adaptive variant further optimizes the algorithm by preprocessing the problem graph to equalize swap probabilities across replicas, avoiding bottlenecks[46,47].

Figure 3a compares DT-SQA and APT for the same problem as in Fig. 2. Adaptive preprocessing produces approximately 33 temperature replicas per instance (see Methods Section). To further enhance APT, we incorporate isoenergetic cluster moves (ICM)[33–35], which, as we demonstrate later, play a crucial role. ICM are non-local Monte Carlo swaps added on top of the standard APT algorithm. A swap attempt follows each network sweep, maintaining a sweep-to-swap ratio of 1. Using 4 replicas per temperature for ICM, the APT algorithm used in this work operates with a total of 132 replicas. As shown in Supplementary Fig. S16, we found that this sweep-to-swap ratio produces the smallest residual energy for a fixed MCS budget.

Optimization of APT parameters (detailed in Supplementary Section IV) reveals that the initial slope of the optimized APT with ICM corresponds to $\kappa_f = 0.53$, slightly lower than DT-SQA with a similar number of replicas ($\kappa_f = 0.647$ at $R = 132$). However, APT achieves lower residual energy for a given MCS budget. Although DT-SQA initially shows a better slope, it plateaus at higher MCS, as shown in Fig. 3a. This trend is observed across various cube sizes $L$ and

Trotter replicas $R$ (see Supplementary Fig. S3) and is consistent with previous findings[38]. In contrast, APT with ICM shows two distinct scaling regimes: an initial gentler slope followed by a steeper one (see Supplementary Figs. S14a and S15). Notably, the APT algorithm without ICM does not exhibit this steeper bending, even with the same number of replicas (see Supplementary Fig. S14a and Supplementary Fig. S19). The presence of this bending suggests the potential for algorithms that incorporate non-local and non-equilibrium moves[48] to further enhance the performance of probabilistic approaches in solving hard optimization problems. As before, a similar performance characteristic is observed for embedded instances (see Supplementary Fig. S17).

This steeper slope is also observed for other cube sizes $L$ (see Supplementary Fig. S18a) and appears to be a universal feature of APT supplemented by ICM. Finite-size scaling analysis confirms that the residual-energy curves for different sizes collapse onto a single universal curve (Fig. 3b). However, at very low residual energies near the ground state, we observe another transition to a gentler slope (not visible in Fig. 3b). This transition occurs at residual energies that are very close to the uncertainty limit of the ground energy estimations used in our analysis (see Supplementary Fig. S18b). As such, it is difficult to reliably confirm the existence of this feature. On the other hand, the robust universal collapse shown in Fig. 3b allows us to extrapolate the time required to reach a target residual energy for cubes of any size.

## Architecture-based scaling improvement and massive parallelism

Beyond scaling improvements, a critical metric for optimization problems is achieving the lowest possible residual energy within a given amount of time. Here, we evaluate the relative performance of CPU, FPGA (see Supplementary Section V for the details of FPGA implementation), and sMTJ-based implementations of p-computers, highlighting architectural advantages. One key feature of the p-computer architecture we adopt here is the ability to probabilistically update all spins in the system in constant time. This differs from the sequential or partially parallel updates typically used in software implementations. In sparse problems, such as 3D spin-glasses, planted Ising benchmarks[15,49] or circuit SAT problems with sparse connectivity[4], p-computer architectures leverage physically parallel nodes to simultaneously update large independent sets[4]. Figure 4a shows the number

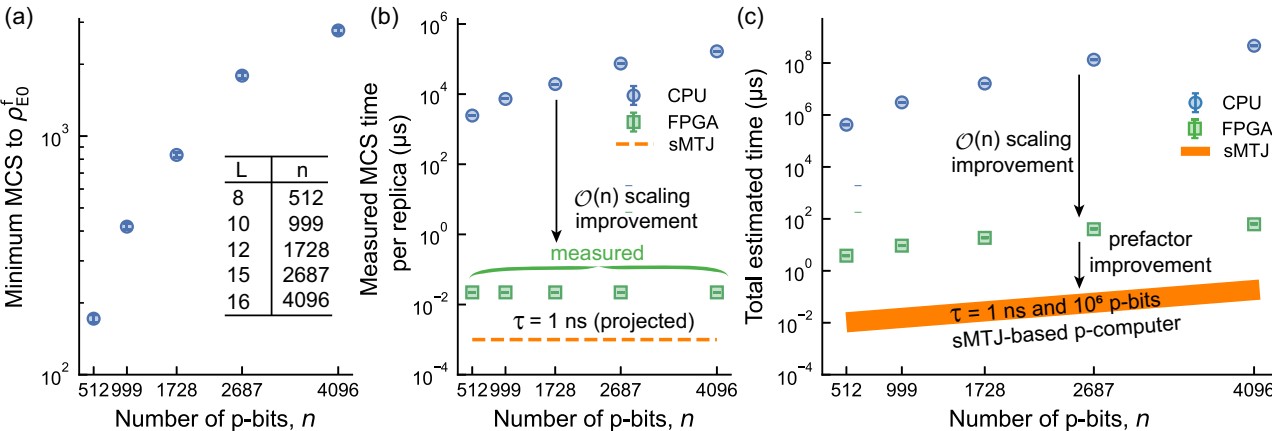

**Fig. 4 | Architectural improvement with probabilistic computers in hardware.
a** Algorithmic complexity of the 3D spin glass problem as a function of the number of p-bits, showing the minimum number of Monte Carlo sweeps (MCS) required to reach a target threshold residual energy, $\rho_{E0}^{f} = 0.007$, using APT + ICM, independent of hardware platform. For each lattice size ($L$), 300 logical instances with 50 independent runs are reported. Swaps are probabilistically performed pairwise between adjacent $\beta$ replicas, alternating between even and odd pairs. Error bars (95% confidence) are small and often invisible. **b** Measured average MCS time per replica for CPU (blue circles) and FPGA (green squares). CPU shows $\mathcal{O}(n)$ scaling, while FPGA achieves $\mathcal{O}(1)$ scaling due to the massively parallel p-computer architecture. sMTJ projections (orange dashed line) assume experimentally demonstrated nanosecond p-bits[61,62]. **c** Total estimated time to reach $\rho_{E0}^{f} = 0.007$, combining (a, b). Time for ICM and swaps are negligible compared to sampling times (see Methods section) and are excluded from this estimation. Also, it is

assumed that FPGA (green squares) can run all the replicas in parallel (as long as they fit on a single chip[28]; see Supplementary Section III). We assume the same for CPU (blue circles) as well and do not include the replica factor when estimating these times. FPGA maintains $\mathcal{O}(n)$ improvement over CPU, while prefactor improvement for sMTJ-based p-computers (orange solid line) refers to the additional constant speed-up expected when the same architecture is implemented with fast, on-chip sMTJ p-bits (approximately 1 ns intrinsic flip time), relative to the measured FPGA sweep time. sMTJ projections assume a single chip with 1 million sMTJs, achieving 1 million flips per ns (the thick orange line assumes 10 and 50 replicas for the upper and lower bounds, respectively). This also includes improvements stemming from additional parallelization, meaning that if the problem size is smaller than the chip's capacity, multiple independent runs or problem instances can be processed simultaneously on the same chip, scaling as $10^6$/ (spins per replica × APT replicas).

of Monte Carlo sweeps (MCS: one MCS involves one update attempt per spin for all spins in the network) needed to reach a target residual energy threshold of $\rho_{E0}^{f} = 0.007$, an arbitrarily chosen optimization goal given a computational budget. Using the APT + ICM algorithm, we show the required number of MCS as a function of the lattice size $L$. Figure 4b compares the time required for one sweep per replica across three architectures: CPU, FPGA-based p-computers, and sMTJ-based p-computers. While sweep times increase with problem size on CPUs, they remain constant, $\mathcal{O}(1)$, for FPGA- and sMTJ-based p-computers, exploiting massive parallelism until resource limits are reached. For the problem considered, $n/2$ or $n/4$ spins (depending on whether logical or embedded instances are used) can be updated simultaneously, resulting in an $\mathcal{O}(n)$ performance improvement over CPUs. The current FPGA implementation yields up to 185 flips/ns approaching performance of the state-of-the-art GPUs and FPGA-based simulators[16,50], with further possible improvements using specialized ASICs and nanodevices. Note however, that the p-computers we propose here can support arbitrary sparse graph topologies beyond the regular and more easily parallelizable topologies shown in[16,50]. Figure 4c combines results from (a) and (b), showing total runtime as the product of sweep count and average sweep time. The analysis confirms an $\mathcal{O}(n)$ scaling advantage for p-computers over CPUs. Furthermore, sMTJ-based devices could achieve an additional 1 to 3 orders of magnitude improvement in prefactors, assuming nanosecond fluctuations in a 1-million p-bit MRAM chip, which are feasible.

The proposed architecture is also highly energy efficient, consuming 2 to 5 orders of magnitude less energy per flip compared to the state-of-the-art GPUs and TPUs used for probabilistic tasks. Our FPGA implementation consumes 9.168 W, which corresponds to $5 \times 10^{-2}$ nJ/flip. sMTJ-based devices, with 1 million p-bits integrated on a single MRAM chip, reduce this further to $2 \times 10^{-5}$ nJ/flip assuming 20 W power consumption[51]. In comparison, an NVIDIA Tesla V100 GPU consumes 21.99 nJ/flip, and a Google TPU v3 requires 7.77 nJ/flip to solve probabilistic problems on simpler graphs[25].

There is a trade-off between reconfigurable and application-specific hardware. FPGAs offer full reconfigurability, ideal for algorithmic exploration across diverse problem structures, but at a significant performance and energy cost. For the 3D spin glasses studied here, the fixed nearest-neighbor topology is an excellent match for a custom ASIC. Connectivity can be hard-wired while programmability for different instances is retained by reloading different weights. For problems with arbitrary sparse topologies, however, achieving reconfigurability on static ASICs is an open problem and may require different approaches, such as higher-order problem formulations or master graph approaches[28,52].

### Physical design feasibility of single p-computing chips
We now evaluate the feasibility of a custom Application-Specific Integrated Circuit (ASIC) designed for replica-based algorithms on sparse, structured problems. A monolithic chip that can house all replicas on-die would eliminate the off-chip communication overhead that constrains current FPGA implementations. To make realistic projections for a full-scale ASIC, our analysis is grounded in a rigorous, bottom-up physical design flow using a 7 nm process. Our findings on chip capacity for the DT-SQA algorithm apply equally to the better performing APT algorithm, which requires significantly fewer replicas, as shown in the subsection where we compared APT results with DT-SQA.

The details of the p-computer architecture are discussed in Supplementary Section III. As shown in Supplementary Fig. S11 and Supplementary Table S1, a full place-and-route analysis was performed using the ASAP7 7 nm open-source process design kit (PDK)[53] for up to 5 replicas of the 15 × 15 × 12 logical instances. The analysis revealed an approximately linear growth (with a slope of 1.05) in chip area requirements.

Based on this scaling, we project that approximately 7.66 million p-bits—corresponding to 2850 replicas can fit on a single chip using 7 nm technology. This translates to a chip area of 28.61 × 28.61 mm², which is within the capabilities of current fabrication technology.

Furthermore, multiple such chips can be interconnected to support even larger number of p-bits as needed. With advances in fabrication technology and the adoption of nanodevice-based p-bits, the number of p-bits per chip can be significantly increased, enabling even greater scalability.

## Outlook

This paper demonstrates that probabilistic computing with p-bits provides a practical and scalable approach to solving 3D Ising spin glass problems. Using the DT-SQA algorithm, we showed how leveraging a large number of replicas greatly improves scaling exponents, well-explained by extreme value theory. We further explored powerful algorithms like APT supported with non-local moves, significantly improving scaling and time-to-solution. Finite-size scaling analysis revealed a universal collapse of residual energy curves for APT, emphasizing the generality of our results. This makes APT particularly well-suited for large-scale optimization tasks when implemented on dedicated probabilistic hardware, as demonstrated by our FPGA-based implementation, achieving high performance through hardware acceleration.

Advances in fabrication technology now allow large-scale replica systems, delivering orders-of-magnitude speedups compared to software methods. Projections for monolithic nanodevice-based p-computers highlight a path toward performance competitive with and potentially exceeding current solvers, all while operating at room temperature and without the specific hardware challenges of qubit decoherence or fixed-connectivity embedding. Co-designed together, powerful algorithms, scalable architectures, and emerging hardware provide a clear pathway for solving hard optimization problems at unprecedented scales. Beyond combinatorial optimization, probabilistic computers hold promise for diverse applications including training and inference in energy-based models and Bayesian learning and in general for sampling over discrete spaces where the performance of traditional solvers have saturated.

## Methods

### p-computing overview

p-computing relies on an interacting network of p-bits $\sigma_i$, which generate two-valued outputs ($\sigma_i \in \{-1, +1\}$) and are governed by two key equations[54]:

$$I_i = \sum_j J_{ij}\sigma_j + h_i \qquad (5)$$

$$\sigma_i = \text{sgn}\left(\tanh(\beta I_i) - r_{[-1, 1]}\right) \qquad (6)$$

Here, $J$, $h$, and $\beta$ represent the interconnection matrix, bias vector, and inverse temperature, respectively. $r_{[-1, 1]}$ is a random number uniformly distributed in the range $[-1, 1]$. Equations (5) and (6) collectively approximate the Boltzmann distribution:

$$p(\{\sigma_i\}) = \frac{1}{Z}\exp(-\beta E(\{\sigma_i\})) \qquad (7)$$

$$E(\{\sigma_i\}) = -\sum_{i<j} J_{ij}\sigma_i\sigma_j - \sum_i h_i\sigma_i \qquad (8)$$

where $p(\{\sigma_i\})$ represents the probability and $E(\{\sigma_i\})$ represents the energy of the state $\{\sigma_i\}$, with $Z$ as the partition function.

### Instances

For comparison, we use the instances from ref. 18 for $L = 8$, 10, and instances of size $15 \times 15 \times 12$. These instances have open boundaries along the $x$ and $y$ directions and periodic boundaries along the $z$ direction. For sizes greater than $L = 9$, some spins are missing due to embedding constraints of the quantum annealer. Consequently, the total number of qubits for $L = 10$, for example, is 999, instead of the expected $L^3 = 1000$. We also use the putative ground energies reported in ref. 18 for these sizes. However, for a few instances, we found lower ground state energies than those reported and therefore used the improved ground energies in our analysis.

For $L = 12$ and 16, we generate instances using the codes provided in ref. 18, ensuring that these instances do not suffer from missing spins. For these instances, the putative ground state energies are obtained by running APT with ICM algorithm up to $10^7$ sweeps (in each sweep all replicas are updated once), in a single run with a sweep-to-swap ratio of 10, choosing the minimum energy found along the whole simulation, and following a similar fitting and limiting procedure discussed in[18]. Our corresponding approximate error estimate per site for these sizes, is $2.5 \times 10^{-4}$ (attributed to the increased problem size and the use of a single run) as indicated in Supplementary Fig. S18. This is comparable to the estimated error in the mean ground state energy per site, $4 \times 10^{-5}$, as reported in ref. 18 for lattice size $15 \times 15 \times 12$. The residual energy ranges used in this work to draw our conclusions are well above the range of these errors, or otherwise carefully discussed.

### Graph coloring of 3D cubic spin glass instances

If two p-bits in a network are not connected, they can be updated in parallel[4]. Graph coloring assigns colors to the network such that any two connected p-bits are given different colors, while p-bits that are not connected can share the same color. This enables massive parallelism for sparse graphs even if they are irregular, allowing a network of p-bits to be updated in constant time, regardless of network size.

A perfect 3D lattice is bipartite and easily 2-colorable. However, the D-Wave instances have missing spins and complex embeddings (due to hardware constraints), which necessitates graph coloring. In this work, graph coloring is performed using DSATUR[55], a heuristic graph coloring algorithm with polynomial-time complexity. Since the underlying graph is identical for all problem instances of a given size, we perform graph coloring for one representative instance of each size as a preprocessing step. These problem instances typically require 2 to 4 colors, depending on their connectivity. In DT-SQA, replicas are connected and periodic boundary condition is applied. As a result, networks with odd number of replicas require an extra color.

### Annealing schedule of DT-SQA

Supplementary Section I details the description of the DT-SQA algorithm. Annealing is performed by gradually changing the transverse field ($\Gamma_x$) from a high value to 0. Change in $\Gamma_x$ is reflected in the coupling strength $J_\perp$ (see Supplementary Eq. (S.3)), which couples the spins of two neighboring replicas.

In our implementation, we use a slightly modified form for $J_\perp$:

$$J_\perp(t) = -\frac{1}{\beta}\ln\tanh\left(\frac{\beta\Gamma_x'(t)}{R}\right) \qquad (9)$$

and anneal $\Gamma_x'(t)$ linearly, from 3.0 to 0. This modification represents a transformation between $\Gamma_x'$ and $\Gamma_x$ and does not alter the underlying physics. We also set $\beta/R = 0.5$ in all our simulations.

### APT details

For the APT algorithm, we start with a preprocessing step to compute the inverse temperature ($\beta$) schedule and determine the required number of replicas. We perform the preprocessing individually for each of the 300 instances, even though schedules and number of replicas obtained are similar (see Supplementary Section IV A). Specific details of the preprocessing algorithm we adopted can be found in[28,48] and Supplementary Algorithm S2. For our simulations, we set the initial inverse temperature to $\beta_0 = 0.5$ and the temperature update factor to

$\alpha$ = 1.25. We calculate the average energy variance across 100 parallel chains, where the variance for each chain is computed from the last 1000 sweeps of a 10000-sweep run before updating the temperature schedule. This process is repeated until the average energy variance drops below $\min(|J_{ij}|)/2$. For the 300 instances with lattice size $15 \times 15 \times 12$, the number of replicas ranged from 32 to 34.

After determining the $\beta$ schedule, each instance is simulated using the parallel tempering algorithm, both with and without the iso-energetic cluster moves (ICM)[33–35]. We employ 4 ICM replicas per temperature. During simulation, each replica undergoes a fixed number of sweeps before a swap is attempted. A swap attempt involves performing an isoenergetic cluster move for each of the two randomly chosen ICM replica pairs at each temperature. This is followed by a swap attempt between neighboring replicas, determined as follows: for an odd-numbered swap attempt, pairs (1, 2), (3, 4), ... are swapped; for an even-numbered swap attempt, pairs (2, 3), (4, 5), ... are swapped. Within a given $\beta$, ICM replicas are labeled $a$, $b$, $c$, $d$, and swaps between neighboring $\beta$ values occur only between replicas with the same label. The algorithm is detailed in Supplementary Algorithm S2.

### APT collapse
The collapse of APT with ICM residual energies was obtained using the open-source library autoScale.py[56]. The parameter $b$ fluctuates slightly around 3.0; we use $b = 3.0$, as it intuitively reflects the fact that the residual energy behaves extensively, scaling with the system size, $L^3$.

### CPU details
All CPU-based simulations were run on a 10-core Intel Core i9-10900 processor (2.80 GHz) with 64 GB RAM, using MATLAB R2023b on a 64-bit Windows 10 machine. p-bits were updated sequentially using Gibbs sampling. All computations used MATLAB's default double-precision arithmetic and Mersenne Twister pseudorandom number generator (PRNG).

### FPGA details
We mapped the physics-inspired, massively parallel p-computer architecture of ref. 4 onto a Xilinx Alveo U250 data-center accelerator card using graph coloring to maximize parallelism on the sparse instances. All arithmetic is fixed-point: DT-SQA uses s{6}{3} precision (1 sign, 6 integer, 3 fractional bits) while APT + ICM uses the higher s{6}{6} precision. Custom RTLs were developed to implement the algorithm based on the p-computing architecture and synthesized, placed and routed with Xilinx Vivado/Vitis tool chain. Further details are provided in Supplementary Sections III (DT-SQA) and V (APT).

### FPGA implementation of APT with ICM algorithm
At this time, we can accommodate only 1 replica in a single FPGA for large scales, such as instances of size $15 \times 15 \times 12$. To implement APT for this size, 32 to 34 replicas are required, while APT with ICM requires 128 to 136 replicas. We address this limitation by employing time-division multiplexing (TDM), allowing the same hardware to be reused for multiple replicas. At the start of each run, the weights of all replicas (scaled by $\beta$) are loaded into the BRAM. During each sweep, based on the replica index that will be sampled next, the weights corresponding to that replica are dynamically fetched from the BRAM. Then after the sampling is done, the state of p-bits of the current replica are also stored in the BRAM. This process is repeated until all replicas are sampled once. Then from MATLAB, we read all the p-bits states, (perform ICM whenever applicable) compute energies and perform the swaps. For the subsequent swap attempts, replicas are reinitialized either from the state saved in the previous swap or from the new biases (to restore certain p-bits to their original states before hardware was reused) generated by the APT swap/ICM. After initialization, the biases are released, and the p-bits resume their usual MCS at their respective $\beta$.

### Measurement of flips per second and time per MCS
To measure the time per MCS in the FPGA accurately, we implemented precise counters within the FPGA to track the number of flip attempts made by each p-bit during a fixed time interval. This reference interval is determined by a predefined counter running at 125 MHz, which counts up to 50,000. All counters are simultaneously enabled by a global signal from MATLAB and stop when the reference counter completes its count. This corresponds to an elapsed time of $50000/(125 \times 10^6)$ seconds ($400\,\mu s$). The total flip attempts during this period are summed across all p-bit counters to compute the total flips per second (fps).

Since one MCS involves updating all p-bits in a single replica, the time per MCS is calculated using the total measured flips and the elapsed time. For each network size $L$, we performed 100 measurements and reported the average time per MCS. FPGA measurements are instance-independent. We reported MCS times for a single replica, given our detailed feasibility analysis that shows all relevant replicas sizes we considered can fit on a single chip. This full integration would eliminate the overhead for time-division multiplexing. It should be noted that, at present, the overhead from swapping and ICM moves dominates the MCS time because it is performed off-chip on the CPU. This is not a fundamental limitation: the overhead can be computed directly on the FPGA rather than off-chip on the CPU. Using standard hardware design flows, such as those enabled by high-level synthesis (HLS), these computations can be seamlessly implemented on-chip, thus making their contribution negligible compared to the MCS time. Consequently, it was excluded from the FPGA measurements. We emphasize that none of the overheads that are omitted here affect our scaling exponent measurements where prefactors in time per MCS do not affect slopes in power laws.

For CPU measurements, MATLAB's built-in tic and toc functions were used to measure the time taken to perform 100 MCS across 10 instances with 10 runs each. The average time per MCS is reported for a single replica of each network size $L$. Swap times were similarly excluded from CPU measurements to ensure a fair comparison with FPGA performance. For both FPGA and CPU measurements, error bars represent 95% confidence intervals and were computed using bootstrapping with replacement.

## Data availability
All generated and processed data used in the plots within this paper can be found in the GitHub repository[57]. Other findings of this study are available from the corresponding authors upon request.

## Code availability
Simplified MATLAB implementations of the SQA and APT + ICM algorithms and the instances used in this study are openly available at the GitHub repository mentioned in the Data Availability Section. Pseudocode for both routines is provided in Supplementary Algorithms S1 and S2.

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

## Acknowledgements

SC, KYC and NAA acknowledge support from the Office of Naval Research (ONR) Young Investigator Program grant, the National Science Foundation (NSF) CAREER Award under grant number CCF 2106260, the Army Research Laboratory under grant number W911NF-24-1-0228, the Semiconductor Research Corporation (SRC) grant, and the ONR-MURI grant N000142312708. Use was made of computational facilities purchased with funds from the National Science Foundation (CNS-1725797) and administered by the Center for Scientific Computing (CSC). The CSC is supported by the California NanoSystems Institute and the Materials Research Science and Engineering Center (MRSEC; NSF DMR 2308708) at UC Santa Barbara. AG, ER and GF acknowledge the support of project number 101070287 – SWAN-on-chip – HORIZON-CL4-2021-DIGITAL-EMERGING-01; the MUR-PNRR project SAMOTHRACE (ECS00000022), funded by European Union (NextGeneration EU); the projects PRIN 2020LWPKH7 - The Italian factory of micromagnetic modeling and spintronics and PRIN 20225YF2S4 - Magneto-Mechanical Accelerometers, Gyroscopes and Computing based on nanoscale magnetic tunnel junctions (MMAGYC), funded by the Italian Ministry of University and Research. PAL, NAA and MM were supported in part under NSF CCF (grant 1918549). PAL was also supported in part through the NASA Academic Mission Services (contract NNA16BD14C) under SAA2-403506, as well as the Intelligent Systems Research and Development-3 (ISRDS-3) Contract 80ARC020D0010 under SAA2-403688. JHM acknowledges funding from the VIDI project no. 223.157 (CHASEMAG) and KIC project no. 22016 which are (partly) financed by the Dutch Research Council (NWO), as well as support from the European Union Horizon 2020 and innovation program under the European Research Council ERC Grant Agreement No. 856538 (3D-MAGiC) and the Horizon Europe project no. 101070290 (NIMFEIA). FRT received financial support from the "National Centre for HPC, Big Data and Quantum Computing - HPC", Project CN_00000013, CUP B83C22002940006, NRP Mission 4 Component 2 Investment 1.5, funded by the European Union - NextGenerationEU. TS and AR acknowledge support from NSF FuSe2 Award 2425218, Carnegie Mellon University Dean's Fellowship and Tan Endowed Graduate Fellowship in Electrical and Computer Engineering, Carnegie Mellon University. TS and AR also acknowledge Tong Wu for discussions.

## Author contributions

S.C., M.M. and K.Y.C. conceived the study. K.Y.C. supervised the study. S.C., A.G., E.R. and N.A.A. performed different parts of the simulations. N.A.A. and S.C. performed FPGA experiments. A.R. and T.S. performed the physical design simulations for chip design. M.M.R., F.R.T. and M.M. analyzed the scaling data with critical feedback. S.C. wrote the initial draft of the manuscript with inputs from P.A.L., J.H.M., M.M.R., F.R.T., M.C., L.S.T., G.F., M.M. and K.Y.C. All authors contributed to improving the draft and participated in designing the experiments, analyzing the results, and editing the manuscript.

## Competing interests

The authors declare no competing interests.
