## [Transparent Peer Review file · Nature Communications]

Pushing the Boundary of Quantum Advantage in Hard Combinatorial Optimization with Probabilistic Computers

Corresponding Author: Professor Kerem Yunus Camsari

Version 0:

Reviewer comments:

Reviewer #1

(Remarks to the Author)

This manuscript constitutes a substantive and timely contribution to the quantum-computing literature. By delivering a meticulous head-to-head assessment of large-scale quantum annealing against state-of-the-art classical heuristics for many-body spin-glass Hamiltonians, it advances the debate well beyond the “single-headline benchmark” paradigm that has often dominated the field. The authors’ protocol—combining finite-size scaling, extreme-value statistics, and hardware-aware resource accounting—parallels, in spirit and interest, the community’s most influential gate-model benchmarking campaigns (e.g., the recent cross-platform comparisons of random-circuit sampling and Fermi-Hubbard dynamics). Such a comprehensive framework is precisely what is needed to clarify where genuine quantum advantages reside and where classical strategies—tensor-network methods, Monte Carlo cluster moves, or p-bit accelerators—still prevail. Because the study situates quantum annealing on the same footing as contemporary CPU/GPU and FPGA-based solvers, it provides an evidence-based roadmap for research groups evaluating which computational paradigm best suits their target Hamiltonians. I therefore expect the paper to resonate broadly—not only with quantum-annealing specialists, but also with condensed-matter physicists, computational chemists, circuit-design and hardware-architecture engineers, high-performance-computing experts, optimization scientists, and the wider quantum-simulation community. In short, this is an exemplary piece of comparative science that sharpens our collective understanding of the practical limits—and latent potential—of quantum hardware. Quantum devices, by their very nature, require specialized hardware. In the same spirit, classical computing warrants tailored algorithms, dedicated hardware, and hardware-software co-optimization.

I strongly recommend this manuscript for publication in Nature Communications, as it presents significant and original contributions to the field. Although the manuscript is already of high quality, I offer the following comments for the authors’ consideration, with the aim of further enhancing its clarity and completeness.

* The main results appear to have been obtained on a CPU (see, for example, Figure 2). How does the reported “annealing time” translate into CPU wall-clock time? The authors should also provide full details of the computational environment used for the CPU simulations.

* The manuscript is expected to explain in detail how “annealing time” on the quantum annealer (a physical time in μs) is converted to, or compared with, the Monte-Carlo-sweep (MCS) count used for the CPU, GPU, FPGA, and ASIC results. Even if the procedure follows King et al., the present paper needs to restate the rationale and the exact conversion, because it is crucial in any cross-platform benchmark. Moreover, if MCS is retained as the only performance metric, the reader cannot directly see the speed gap among CPU, GPU, FPGA, and ASIC hardware. Is MCS, by itself, an appropriate metric in this context?

* DT-SQA appears to outperform the SQA baseline reported in King’s paper (based on the power-law fit). I presume you first reproduced King’s original SQA results to ensure consistency. Could the authors specify, in detail, how much of the total performance gain comes from each modification—for example, increasing the number of Trotter replicas or applying

extreme-value statistics (EVT)?

* Although using an off-the-shelf FPGA is straightforward, a custom ASIC—despite its potential efficiency—seems constrained in this study by die-size limits and off-chip communication overhead. Could the authors comment more on how these factors restrict its practical use?

* A classical solver could in principle be implemented on dedicated hardware such as an ASIC. However, programmability and flexibility—including the constraints of fixed routing—could then become significant issues. Could the authors discuss the trade-off between chip-level flexibility and performance (speed or energy efficiency) in this context?

* The figures inconsistently interchange the notations R , $R \times P$, and R_T .

Reviewer #2

(Remarks to the Author)

The manuscript “Pushing the Boundary of Quantum Advantage in Hard Combinatorial Optimization with Probabilistic Computers” revisits claims of scaling advantage by quantum annealers (QAs) on hard combinatorial optimization benchmarks (specifically 3D Edwards–Anderson spin glasses) and asks whether probabilistic computers (“p-computers”) running advanced Monte Carlo algorithms can match or exceed QA performance on the same problems. The authors focus on two replica-based algorithms such as Discrete-Time Simulated Quantum Annealing (DT-SQA) and Adaptive Parallel Tempering (APT) with Isoenergetic Cluster Moves (ICM) and co-design them with dedicated p-bits hardware architectures (CPUs, FPGAs, and projected SMTJ chips).

This is an important paper because it raises the bar for claims of “practical quantum advantage” in optimization by showing that classical p-computers, when co-designed with replica-based algorithms and hardware architectures, can match or exceed current QA performance on benchmark spin glasses. The authors emphasize that observing quantum-critical dynamics is neither necessary nor sufficient for superior real-world optimization: algorithmic innovations (e.g., ICM) and hardware parallelism can deliver comparable or better speedups. They project that further nanodevice integration (SMTJs) and ASICs will extend these advantages to larger problems. So I am in favour of publishing this paper. My comments are mostly to make the story more coherent.

Comments:

(1) Eq. (1) could be read as summing over all pairs (i, j) . Since the paper’s benchmarks are 3D spin glasses with only nearest-neighbour (NN) couplings, the authors should explicitly state that the J_{ij} are nonzero only for NN pairs on the lattice.

(2) It isn’t clear from the current text what κ_f represents when you first see Eq. (3). Perhaps, say “where κ_f is the fitted scaling exponent of the residual-energy decay.” Or something along these lines.

(3) The top paragraph on the right panel of page 3 relies heavily on specialized terminology without defining key concepts, which may confuse readers. E.g. It refers to “the density of defects” but never explains what a defect is in this context (e.g., domain walls, kinks, spin flips). A brief definition is needed. The term “correlation length” is never clarified. The authors should state how it’s defined and measured (for instance, the characteristic distance over which spins are correlated). “Binder cumulant” is mentioned as scaling with KZM predictions but not defined. A short explanation (e.g., a dimensionless ratio used to quantify critical fluctuations) or a reference would help. The paragraph contrasts the residual energy exponent with the KZM prediction but doesn’t remind the reader of its definition or where it appears (e.g., Eq. (3)). A cross-reference or brief recap would help.

(4) “using logical instances of size $15 \times 15 \times 12$ ” is potentially opaque. Perhaps it is better to say, “We generate logical problem instances on a $15 \times 15 \times 12$ 3D lattice of Ising spins.”

(5) Define “broken dimers” and “embedded instances”.

(6) “Artificially include embedding overheads”: do the authors mean “we omit the additional time and resource costs needed to embed problems on QA hardware”?

(7) “There is no such concern ... however.” The “however” feels detached and the double negative (“no such concern ... since quantum emulation is not a target”) makes the logic a bit hard to follow on first read. Also, the authors begin by talking about Monte Carlo simulations “whose goal is to faithfully simulate ...,” then switch to “however” about optimization problems, and then immediately refer to “our replicas” and “physical spins.” It might help to split these ideas.

(8) The authors say DT-SQA “transitions into a plateau” : do they mean its residual-energy decay stagnates after some time? Be explicit.

(9) What does $L=15$ mean? The lattice size was quoted as $15 \times 15 \times 12$. May be “we study 3D lattices whose two longest sides have length $L=15$.” Or “We consider 3D spin glasses on lattices of size $L_x \times L_y \times L_z$, with $L_x=L_y=15$, and $L_z=12$.”

(10) Not every reader will recall what stands for “Monte Carlo sweeps.”

(11) “Slope” vs. “plateau”. Do the authors mean DT-SQA’s residual-energy decay exponent is larger at first (steeper decline) but then levels off (little further improvement) at long runtimes?

(12) The phrase “absolute residual errors” is potentially confusing for two reasons: (1) is there a single quantity (the residual energy per instance)? In this case “errors” (plural) feels odd. (2) It’s unclear whether the authors mean the absolute value of a signed residual or simply the raw, non-normalized residual energy. Also, ‘Residual energy vs. Residual error. I do not understand this: residual energy = $\epsilon_{\text{res}} = E(t) - E_{\text{min}}$ is always nonnegative, so the adjective “absolute” seems redundant.

(13) The community often benchmarks QA and DT-SQA on the standard 3D tile graphs that have a controllable hardness [Hamze, Firas, et al. "From near to eternity: spin-glass planting, tiling puzzles, and constraint-satisfaction problems." Physical Review E 97.4 (2018): 043303]. Could the authors comment on their choice of Edwards–Anderson spin glass instead? Specifically, how do their conclusions extend to or differ on the tiles3D benchmark?

(14) Algorithm S1 is based on the colouring problem but the paper never explicitly explains that “colours” correspond to graph-colouring classes used to identify independent spin sets for parallel updates. It simply presents the loop over colours without motivation.

(15) Caption for Fig 4: what is “prefactor improvement”? is it a ratio of CPU to sMTJ-p-computer sweep times?

(16) Fig S3: Define “95% bootstrap confidence interval” (e.g. over 300 instances).?

(17) Fig. S15: Define “Houdayer move”. Add, perhaps, “A Houdayer move [33] is an isoenergetic cluster swap in which one identifies a cluster of antiparallel spins between two replicas and exchanges them to enhance mixing in spin-glass Monte Carlo.”

(18) Time-to-solution is introduced in the Introduction, but the precise definition (e.g. time to reach a target residual energy with 99% probability) is only buried in the Methods.

(19) Supplementary Fig. S2 caption: “ $15 \times 15 \times 12 \times 2$ ” If it is a chair length for embedding, it should be described as such.

(20) Which FPGA board was used, what precision was assumed, or how the logic was synthesized?

(21) What are ASIC projections?

(22) sweep-to-swap ratio is introduced in Supplementary Figs. S14–S16 captions without any definition. What exactly is being counted?

(23) Use either n or N for the number of spins consistently. (e.g. Fig. S6 caption uses N , but Eq.(2) uses n .)

Reviewer #3

(Remarks to the Author)
Review attached.

Version 1:

Reviewer comments:

Reviewer #1

(Remarks to the Author)
Overall, the authors responded thoroughly to the reviewers’ questions and made appropriate, substantive revisions. The kf-centered algorithmic comparison is now clearly framed, APT+ICM shows convincing universal scaling, DT-SQA is well positioned for hardware implementation, and the embedding discussion is concise and clarifying. These improvements enhance both scientific rigor and practical relevance. I recommend acceptance.

Reviewer #2

(Remarks to the Author)
The authors have comprehensively addressed the referees’ comments and suggestions through thorough revisions to the manuscript. Based on the quality of these revisions and the strength of the work, I am pleased to recommend this paper for publication.

Reviewer #3

(Remarks to the Author)

After carefully reviewing the authors' point-by-point responses to the reviewers' comments, it appears that the reviewers' major concerns have been sufficiently addressed. Authors have clarified the novelty of their contributions relative to prior work, strengthened the theoretical justification for their benchmarking methodology, and discussed the scalability of their approach with reasonable depth. Minor issues related to wording and typographical errors have also been appropriately corrected. Given the importance of the topic, the improved clarity of the manuscript, and the adequacy of the responses, the reviewer believes that this work is now suitable for publication in Nature Communications.

Re: NCOMMS-25-33449-T

Pushing the Boundary of Quantum Advantage in Hard Combinatorial Optimization with Probabilistic Computers

Shuvro Chowdhury,¹ Navid Anjum Aadit,¹ Andrea Grimaldi,^{2,3} Eleonora Raimondo,^{2,4} Atharva Raut,⁵ P. Aaron Lott,^{6,7} Johan H. Mentink,⁸ Marek M. Rams,⁹ Federico Ricci-Tersenghi,¹⁰ Massimo Chiappini,⁴ Luke S. Theogarajan,¹ Tathagata Srimani,⁵ Giovanni Finocchio,² Masoud Mohseni,¹¹ and Kerem Y. Camsari¹

¹Department of Electrical and Computer Engineering,
University of California, Santa Barbara, Santa Barbara, CA 93106, USA

²Department of Mathematical and Computer Sciences,
Physical Sciences and Earth Sciences, University of Messina, 98166, Messina, Italy

³Department of Electrical and Information Engineering, Politecnico di Bari, 70126 Bari, Italy

⁴Istituto Nazionale di Geofisica e Vulcanologia, Via di Vigna Murata 605, 00143 Roma, Italy

⁵Department of Electrical and Computer Engineering, Carnegie Mellon University

⁶USRA Research Institute for Advanced Computer Science (RIACS)

⁷Quantum Artificial Intelligence Laboratory (QuAIL), NASA Ames Research Center

⁸Institute for Molecules and Materials, Radboud University,
Heyendaalseweg 135, Nijmegen, The Netherlands

⁹Institute of Theoretical Physics, Jagiellonian University, Lojasiewicza 11, PL-30348 Kraków, Poland

¹⁰Dipartimento di Fisica, Sapienza Università di Roma, and CNR-Nanotec,

Rome unit and INFN, Sezione di Roma 1, 00185 Rome, Italy

¹¹Emergent Machine Intelligence, Hewlett Packard Labs, CA, USA

(Dated: July 28, 2025)

I. REVIEWER #1

This manuscript constitutes a substantive and timely contribution to the quantum-computing literature. By delivering a meticulous head-to-head assessment of large-scale quantum annealing against state-of-the-art classical heuristics for many-body spin-glass Hamiltonians, it advances the debate well beyond the “single-headline benchmark” paradigm that has often dominated the field. The authors’ protocol—combining finite-size scaling, extreme-value statistics, and hardware-aware resource accounting—parallels, in spirit and interest, the community’s most influential gate-model benchmarking campaigns (e.g., the recent cross-platform comparisons of random-circuit sampling and Fermi-Hubbard dynamics). Such a comprehensive framework is precisely what is needed to clarify where genuine quantum advantages reside and where classical strategies—tensor-network methods, Monte Carlo cluster moves, or p-bit accelerators—still prevail. Because the study situates quantum annealing on the same footing as contemporary CPU/GPU and FPGA-based solvers, it provides an evidence-based roadmap for research groups evaluating which computational paradigm best suits their target Hamiltonians. I therefore expect the paper to resonate broadly—not only with quantum-annealing specialists, but also with condensed-matter physicists, computational chemists, circuit-design and hardware-architecture engineers, high-performance-computing experts, optimization scientists, and the wider quantum-simulation community. In short, this is an exemplary piece of comparative science that sharpens our collective understanding of the practical limits—and latent potential—of quantum hardware. Quantum devices, by their very nature, require specialized hardware. In the same spirit, classical computing warrants tailored algorithms, dedicated hardware, and hardware-software co-optimization.

I strongly recommend this manuscript for publication in Nature Communications, as it presents significant and original contributions to the field. Although the manuscript is already of high quality, I offer the following comments for the authors’ consideration, with the aim of further enhancing its clarity and completeness.

AUTHOR RESPONSE

We sincerely thank the reviewer for the careful and thorough reading of the manuscript and greatly appreciate the positive comments and constructive feedback from the reviewer. All comments have been addressed in the revised manuscript, and a color-tracked `diff` file highlighting every change is appended for the reviewer’s convenience.

(1) The main results appear to have been obtained on a CPU (see, for example, Figure 2). How does the reported “annealing time” translate into CPU wall-clock time? The authors should also provide full details of the computational environment used for the CPU simulations.

AUTHOR RESPONSE/ACTION

We agree that the details of the CPU environment are important for reproducibility. In the revised manuscript, we have now added a dedicated subsection, “CPU details” in the Methods section which provides the full specifications of the processor, RAM, and software environment used for the simulations.

Regarding the translation of annealing time into a CPU wall-clock time, please see our response to the next point.

(2) The manuscript is expected to explain in detail how “annealing time” on the quantum annealer (a physical time in μs) is converted to, or compared with, the Monte-Carlo-sweep (MCS) count used for the CPU, GPU, FPGA, and ASIC results. Even if the procedure follows King et al., the present paper needs to restate the rationale and the exact conversion, because it is crucial in any cross-platform benchmark. Moreover, if MCS is retained as the only performance metric, the reader cannot directly see the speed gap among CPU, GPU, FPGA, and ASIC hardware. Is MCS, by itself, an appropriate metric in this context?

AUTHOR RESPONSE

The reviewer identifies a crucial point. We want to clarify that **we are not performing a direct time conversion** between MCS and the quantum annealer’s physical time (e.g., we do not assume $1 \text{ MCS} = X \mu\text{s}$). The primary comparison in Fig. 2 is of the dimensionless **scaling exponent**, κ_f , which characterizes the power-law decay of the residual energy ($\rho_E^f \propto t^{-\kappa_f}$). Crucially, κ_f is independent of constant prefactors, which are washed out in a log-log plot.

Since both the quantum annealer (using physical time) and our classical simulations (using MCS) exhibit this power-law behavior, the exponent κ_f serves as a hardware-independent metric of algorithmic performance. The axes in plots like Fig. 2(b) are overlaid simply to provide a direct visual comparison of these slopes. We have added a sentence to the manuscript to make this important distinction clear.

Regarding whether MCS is an appropriate metric on its own, the reviewer is correct that it does not capture the speed differences between classical hardware. This is precisely why our analysis is split into two parts:

- **Algorithmic Scaling (Fig. 2):** We use MCS here to isolate the algorithm’s performance, which is hardware-agnostic.
- **Time-to-Solution (Fig. 4):** Here, we explicitly address the hardware-dependent speed by translating the required MCS into wall-clock time for each platform (CPU, FPGA, etc.).

This two-step approach allows us to decouple algorithmic efficiency from hardware acceleration, providing a clearer comparison.

AUTHOR ACTION

Based on the points above, we have made the following clarifications to the manuscript to improve its clarity and precision:

1. In the main text (Section III, Paragraph 2), where we first introduce the comparison with the quantum annealer, we have added a sentence to explicitly state the nature of the comparison:

“It is important to note that this is a comparison of the dimensionless scaling exponent, κ_f , which is independent of the units on the time axis (MCS for p -computers, nanoseconds for the QA).”

2. In the caption of Fig. 2, we have expanded the explanation of our benchmarking methodology to make the figure’s interpretation self-contained. The revised text now reads:

“An MCS is an algorithmic unit representing one update attempt per spin; hardware differences alter the wall-clock time per sweep but not the scaling exponent κ_f . The comparison with the quantum annealer’s physical time in (b) is therefore a comparison of this dimensionless exponent, not a direct conversion of time units. A full analysis of wall-clock time, which incorporates these hardware-specific prefactors, is presented in Section V and Fig. 4.”

(3) DT-SQA appears to outperform the SQA baseline reported in King’s paper (based on the power-law fit). I presume you first reproduced King’s original SQA results to ensure consistency. Could the authors specify, in detail, how much of the total performance gain comes from each modification—for example, increasing the number of Trotter replicas or applying extreme-value statistics (EVT)?

AUTHOR RESPONSE

Regarding the comparison to the SQA baseline in King et al., it is important to note that their work uses the **continuous-time** simulated quantum annealing (CT-SQA) algorithm. Our work, in contrast, uses the **discrete-time** variant (DT-SQA), **a different algorithm**. We chose DT-SQA deliberately because, as we argue in the revised manuscript, its structure of explicit, interconnected replicas maps directly onto a feasible, massively parallel hardware architecture. While CT-SQA is an important theoretical algorithm, it lacks this straightforward path to hardware implementation.

Therefore, our goal was not to reproduce the CT-SQA results, but rather to benchmark our hardware-motivated DT-SQA algorithm against the final performance of the quantum annealer itself. We have now revised the manuscript to make this distinction clearer.

Regarding the source of the performance gain, our work demonstrates that the scaling exponent can be improved via **two distinct, alternative strategies**, both of which can be explained using extreme value theory (EVT):

1. **Increasing replicas in a single run** ($P = 1, R \gg 1$): This is the approach shown in Figure 2. Here, we run a single, large, interconnected system with a very high number of Trotter replicas ($R=2850$). The performance gain comes from selecting the lowest-energy state from among these many correlated replicas at the end of a single annealing run.
2. **Using multiple independent runs** ($P > 1$, smaller R): This is the alternative strategy discussed in Section III and Supplementary Fig. S5. Here, we run a smaller system ($R = 32$) multiple times independently ($P = 50$) and select the best result from the entire set of runs. The performance gain comes from sampling the tail of the distribution of outcomes across many independent trials.

These are not cumulative modifications but rather two different knobs to enhance performance. Our results show that both approaches can achieve a scaling exponent comparable to the quantum annealer, though they have different performance characteristics and practical trade-offs, as we discuss in the manuscript. We have added text to clarify this “either/or” nature of the two approaches.

AUTHOR ACTION

We have revised the manuscript to clarify the distinction between our SQA algorithm and the one used in the benchmark study, and to better explain the two distinct strategies we use to enhance performance.

1. In Section III, paragraph 4, we have added a passage to explain that our work deliberately uses discrete-time SQA (DT-SQA) for its direct hardware-implementation advantages, in contrast to the continuous-time SQA (CT-SQA) used as a baseline in Ref. [32]. The revised text is:

“Although both our work and Ref. [32] employ SQA, our approach and goals are different. Ref. [32] uses the continuous-time variant (CT-SQA) as a theoretical baseline, whereas we deliberately use the discrete-time SQA (DT-SQA) with R explicit Trotter replicas. We chose DT-SQA as it maps naturally onto parallel hardware architectures, making it a more relevant algorithm for assessing the performance of physically realizable classical systems. Therefore, we benchmark our hardware-amenable algorithm directly against the quantum annealer’s performance, rather than reproducing the CT-SQA baseline.”

2. In Section III, paragraph 6, we have added text to explicitly frame the use of many replicas in one run ($P = 1$) versus multiple independent runs ($P > 1$) as two alternative strategies rooted in extreme value theory. The revised sentence is:

“The modified EVT approach explains how increasing replicas within a single run improves the scaling. An alternative strategy, also based on EVT, is to leverage multiple independent runs: use a fixed number of interconnected Trotter replicas, run the algorithm P times independently and then select the best energy from all runs.”

(4) Although using an off-the-shelf FPGA is straightforward, a custom ASIC—despite its potential efficiency—seems constrained in this study by die-size limits and off-chip communication overhead. Could the authors comment more on how these factors restrict its practical use?

AUTHOR RESPONSE

Our physical design study (Sec. VI) was performed specifically to address these practical concerns for custom ASICs.

We find that for sparse problems with a fixed topology, like the 3D spin glasses studied here, a monolithic ASIC is a compelling solution. Our place-and-route analysis projects that a single 7nm chip is large enough to house all ≈ 2850 replicas required for our largest DT-SQA simulations, showing that die size should not be a prohibitive constraint for these problem scales.

Crucially, this monolithic approach solves the off-chip communication overhead. As the reviewer correctly notes our current experiments on a commercial FPGA are I/O-limited, however, a custom ASIC would enable all inter-replica operations (swaps, cluster moves) to occur on-chip. This ensures that performance is governed by the intrinsic computation time, not communication latency. We have revised the manuscript to clarify this.

AUTHOR ACTION

We have revised the opening paragraph of Section VI as:

“We now evaluate the feasibility of a custom ASIC designed for replica-based algorithms on sparse, structured problems. A monolithic chip that can house all replicas on-die would eliminate the off-chip communication overhead that constrains current FPGA implementations. To make realistic projections for a full-scale Application-Specific Integrated Circuit (ASIC), our analysis is grounded in a rigorous, bottom-up physical design flow using a 7 nm process. Our findings on chip capacity for the DT-SQA algorithm apply equally to the better performing APT algorithm, which requires significantly fewer replicas, as shown in Section IV.”

(5) A classical solver could in principle be implemented on dedicated hardware such as an ASIC. However, programmability and flexibility—including the constraints of fixed routing—could then become significant issues. Could the authors discuss the trade-off between chip-level flexibility and performance (speed or energy efficiency) in this context?

AUTHOR RESPONSE

As the reviewer notes, while the FPGA is maximally reconfigurable, ideal for rapid algorithm exploration and instance-dependent topologies, a custom ASIC has fixed routing.

The 3D spin glass problems considered in this work have a regular cubic nearest-neighbor topology, which is identical for every problem instance. The only change is the specific ± 1 couplings between neighbors across instances. Consequently, a single hard-wired ASIC fabric can house any instance of a given lattice size by reloading different weights for the different instances, preserving programmability while maintaining the efficiency.

For more general problems, even if they are sparse, the programmability challenges of ASICs for Ising machines and probabilistic computers remain an open question. Several ideas, including designing master-graph approaches with fixed but multiplexed connectivity, or changing problem formulation at the algorithmic level have been proposed without a clear solution yet. We touched on these ideas in the revised manuscript.

AUTHOR ACTION

We have added a new paragraph to the end of Section V to explicitly discuss the trade-offs between reconfigurable (FPGA) and application-specific (ASIC) hardware. The added paragraph in the manuscript is:

“There is a trade-off between reconfigurable and application-specific hardware. FPGAs offer full reconfigurability, ideal for algorithmic exploration across diverse problem structures, but at a significant performance and energy cost. For the 3D spin glasses studied here, the fixed nearest-neighbor topology is an excellent match for a custom ASIC. Connectivity can be hard-wired while programmability for different instances is retained by reloading different weights. For problems with arbitrary sparse topologies, however, achieving reconfigurability on static ASICs is an open problem and may require different approaches, such as higher-order problem formulations or master graph approaches [19, 57].”

(6) The figures inconsistently interchange the notations R , $R \times P$, and R_T .

AUTHOR RESPONSE

We appreciate the careful and comprehensive reading. In Fig. 2 of the main text, we only show result from the vanilla DT-SQA algorithm which comes from a single-run of the algorithm ($P = 1$ which implies $R_T = RP = R$), which is why we do not introduce any new symbol for the simplicity of the discussion. However, when the difference between two EVT approaches where $P \neq 1$ is compared with $P = 1$ is discussed, it becomes necessary to distinguish between the total number of replicas used in the two different approaches. That is why in Supplementary Fig. S5, we introduce a new symbol $R_T = 32P$. We also explicitly mention ($P = x, R = y$) whenever two different values of P are compared.

AUTHOR ACTION

We have fixed one typo in Supplementary Fig. S5 where $R = 1600$ has been replaced by $R_T = 1600$. We have also added a line at the end of this caption which guides the reader how to compare the total number of replicas of Supplementary Fig. S5 with main Fig. 2.

“The total number of replicas R_T should be compared with the total number of replicas R in Fig. 2 in main text. Fig. 2 uses only $P = 1$ which implies $R_T = R$, for simplicity we did not use R_T there.”

II. REVIEWER #2

The manuscript “Pushing the Boundary of Quantum Advantage in Hard Combinatorial Optimization with Probabilistic Computers” revisits claims of scaling advantage by quantum annealers (QAs) on hard combinatorial optimization benchmarks (specifically 3D Edwards–Anderson spin glasses) and asks whether probabilistic computers (“p-computers”) running advanced Monte Carlo algorithms can match or exceed QA performance on the same problems. The authors focus on two replica-based algorithms such as Discrete-Time Simulated Quantum Annealing (DT-SQA) and Adaptive Parallel Tempering (APT) with Isoenergetic Cluster Moves (ICM) and co-design them with dedicated p-bits hardware architectures (CPUs, FPGAs, and projected SMTJ chips).

This is an important paper because it raises the bar for claims of “practical quantum advantage” in optimization by showing that classical p-computers, when co-designed with replica-based algorithms and hardware architectures, can match or exceed current QA performance on benchmark spin glasses. The authors emphasize that observing quantum-critical dynamics is neither necessary nor sufficient for superior real-world optimization: algorithmic innovations (e.g., ICM) and hardware parallelism can deliver comparable or better speedups. They project that further nanodevice integration (SMTJs) and ASICs will extend these advantages to larger problems. So I am in favour of publishing this paper. My comments are mostly to make the story more coherent.

AUTHOR RESPONSE

We are grateful to the reviewer for the positive evaluation and a comprehensive and careful reading of the manuscript. All comments have been addressed in the revised manuscript, and a color-tracked `diff` file highlighting every change is appended for the reviewer’s convenience.

(1) Eq. (1) could be read as summing over all pairs (i, j) . Since the paper’s benchmarks are 3D spin glasses with only nearest-neighbor (NN) couplings, the authors should explicitly state that the J_{ij} are nonzero only for NN pairs on the lattice.

AUTHOR RESPONSE/ACTION

We agree with the reviewer and we modified the line after Eq. (1) which defines J_{ij} :

... The coupling weights J_{ij} are non-zero exclusively for nearest-neighbor pairs and, for those pairs, each J_{ij} is randomly selected from $\{-1, +1\}$ with equal probability...

(2) it isn’t clear from the current text what κ_f represents when you first see Eq. (3). Perhaps, say “where κ_f is the fitted scaling exponent of the residual-energy decay.” Or something along these lines.

AUTHOR RESPONSE/ACTION

We agree. To clarify, we have added an explicit definition following Eq. (3), stating that κ_f is the fitted scaling exponent describing the power-law decay of the residual energy.

“... where κ_f is the fitted scaling exponent describing the power-law decay of the residual energy.”

(3) The top paragraph on the right panel of page 3 relies heavily on specialized terminology without defining key concepts, which may confuse readers. E.g. It refers to “the density of defects” but never explains what a defect is in this context (e.g., domain walls, kinks, spin flips). A brief definition is needed. The term “correlation length” is

never clarified. The authors should state how it's defined and measured (for instance, the characteristic distance over which spins are correlated). "Binder cumulant" is mentioned as scaling with KZM predictions but not defined. A short explanation (e.g., a dimensionless ratio used to quantify critical fluctuations) or a reference would help. The paragraph contrasts the residual energy exponent with the KZM prediction but doesn't remind the reader of its definition or where it appears (e.g., Eq. (3)). A cross-reference or brief recap would help.

AUTHOR RESPONSE/ACTION

Our intention in the original paragraph was to clarify that practical optimization algorithms typically operate far from critical points, rendering KZM dynamics less relevant to our analysis. However, we agree with the reviewer that our previous phrasing was unclear and included specialized terminology ("defect density," "correlation length," "Binder cumulant") without adequate definitions. To address this, we have now completely rewritten the paragraph to explicitly emphasize that our study focuses primarily on practical optimization regimes, where critical dynamics such as those described by KZM play no direct role. As an additional benefit, this revision naturally eliminated the need for these specialized terms. All changes can be tracked clearly in the accompanying `diff` file.

(4) "using logical instances of size $15 \times 15 \times 12$ " is potentially opaque. Perhaps it is better to say, "We generate logical problem instances on a $15 \times 15 \times 12$ 3D lattice of Ising spins."

AUTHOR RESPONSE/ACTION

We would like to clarify that we did not generate these logical problem instances ourselves; rather, we used the problem instances directly obtained from Ref. [32]. To enhance clarity, we have modified the sentence accordingly:

"...and the simulations were performed on CPUs using logical problem instances defined on a 3D cubic lattice of Ising spins with dimensions $15 \times 15 \times 12$, obtained directly from Ref. [32]..."

(5) Define "broken dimers" and "embedded instances".

AUTHOR RESPONSE/ACTION

We agree and to improve readability, we have explicitly defined both terms in the revised manuscript:

"...based on the observation that broken dimers (when physical spins representing the same logical spin do not agree after annealing) are rare ..."

and

"The slopes quoted above are based on embedded instances (logical problem instances mapped onto the quantum annealer's physical qubit connectivity graph). Quantum annealers (QAs) typically require complex embedding schemes for combinatorial optimization problems, in which a single logical spin is represented by multiple physical spins grouped into structures called dimers, due to their fixed hardware topology (such as the Chimera or Pegasus graphs), even for relatively sparse problems like 3D spin glasses..."

(6) "Artificially include embedding overheads": do the authors mean "we omit the additional time and resource costs needed to embed problems on QA hardware"?

AUTHOR RESPONSE

To clarify our intent: our probabilistic solvers incur no embedding overhead because they directly represent the logical optimization problem. By contrast, quantum annealers inherently require embedding overhead due to their rigid hardware topologies, such as the Chimera or Pegasus graphs, even for relatively sparse problems like 3D spin glasses. Given our motivation in this manuscript is the practical solution of hard and *sparse* optimization problems

(see our response to reviewer’s comment 13), we do not artificially emulate the embedding overhead of quantum annealers in our results. This point has been clarified explicitly in the revised manuscript, as described below.

AUTHOR ACTION

The paragraph is revised to:

“The slopes quoted above are based on embedded instances (logical problem instances mapped onto the quantum annealer’s physical qubit connectivity graph). Quantum annealers (QAs) typically require complex embedding schemes for combinatorial optimization problems, in which a single logical spin is represented by multiple physical spins grouped into structures called dimers, due to their fixed hardware topology (such as the Chimera or Pegasus graphs), even for relatively sparse problems like 3D spin glasses. By contrast, probabilistic computers implemented on flexible classical hardware, such as FPGAs or ASICs, can directly represent and solve the logical problem graph without embedding overhead. Since our goal is to evaluate the intrinsic performance of algorithms solving practical combinatorial optimization problems, our main results do not include embedding overheads that are specific to current quantum annealer architectures. . . .”

(7) “There is no such concern . . . however.” The “however” feels detached and the double negative (“no such concern . . . since quantum emulation is not a target”) makes the logic a bit hard to follow on first read. Also, the authors begin by talking about Monte Carlo simulations “whose goal is to faithfully simulate . . .,” then switch to “however” about optimization problems, and then immediately refer to “our replicas” and “physical spins.” It might help to split these ideas.

AUTHOR RESPONSE/ACTION

We agree, we rewrote this paragraph to:

“In Monte Carlo simulations intended to accurately emulate equilibrium quantum physics, selecting the best-performing replica among multiple Trotter replicas is usually avoided, as this could bias equilibrium observables [41, 42]. However, such concerns are not relevant in our context, because our goal is not quantum emulation but rather practical combinatorial optimization. Indeed, since our replicas represent independent physical entities realized by separate physical spins in hardware, identifying and selecting the replica with the lowest residual energy is both natural and appropriate.”

(8) The authors say DT-SQA “transitions into a plateau” : do they mean its residual-energy decay stagnates after some time? Be explicit.

(9) What does $L = 15$ mean? The lattice size was quoted as $15 \times 15 \times 12$. May be “we study 3D lattices whose two longest sides have length $L=15$.” Or “We consider 3D spin glasses on lattices of size $L_x \times L_y \times L_z$, with $L_x=L_y=15$, and $L_z=12$.”

(11) “Slope” vs. “plateau”. Do the authors mean DT-SQA’s residual-energy decay exponent is larger at first (steeper decline) but then levels off (little further improvement) at long runtimes?

AUTHOR RESPONSE/ACTION

We have combined these three comments as they are related to the same paragraph. We have revised the relevant sentence to explicitly state that the “slope” refers to the rate of residual-energy decay and the “plateau” describes the stagnation of this decay at long annealing times. The new phrasing clarifies that DT-SQA exhibits an initial steep decline in residual energy that eventually levels off. Additionally, we have also replaced $L = 15$ with the exact size.

“As discussed in Section IV, DT-SQA, despite exhibiting a steep power-law decay in residual energy at early times in instances defined on a fixed 3D spin glass lattice of size $15 \times 15 \times 12$, transitions into a plateau at longer annealing times, where the residual energy stagnates and shows little further improvement. It is ultimately outperformed by the APT algorithm, which is easier to implement and parallelize in hardware.”

(10) Not every reader will recall what stands for “Monte Carlo sweeps.”

AUTHOR RESPONSE/ACTION

In our revised manuscript, we have made the definition of MCS explicit (see the caption of Fig. 2 for example)

“...in Monte Carlo sweeps, (MCS) which involve attempting to update all the spins in the system once ...”

(12) The phrase “absolute residual errors” is potentially confusing for two reasons: (1) is there a single quantity (the residual energy per instance)? In this case “errors” (plural) feels odd. (2) It’s unclear whether the authors mean the absolute value of a signed residual or simply the raw, non-normalized residual energy. Also, ‘Residual energy vs. Residual error. I do not understand this: residual energy = $\epsilon_{\text{res}} = E(t) - E_{\text{min}}$ is always nonnegative, so the adjective “absolute” seems redundant.

AUTHOR RESPONSE/ACTION

We thank the reviewer for pointing out the ambiguity. We agree that the phrase “absolute residual errors” is misleading and not standard. As the reviewer correctly noted, residual energy is already a nonnegative quantity and, as commonly defined in the spin-glass literature, refers to an average over multiple instances. Our intended meaning was simply that one method achieves lower residual energy. We have revised the text to use the clearer and more appropriate phrasing “lower residual energy”.

“...However, APT achieves lower residual energy for a given MCS budget. ...”

*(13) The community often benchmarks QA and DT-SQA on the standard 3D tile graphs that have a controllable hardness [Hamze, Firas, et al. “From near to eternity: spin-glass planting, tiling puzzles, and constraint-satisfaction problems.” *Physical Review E* 97.4 (2018): 043303]. Could the authors comment on their choice of Edwards–Anderson spin glass instead? Specifically, how do their conclusions extend to or differ on the tiles3D benchmark?*

AUTHOR RESPONSE

Our primary motivation in selecting the Edwards–Anderson spin-glass model was the extensive availability of benchmark data from quantum annealers for this problem, as well as previously claimed scaling advantages for quantum annealing. However, our conclusions are not limited to this particular model. Indeed, as the reviewer points out, the community often employs other sparse benchmarks, such as the 3D tiles problem [54]. Our architectural and algorithmic results (Sec. V, Fig. 4) indicate that the demonstrated advantages of probabilistic computers apply generally to sparse optimization problems, including tiles3D or any other graph topology with sufficiently low connectivity (e.g., ≤ 50 nodes per spin). Because the probabilistic architecture with graph coloring naturally exploit parallel updates on sparse topologies, updating spins in constant $O(1)$ time rather than the $O(N)$ serial updates typical of CPU implementations, they achieve a substantial $O(N)$ overall scaling advantage per sweep. Thus, we expect our conclusions regarding p-computer performance to remain robust for the tiles3D benchmark and other sparse optimization problems.

AUTHOR ACTION

We have revised the manuscript (Sec. V) to explicitly note that our architectural conclusions broadly apply to sparse optimization problems beyond the Edwards–Anderson spin glass, including common benchmarks such as the tiles3D problem [54].

“In sparse problems, such as 3D spin-glasses, planted Ising benchmarks [30, 54] or circuit SAT problems with sparse connectivity [4], p-computer architectures leverage physically parallel nodes to simultaneously update large independent sets [4]”

(14) Algorithm S1 is based on the colouring problem but the paper never explicitly explains that “colours” correspond to graph-colouring classes used to identify independent spin sets for parallel updates. It simply presents the loop over colours without motivation.

AUTHOR RESPONSE/ACTION

We thank the reviewer for pointing out the missing connection between our graph-coloring explanation and Algorithm S1. While we had described the role of graph coloring in the Methods section, we have now added a clarifying sentence before referring to Algorithm S1, explicitly noting that “colors” correspond to independent spin sets obtained via graph coloring for parallel updates.

“... As described in the Methods section, graph coloring is used to partition the spin system into independent sets that can be updated in parallel. Each color corresponds to one of these independent sets. Algorithm S1 implements this update scheme by looping over colors.”

(15) Caption for Fig 4: what is “prefactor improvement”? is it a ratio of CPU to sMTJ-p-computer sweep times?

AUTHOR RESPONSE/ACTION

Our original description was not clear. In Fig. 4(c) “prefactor improvement” of the sMTJ p-computer refers to the measured FPGA sweep time, not to the CPU baseline, since the FPGA already realizes the $\mathcal{O}(n)$ speed-up from parallel updating of all spins. The orange band simply shows the *additional* constant speed-up expected when the same architecture is implemented with fast, on-chip sMTJ p-bits (approximately 1 ns intrinsic flip time). This factor shifts the timing curve vertically but does not change its scaling with system size.

In our revised manuscript, we have clarified this in the caption of Fig. 4.

“... FPGA maintains $\mathcal{O}(n)$ improvement over CPU, while ‘prefactor improvement’ for sMTJ-based p-computers refers to the additional constant speed-up expected when the same architecture is implemented with fast, on-chip sMTJ p-bits (approximately 1 ns intrinsic flip time), relative to the measured FPGA sweep time. sMTJ projections assume a single chip with 1 million sMTJs, achieving 1 million flips per nanosecond (the thick orange line assumes 10 and 50 replicas for the upper and lower bounds, respectively). This also includes improvements stemming from additional parallelization, meaning that if the problem size is smaller than the chip’s capacity, multiple independent runs or problem instances can be processed simultaneously on the same chip, scaling as $10^6 / (\text{spins per replica} \times \text{APT replicas})$.”

(16) Fig S3: Define “95% bootstrap confidence interval” (e.g. over 300 instances).?

AUTHOR RESPONSE/ACTION

We have added this information in the caption of Supplementary Fig. S3 of our revised manuscript.

(17) Fig. S15: Define “Houdayer move”. Add, perhaps, “A Houdayer move [33] is an isoenergetic cluster swap in which one identifies a cluster of antiparallel spins between two replicas and exchanges them to enhance mixing in spin-glass Monte Carlo.”

AUTHOR RESPONSE/ACTION

We have now added a brief definition of the Houdayer move in the caption of Fig. S15, clarifying its role in improving mixing via isoenergetic cluster swaps.

“A Houdayer move [36] is an isoenergetic cluster update used in spin-glass Monte Carlo simulations. It operates by identifying a cluster of antiparallel spins between two replicas and swapping them to enhance mixing and improve sampling efficiency.”

(18) Time-to-solution is introduced in the Introduction, but the precise definition (e.g. time to reach a target residual energy with 99% probability) is only buried in the Methods.

AUTHOR ACTION

We agree. We have revised the final paragraph of the introduction section to briefly introduce the concepts of residual energy and time-to-solution. We also want to clarify that we are not using the formal definition of "time-to-solution" (e.g., time to 99% success probability for finding the ground state). Instead, our metric is the average time to reach a specific, non-zero target residual energy, which serves as a practical benchmark for optimization progress.

The revised text now reads:

"Finite-size scaling analysis reveals a collapse of residual energy curves, our primary metric of solution quality, for APT with the steeper slope emerging as a universal feature that delivers superior performance in large-scale optimization problems. This indicates superior performance in large-scale optimization problems, where minimizing the time-to-solution for a target residual energy is the key objective."

(19) Supplementary Fig. S2 caption: "15×15×12×2" If it is a chain length for embedding, it should be described as such.

AUTHOR RESPONSE/ACTION

We agree and have added a clarification inside a parenthesis in the caption to explicitly describe the meaning of the embedding dimension:

"...for embedded instances of size $15 \times 15 \times 12 \times 2$ (the latter 2 represents the number of physical qubits used to represent a logical spin),..."

(20) Which FPGA board was used, what precision was assumed, or how the logic was synthesized?

AUTHOR RESPONSE/ACTION

Details of the FPGA board, arithmetic precision, and synthesis methodology are provided in Supplementary Sections III and V. To improve clarity, we have now added a separate subsection "FPGA details" under the Methods section of the main text containing a brief summary in our revised manuscript.

"We mapped the physics-inspired, massively parallel p-computer architecture of Ref. [4] onto a Xilinx Alveo U250 data-center accelerator card using graph coloring to maximize parallelism on the sparse instances. All arithmetic is fixed-point: DT-SQA uses s{6}{3} precision (1 sign, 6 integer, 3 fractional bits) while APT+ICM uses the higher s{6}{6} precision. Custom RTLs were developed to implement the algorithm based on the p-computing architecture and synthesized, placed and routed with Xilinx Vivado/Vitis tool chain. Further details are provided in Supplementary Sections III (DT-SQA) and V (APT)."

(21) What is an ASIC projections?

AUTHOR RESPONSE

In our work, an "ASIC projection" is an estimate for a large-scale, custom Application-Specific Integrated Circuit. These projections are obtained from a rigorous physical design methodology detailed in Section VI and Supplementary Section III. Specifically, we performed a full place-and-route synthesis for smaller-scale versions of our p-computer using an industry-standard 7 nm process design kit. This process provided concrete data on chip area and resource scaling (as shown in Supplementary Fig. S10 and Table S1). By validating the linear scaling of area with the number of p-bits (Supplementary Fig. S11), we could then reliably project the requirements for a full-scale chip. To improve clarity, we have revised the text to better introduce this methodology.

AUTHOR ACTION

We have revised the beginning of the Section VI to more clearly explain the basis of our projections.

(22) sweep-to-swap ratio is introduced in Supplementary Figs. S14–S16 captions without any definition. What exactly is being counted?

AUTHOR RESPONSE/ACTION

The “sweep-to-swap ratio” refers to the number of Monte Carlo sweeps performed for each replica before attempting a swap operation between replicas. In our revised manuscript, we have explicitly included this definition in the captions of Supplementary Figures S14–S16 to clarify precisely what is being counted.

(23) Use either n or N for the number of spins consistently. (e.g. Fig. S6 caption uses N , but Eq.(2) uses n .)

AUTHOR RESPONSE/ACTION

We have replaced all instances of ‘ N ’ with ‘ n ’.

III. REVIEWER #3

Review of the manuscript entitled “Pushing the Boundary of Quantum Advantage in Hard Combinatorial Optimization with Probabilistic Computers” authored by Shuvro Chowdhury, Navid Anjum Aadit, Andrea Grimaldi, Eleonora Raimondo, Atharva Raut, P. Aaron Lott, Johan H. Mentink, Marek M. Rams, Federico Ricci-Tersenghi, Massimo Chiappini, Luke S. Theogarajan, Tathagata Srimani, Giovanni Finocchio, Masoud Mohseni and Kerem Y. Camsari. (Manuscript #: NCOMMS-25-33449-T)

This manuscript investigates the performance of probabilistic computers on hard combinatorial optimization problems, comparing the DT-SQA and APT algorithms with state-of-the-art quantum annealers. The authors argue that, when implemented on massively parallel architectures such as FPGA or projected sMTJ chips, these classical algorithms can match or even exceed the residual energy scaling observed in quantum annealing. While the approach is promising and includes novel architectural insights and new analytical perspectives, further clarification is needed regarding the originality relative to prior work, the justification for using residual energy as a cross- platform benchmark, and the scalability of the methods to larger problem sizes. I believe the manuscript can be significantly strengthened by addressing the following major points.

AUTHOR RESPONSE

We sincerely thank the reviewer for the careful reading of our manuscript and the constructive feedback provided. All comments have been addressed in the revised version, and a color-coded `diff` file showing every change is appended at the end of this document for the reviewer’s convenience.

Major Comments:

(1) Clarification of novelty relative to prior work (Lee et al., Communications Physics, 2025)

While this manuscript presents a valuable comparison by incorporating QMC and quantum annealing (QA) scaling behaviors, its conceptual overlap with the authors’ prior work requires further clarification. In particular, the scaling behavior of residual energy and the interpretation of deviations from Kibble–Zurek mechanism (KZM) predictions—framed through extreme value theory (EVT), coarsening dynamics, and the use of asynchronous probabilistic samplers—appear closely related to the results presented in Figure 4 of the earlier publication. Although new terminology (e.g., isoenergetic cluster moves) and theoretical framing are introduced, the overall narrative feels incremental. A more explicit and detailed discussion of what constitutes new physical insights, experimental regimes, or theoretical contributions in this manuscript would be needed to warrant the publication of this work.

AUTHOR RESPONSE

Let us emphasize unequivocally that apart from the superficial resemblance of employing the 3D spin glass as a benchmark, the scientific motivation, technical scope, and conceptual thrust of the present manuscript are fundamentally distinct from our previous publication (Lee et al., Communications Physics 2025).

Specifically, the Lee et al. paper is entirely focused on examining whether fully deterministic, “chaotic” bits (c-bits), operating without explicit random number generators, can emulate the scaling behavior and stochastic dynamics exhibited by conventional probabilistic bits (p-bits) that inherently rely on randomness. This earlier work restricted itself exclusively to algorithms like Simulated Annealing and explicitly analyzed the problem within the critical region described by the Kibble–Zurek mechanism (KZM). The key result was the demonstration that deterministic dynamics alone could faithfully reproduce stochastic scaling behaviors, opening the possibility of eliminating resource-intensive random number generators in hardware implementations. While these are interesting and worthwhile findings in

their own right, they are fundamentally unrelated to the conceptual framework or scientific objectives of our current manuscript.

In sharp contrast, our present manuscript sets out to investigate an entirely different question, operating in a vastly different algorithmic and experimental regime: we ask whether probabilistic bits, now explicitly harnessing randomness, can match or surpass the performance of coherent quantum annealers via discrete-time Simulated Quantum Annealing (DT-SQA) and Adaptive Parallel Tempering (APT) supplemented by isoenergetic cluster moves. Crucially, all algorithms in our work operate *far from* critical regions relevant to KZM physics, since practical annealing protocols typically explore dynamics well above criticality. We employ extreme-value theory (EVT) concepts and coarsening dynamics to interpret residual-energy scaling in a fundamentally different theoretical setting than the one analyzed previously.

Moreover, there is a profound difference in the problem scales between the two manuscripts. Lee et al. employed modest spin-glass lattices, up to $L^3 = 11^3 = 1331$ spins, sufficient only to illustrate equivalence between deterministic and stochastic classical dynamics. In our present manuscript, we have systematically solved spin-glass lattices reaching unprecedented sizes ($L^3 = 16^3 = 4096$ spins with $R = 2850$ Trotter replicas, corresponding to effective system sizes exceeding 11 million spins!). Importantly, these massive computations were not only demonstrated theoretically but also explicitly shown to be feasible in state-of-the-art probabilistic computing hardware, such as Application Specific Integrated Circuits (ASIC) and emerging spintronic technologies, establishing their practical relevance.

We emphasize that our results explicitly demonstrate the significant algorithmic advantage provided by the introduction of isoenergetic cluster moves (ICM) within the Adaptive Parallel Tempering algorithm, as illustrated in Supplementary Figures 18 and 19. Without ICM, the dynamic exponent, μ , increases substantially from 6.34 (with ICM) to 6.86 (without ICM), quantitatively underscoring the scaling benefit derived from these isoenergetic moves.

In short, apart from employing a common canonical benchmark, the two manuscripts explore entirely separate physical questions, algorithmic strategies, theoretical frameworks, and technological implications. We have revised the manuscript accordingly to emphasize this distinction more clearly and avoid any further confusion.

AUTHOR ACTION

Please see the added sentence in the introduction where we cite a few papers that use 3D spin glasses as benchmarks to compare different algorithms, including the reference mentioned by the referee:

“Due to their hardness, 3D spin glasses have long served as canonical benchmarks for evaluating scaling behavior of various algorithms [33-35].”

(2) Support for claims about probabilistic computing and its advantages

The manuscript makes strong claims about the competitiveness or even superiority of probabilistic computing hardware relative to QMC and QA approaches. While the reviewer agrees that such architectures may offer practical advantages in specific applications, the argument would benefit from further support. Specifically, the transition in Section 2—from a formal scaling framework (via KZM) to using residual energy as a single empirical benchmark—should be more clearly justified. For example, explaining how residual energy captures relevant non-equilibrium dynamics beyond what KZM predicts, and under what assumptions it remains a valid comparison metric across such heterogeneous platforms, would strengthen the theoretical foundation for this claim.

AUTHOR RESPONSE

We agree that the earlier discussion in Section II may have inadvertently suggested an overly strong connection to the Kibble–Zurek mechanism (KZM). In fact, as we clarified in our response to the previous comment, the annealing algorithms explored in this manuscript explicitly operate far away from critical regions relevant to KZM physics. Consequently, we do not rely on KZM predictions, nor do we use them to justify our comparisons. To prevent confusion, we have now completely rewritten the paragraph in Section 2, clearly emphasizing that residual energy serves solely as a practical empirical benchmark appropriate for quantifying performance in non-critical regimes relevant to optimization.

AUTHOR ACTION

Please see the revised paragraph in Section 2:

“While the performance scaling in Ref. [32] is analyzed in the context of the Kibble-Zurek mechanism (KZM), it is noted there that the residual energy does not follow a simple prediction from critical dynamics alone, as also noted in Ref. [31]. In our work, we are therefore primarily focused on the quality of solutions, using the residual energy scaling (Eq. (3)) simply as an empirical benchmark for different optimizers. We do not attempt to map our data onto specific KZM exponents, nor do we assume that near-critical power laws fully govern the eventual solution quality for these optimization problems.”

(3) *Generalization of scaling behavior to larger problem size in APT + ICM algorithm.*

While the authors demonstrate compelling scaling collapse across $L = 8$ to 16 using the APT + ICM algorithm, it remains unclear whether this performance persists for larger problem sizes beyond the tested regime. Given that real-world combinatorial optimization applications often involve far larger systems, it would be valuable to discuss how the proposed methods scale in practice beyond $L = 16$. Could the authors provide additional justification—either through extrapolation, simulation evidence, or architectural considerations—that the observed scaling behavior is robust for larger instances?

AUTHOR RESPONSE

The universal collapse shown in Fig. 3(b) and Supplementary Fig. S18 provides compelling simulation evidence that the observed scaling behavior, characterized by a dynamic exponent $\mu \approx 6.34$ and scaling exponent $b \approx 3.0$, holds across all studied system sizes. Standard finite-size scaling theory implies this functional form remains valid beyond the explicitly tested range ($L = 8$ to 16), **at all sizes**. Consequently, as long as the APT+ICM protocol remains unchanged, our scaling law can reliably predict the time-to-solution for a target residual energy at substantially larger problem sizes.

AUTHOR ACTION

We have added a sentence at the end of Section IV highlighting this extrapolation.

“... On the other hand, the robust universal collapse shown in Fig. 3(b) allows us to extrapolate the time required to reach a target residual energy for cubes of any size.”

Minor Comments:

(1) *Awkward phrasing in the main text*

‘We would like to emphasize that as hardware accelerators the existence of classically hard to simulate quantum dynamics during an annealing schedule is neither necessary nor sufficient for observing speedup over certain state-of-the-art solvers for combinatorial optimization problems. → This sentence does not make a good sense.’

AUTHOR RESPONSE/ACTION

We agree with the reviewer and have removed this sentence from the Outlook. The revised text now focuses on the rationale for probabilistic algorithms without drawing a direct comparison to quantum annealers.

(2) *Some typos in the manuscript*

(i) On page 4, ‘(Supplementary Fig. S7(b))’ → ‘(Supplementary Fig. S7(b))’

(ii) In Fig. S9 on page 21, ‘Interconnect’ → ‘Interconnect’

(iii) In Fig. S17 on page 26, ‘The embedded instances show similar characteristics to those of logical instanes.’ → ‘The embedded instances show similar characteristics to those of logical instances.’

AUTHOR RESPONSE/ACTION

We thank the reviewer for identifying these typos. We fixed all of them in the revised manuscript.

Pushing the Boundary of Quantum Advantage in Hard Combinatorial Optimization with Probabilistic Computers

Shuvro Chowdhury,¹ Navid Anjum Aadit,¹ Andrea Grimaldi,^{2,3} Eleonora Raimondo,^{2,4} Atharva Raut,⁵ P. Aaron Lott,^{6,7} Johan H. Mentink,⁸ Marek M. Rams,⁹ Federico Ricci-Tersenghi,¹⁰ Massimo Chiappini,⁴ Luke S. Theogarajan,¹ Tathagata Srimani,⁵ Giovanni Finocchio,² Masoud Mohseni,¹¹ and Kerem Y. Camsari¹

¹*Department of Electrical and Computer Engineering,
University of California, Santa Barbara, Santa Barbara, CA 93106, USA*

²*Department of Mathematical and Computer Sciences,
Physical Sciences and Earth Sciences, University of Messina, 98166, Messina, Italy*

³*Department of Electrical and Information Engineering, Politecnico di Bari, 70126 Bari, Italy*

⁴*Istituto Nazionale di Geofisica e Vulcanologia, Via di Vigna Murata 605, 00143 Roma, Italy*

⁵*Department of Electrical and Computer Engineering, Carnegie Mellon University*

⁶*USRA Research Institute for Advanced Computer Science (RIACS)*

⁷*Quantum Artificial Intelligence Laboratory (QuAIL), NASA Ames Research Center*

⁸*Institute for Molecules and Materials, Radboud University, Heyendaalseweg 135, Nijmegen, The Netherlands*

⁹*Institute of Theoretical Physics, Jagiellonian University, Lojasiewicza 11, PL-30348 Kraków, Poland*

¹⁰*Dipartimento di Fisica, Sapienza Università di Roma, and CNR-Nanotec,*

Rome unit and INFN, Sezione di Roma 1, 00185 Rome, Italy

¹¹*Emergent Machine Intelligence, Hewlett Packard Labs, CA, USA*

(Dated: July 28, 2025)

Recent demonstrations on specialized benchmarks have reignited excitement for quantum computers, yet whether they can deliver an advantage for practical real-world problems remains an open question. Here, we show that probabilistic computers (p-computers), when co-designed with hardware to implement powerful Monte Carlo algorithms ~~can surpass state-of-the-art quantum annealers King et al., Nature (2023) in solving certain~~, provide a compelling and scalable classical pathway for solving hard optimization problems. We focus on two key algorithms applied to 3D spin glasses: discrete-time simulated quantum annealing (DT-SQA) and adaptive parallel tempering (APT), ~~both applied to 3D spin glasses.~~ We benchmark these methods against the performance of a leading quantum annealer on the same problem instances. For DT-SQA, we find that increasing the number of replicas improves residual energy scaling, ~~while parallelizing fewer replicas across independent runs also achieves comparable scaling.~~ Both strategies align with the theoretical in line with expectations from extreme value theory. ~~In addition, APT outperforms DT-SQA. We then show that APT, when supported by non-local isoenergetic cluster moves.~~ Finite-size scaling analysis suggests a universal behavior that explains the superior performance of APT over both DT-SQA and quantum annealing. ~~We show that,~~ exhibits a more favorable scaling and ultimately outperforms DT-SQA. We demonstrate these algorithms are readily implementable in modern hardware ~~thanks to the mature semiconductor technology.~~ Unlike software simulations, replicas can be monolithically housed on a single chip and a large number of spins can be updated in parallel and asynchronously, similar to a quantum annealer. ~~We project,~~ projecting that custom Field Programmable Gate Arrays (FPGA) or specialized chips leveraging massive parallelism can further can leverage massive parallelism to accelerate these algorithms by orders of magnitude ~~while drastically improving energy efficiency.~~ Our results raise the bar for establish a new, rigorous classical baseline, clarifying the landscape for assessing a practical quantum advantage ~~in optimization and present and presenting~~ p-computers as scalable, energy-efficient hardware a scalable platform for real-world optimization ~~problems~~ challenges.

I. Introduction

Richard Feynman is widely credited with starting the field of quantum computing in a 1982 lecture [1]. In the same lecture, Feynman also introduced the notion of a probabilistic computer, one that naturally simulates probabilistic processes. Feynman’s broader vision of *physical computers*, or programmable physical devices that solve a problem of interest through their natural evolution, has recently inspired a growing array of physical and physics-inspired classical computing paradigms, including systems to train deep neural networks [2], solve linear algebra problems [3], and tackle combinatorial optimization problems [4].

Building on this vision, a key challenge is identifying scenarios where scalable and error-corrected quantum

computers [5] could outperform probabilistic or classical approaches, particularly in optimization and sampling tasks. Prominent examples include Shor’s algorithm for factoring large integers [6], sampling random quantum circuits [7], and learning quantum data on quantum processors [8, 9], each offering potential exponential speedups over all known probabilistic alternatives, typically due to the interference of probability amplitudes in a high-dimensional Hilbert space. However, the scaling challenges and quantum error correction overhead might diminish or eliminate such quantum advantages [5]. Notably, while probabilistic computers can emulate quantum interference with polynomial resources, their convergence is in general believed to require exponential time [10]. This challenge is known as the sign-problem in Monte Carlo algorithms [11].

Fig. 1. Schematic overview of the probabilistic algorithms and technological platforms: (a) Representation of a 3D Ising spin glass, where the weights connecting spins can be +1 (ferromagnetic, red) or -1 (antiferromagnetic, blue). Many real-world hard optimization problems can be mapped onto such spin-glass systems. (b, c) Two replica-based algorithms investigated in this work: discrete-time simulated quantum annealing (DT-SQA), in (b), and adaptive parallel tempering (APT) with isoenergetic cluster moves (ICM), in (c). In SQA, replicas are interconnected, and the strength of these interconnections is annealed according to a schedule. APT uses multiple replicas of the problem, each running at different temperatures, and periodically exchanges states between replicas to escape local minima. (d) The p-computing scheme, where a general-purpose CPU interfaces with a specialized probabilistic computer to efficiently implement Monte Carlo algorithms. Various implementations of probabilistic computers are shown, including CPU [4, 12], GPU [13–17], Field Programmable Gate Arrays (FPGA) [4, 18–22], interconnected FPGAs, and monolithic CMOS + sMTJ (stochastic magnetic tunnel junction) chips [23–26]. Each platform offers trade-offs in sampling throughput, energy efficiency, problem size, and technological complexity.

On the other hand, establishing a quantum advantage becomes much harder in cases where quantum fluctuations or quantum interference may be present, but not known to play a significant role. For example, even though quantum annealers by D-Wave operate on the transverse field Ising Hamiltonian, which does not suffer from the sign problem, empirical performance advantages have been sought to be demonstrated over the years [27–30]. In a similar attempt, Bernaschi et al. [31] clarified that for a 2D quantum spin glass, quantum annealing could still provide a speedup in entering the spin-glass phase under certain conditions. However, it is unclear whether these advantages extend to solving optimization problems and represent a fundamental improvement over classical algorithms, such as simulated quantum annealing (SQA) and adaptive parallel tempering (APT), or if they are primarily due to hardware acceleration. Speedups of this second type are also a feature of dedicated probabilistic computers when the hardware architecture is tailored for probabilistic algorithms [12].

Recently, King et al. [32] demonstrated another empirical scaling advantage over continuous-time simulated quantum annealing (CT-SQA) and simulated annealing (SA) in solving classical 3D cubic Ising spin glass problems (Fig. 1(a)). Due to their hardness, 3D spin glasses have long served as canonical benchmarks for evaluating scaling behavior of various algorithms [33–35].

The performance reported in Ref. [32] provides a timely and valuable benchmark for the field. In this work, we ~~first evaluate whether specialized probabilistic computers can outperform quantum annealers in similar optimization tasks. Our broader goal is to draw attention to how effective probabilistic computing approaches can be, especially when algorithms are use this benchmark to evaluate a powerful classical alternative: probabilistic computers~~ co-designed with domain-specific hardware. ~~We do not challenge the observation of~~ While the quantum critical dynamics reported in Ref. [32] that might be classically hard, but we show that the claim of scaling advantage for energy

optimization or implying quantum speedup for combinatorial optimization applications is not warranted according to our study here. We use two observed in that study are a significant physical finding, it is crucial to assess whether the resulting optimization performance is competitive with the most advanced classical techniques. Our study addresses this by demonstrating that p-computers, implementing state-of-the-art replica-based Monte Carlo algorithms (Fig. 1(b, c))—the—, can achieve a comparable, and in some cases more favorable, performance scaling on the same 3D spin glass problems. Specifically, we investigate discrete-time simulated quantum annealing (DT-SQA) and APT, implementing them on different realizations of p-computer adaptive parallel tempering (APT) on various p-computer realizations (Fig. 1(d)).

Our hybrid computing platform combines a general-purpose computer with a p-computer specializing in fast Monte Carlo sampling (Fig. 1(d)). p-computers have been implemented in CPUs [4, 12], GPUs [13–17], Field Programmable Gate Arrays (FPGAs) [4, 18–22], and interconnected FPGAs. Specialized accelerators using single and distributed FPGAs already provide orders of magnitude performance improvements over CPUs [4]. Although small-scale p-computers using CMOS + stochastic magnetic tunnel junction technology (sMTJ) have been developed [23], monolithically integrated CMOS + sMTJ chips hold the greatest promise in terms of energy efficiency and performance. However, the large-scale monolithic integration of CMOS + sMTJ remains to be seen.

For our experiments, we use CPU and FPGA implementations of p-computers. For scaling studies, we use CPUs when prefactors of solution times are not critical and FPGAs when they are a priority. Specifically, we use DT-SQA with a large number of physical replicas and select the best replica at the end of the annealing. Using extreme value theory, we relate scaling exponents to the number of replicas, achieving good agreement with our experiments. In addition, a powerful variant of PT, equipped with isoenergetic cluster moves (ICM) [36–38], exhibits a transition from an initial gentler slope to a steeper one due to the non-local moves, providing superior scaling to DT-SQA. Finite-size scaling analysis reveals a collapse of residual energy curves for APT, our primary metric of solution quality, for APT with the steeper slope emerging as a universal feature with that delivers superior performance in large-scale optimization problems. This indicates superior performance in large-scale optimization problems, where minimizing the time-to-solution for a target residual energy is the key objective. Projections based on open-source process design kits show that modern digital chip technology can accommodate a large number of on-chip replicas, making all of our algorithms readily manufacturable in single chips. We also analyze the prefactor and architectural improvements achievable through dedicated FPGA and ASIC implementations. The projections further extend to modern digital chips and CMOS + X-type architectures incorporating nanodevices.

II. Residual Energy of 3D Spin Glasses

The problem setting is the Edwards-Anderson spin glass on the 3D cubic lattice:

$$H = - \sum_{i < j} J_{ij} \sigma_i \sigma_j, \quad (1)$$

where σ_i are Ising spins, $\sigma_i \in \{-1, +1\}$. The coupling weights J_{ij} are non-zero exclusively for nearest-neighbor pairs and, for those pairs, each J_{ij} is randomly selected from $\{-1, +1\}$ with equal probability. One quantity of interest is the residual energy ρ_E^f defined as a function of the annealing time t_a :

$$\rho_E^f(t_a) = \frac{\langle E(t_a) - E_0 \rangle}{n}, \quad (2)$$

where E_0 is the ground energy of the Hamiltonian H , $E(t_a)$ is the energy measured at the end of the annealing time t_a and n is the number of spins in the system. The averaging is performed over different problem instances and multiple independent runs.

Experimental observations from probabilistic Monte Carlo algorithms and quantum annealers show that the residual energy scales as a power-law in t_a :

$$\rho_E^f \propto t_a^{-\kappa_f}. \quad (3)$$

where κ_f is the fitted scaling exponent describing the power-law decay of the residual energy.

While the performance scaling in Ref. [32] investigates it in terms is analyzed in the context of the Kibble-Zurek mechanism (KZM), which predicts how the density of defects scales with the annealing time following a quench across a continuous critical point. In the coherent quantum annealer, quantities such as the Binder cumulant scale consistently with the KZM prediction with the power-law exponent given as a combination of critical exponents characterizing the universality class of the quantum critical point. However, the residual energy exponent does not align with a simple KZM prediction. This shows that the scaling of the residual energy gets additionally affected by the coarsening dynamics when the system excited during critical point crossing is further quenched across the spin-glass phase to the zero-field where it is finally measured. There is no simple theoretical formula linking the KZM correlation length directly to the final residual energy it is noted there that the residual energy does not follow a simple prediction from critical dynamics alone, as also discussed noted in Ref. [31].

In our work, we are therefore primarily focused on the quality of the solutions and solutions, using the residual energy. Therefore, we use scaling (Eq. (3)) simply as an empirical comparison of scaling exponents for optimizers of interest; we benchmark for different optimizers. We do not attempt to map our data onto specific KZM exponents, nor do we assume that near-critical power laws fully govern the eventual solution quality for these optimization problems.

Fig. 2. Scaling improvement of the discrete-time simulated quantum annealing (DT-SQA) algorithm: The final residual energy, ρ_E^f , as a function of annealing time, t_a (in Monte Carlo sweeps, (MCS)), for DT-SQA simulations with varying Trotter replicas, R , is shown for 3D Ising spin glass problems ($15 \times 15 \times 12$ with 2687 spins—using the logical instances in Ref. [32] with missing spins) using CPUs. The slope increases in absolute value with R . See Supplementary Fig. S1 for a discussion on the x-axis for part (a). Each data point is averaged over 300 instances and 50 independent runs. The best energy among all Trotter replicas is selected at the end of t_a . Error bars denote the 95% bootstrap confidence interval of the mean across spin-glass instances. An MCS is an algorithmic unit representing one update attempt per spin; hardware differences alter the wall-clock time per sweep but not the scaling exponent κ_f . The comparison with the quantum annealer’s physical time in (b) is therefore a comparison of this dimensionless exponent, not a direct conversion of time units. A full analysis of wall-clock time, which incorporates these hardware-specific prefactors, is presented in Section V and Fig. 4. (b) Comparison of the DT-SQA and scaling exponent with that of the quantum annealer (QA) slopes from Ref. [32] for ρ_E^f shows. With a sufficient number of replicas ($R = 2850$), DT-SQA outperforming QA in achieves a more favorable scaling of final residual energy exponent. See Supplementary Fig. S2 for similar results on embedded instances. (c) The slope improvement is plotted against R , showing alignment with extreme value theory predictions (red dashed line; see Supplementary Section II). The DT-SQA slopes outperform scaling exponent becomes comparable to the QA when $R > 2800$ ’s at $R \approx 2800$ and exceeds it for larger values. Error bars denote 95% confidence interval of fitting.

III. Analysis of Residual Energy Scaling with DT-SQA

The DT-SQA is an annealing-based algorithm inspired by the principle of adiabatically reducing the transverse field in a quantum system. Using the well-known Suzuki-Trotter transformation [39], a d -dimensional quantum system is mapped onto a $(d+1)(d+1)$ -dimensional classical system. The additional “imaginary-time” dimension is composed of R interconnected Trotter replicas of the original quantum system, where qubits are replaced by Ising spins. As proposed in Ref. [40], our strategy is to implement the DT-SQA algorithm directly on probabilistic hardware, using distinct physical replicas.

In Fig. 2(a), we evaluate the scaling performance of DT-SQA by plotting ρ_E^f as a function of the annealing time, t_a , with varying R . The inverse temperature is set to $\beta = 0.5R$ and the simulations were performed on CPUs using logical instances of size $15 \times 15 \times 12$, obtained problem instances defined on a 3D cubic lattice of Ising spins with dimensions $15 \times 15 \times 12$, obtained directly from Ref. [32]. The results show that the absolute value of the slope, κ_f , increases with R when the minimum energy among the R replicas is selected. A comparison in Fig. 2(b) reveals that the scaling exponent of DT-SQA reaches the scaling of becomes comparable to that of the quantum annealer around $R = 2850$ replicas (with $\kappa_f = 0.805$) and above. It is important to note that this is a comparison of the dimensionless scaling exponent, κ_f , which is independent of the units on the time axis (MCS for p-computers, nanoseconds for the QA). The QA residual energy data is multiplied by a factor of 2 to align with logical instances, based on the observation that broken dimers (when physical spins representing the same logical spin do not agree

after annealing) are rare [32]. In Ref. [32], $\kappa_f = 0.785$ and $\kappa_f = 0.51$ are quoted for the quantum annealer and CT-SQA algorithm, respectively.

The slopes quoted above are based on embedded instances because quantum annealers (QA) (logical problem instances mapped onto the quantum annealer’s physical qubit connectivity graph). Quantum annealers (QAs) typically require complex embedding schemes to represent for combinatorial optimization problems on a fixed and inflexible topology, in which a single logical spin is represented by multiple physical spins grouped into structures called dimers, due to their fixed hardware topology (such as the Chimera or Pegasus graphs), even for sparse problems, such as relatively sparse problems like 3D spin glasses. On the other hand, probabilistic computers can natively represent these problems without any implemented on flexible classical hardware, such as FPGAs or ASICs, can directly represent and solve the logical problem graph without embedding overhead. Since our point of comparison is to solve practical goal is to evaluate the intrinsic performance of algorithms solving practical combinatorial optimization problems, we do not artificially our main results do not include embedding overheads of quantum annealers in our main results that are specific to current quantum annealer architectures. Nevertheless, we essentially obtain similar results also on embedded graphs (see Supplementary Fig. S2) noting that the DT-SQA algorithm can match the scaling of quantum annealers in both cases.

Although both our work and Ref. [32] employ SQA, our approach and goals are different. Ref. [32] uses the continuous-time variant (CT-SQA) as a theoretical baseline, whereas we deliberately use the discrete-time SQA (DT-SQA) with R explicit Trotter replicas. We chose DT-SQA as it

maps naturally onto parallel hardware architectures, making it a more relevant algorithm for assessing the performance of physically realizable classical systems. Therefore, we benchmark our hardware-amenable algorithm directly against the quantum annealer's performance, rather than reproducing the CT-SQA baseline. In Monte Carlo simulations whose goal is to faithfully simulate the equilibrium physics of the quantum system intended to accurately emulate equilibrium quantum physics, selecting the best replica among Trotter replicas should be avoided [41, 42]. There is no such concern when solving optimization problems since quantum emulation is not a target, however. Since best-performing replica among multiple Trotter replicas is usually avoided, as this could bias equilibrium observables [41, 42]. However, such concerns are not relevant in our context, because our goal is not quantum emulation but rather practical combinatorial optimization. Indeed, since our replicas represent independent physical entities realized by separate physical entities implemented with physical spins in hardware selecting the best residual energy among all replicas is natural in our context, identifying and selecting the replica with the lowest residual energy is both natural and appropriate.

Next, we show that the observed increase in slope κ_f with respect to R can be explained using extreme value theory (EVT; see Supplementary Section II for details) with modifications to account for correlations among replicas. In conventional EVT, the minimum energy is selected from P independent runs of an algorithm, shifting the expected value of the minimum energy by $\mathcal{O}(\sqrt{\ln P})$ from the mean of the original distribution ($P = 1$). In DT-SQA, the Trotter replicas are interconnected and correlated, complicating a direct application of EVT. We observe however that the replica correlations decay over a distance, allowing us to extract effectively independent block sizes. Another complication is the dependence of the correlations with t_a , as the transverse coupling among the replicas (J_\perp) strengthens (see Supplementary Section I and Supplementary Fig. S6). To apply an EVT theory despite these complications, we partition the R Trotter replicas into P effective blocks, where replicas within a block are correlated but largely uncorrelated with other blocks. We then treat these blocks as separate runs and observe that the predicted scaling behavior that aligns closely with the slopes observed in Fig. 2(c). The sizes of the extracted effective blocks correspond well to the measured replica-to-replica correlations (Supplementary Fig. S7(b)), providing further support for our modified EVT analysis.

While the modified EVT approach explains the increase in the slope of DT-SQA as a function of the number of replicas, the algorithm is not inherently parallel due to the interconnections among the Trotter replicas. A more direct approach is also possible how increasing replicas within a single run improves the scaling. An alternative strategy, also based on EVT, is to leverage multiple independent runs: use a fixed number of interconnected Trotter replicas, run the algorithm P times independently and then select the best energy from all runs. We find that by setting $R = 32$ and running $P = 50$ independent iterations (a total

of 1600 replicas), followed by selecting the best solution, DT-SQA also achieves slopes comparable to those of the quantum annealer (see Supplementary Fig. S5). However, this approach remains valid over a shorter range of t_a before the power law breaks down and transitions to a flat plateau region (see Supplementary Fig. S8), showing that the two approaches are not equivalent. Nevertheless, both DT-SQA approaches— $R = 2850$, $P = 1$ (shown in Fig. 2) and $R = 32$, $P = 50$ —are feasible for implementation on a single classical chip, as we discuss in Section VI where large groups of spins on the chip can be updated simultaneously with massive parallelism.

It is important to note that there is a fundamental difference between the two approaches we demonstrated. When $P = 1$, all the DT-SQA replicas are connected and we perform a single run of the algorithm to match (and exceed as needed) QA slopes. The large number of replicas necessary to match the QA slopes is a strong evidence of the efficiency of the quantum annealer, nonetheless, it can be concluded that the QA data does not establish a scaling advantage over our results show that DT-SQA equipped with a sufficient number of replicas, when equipped with sufficient replicas, can achieve a comparable or more favorable scaling exponent. Naturally, the second approach where we pick the best results out of $P > 1$ runs can also be applied to the quantum annealer to increase residual energy slopes, however, this does not change our main findings for $P = 1$.

Finally, we note that while κ_f is a useful metric, it may not fully capture the practical relevance of algorithms for large-scale optimization. As discussed in Section IV, DT-SQA, despite better initial scaling at a fixed problem size $L = 15$ exhibiting a steep power-law decay in residual energy at early times in instances defined on a fixed 3D spin glass lattice of size $15 \times 15 \times 12$, transitions into a plateau and at longer annealing times, where the residual energy stagnates and shows little further improvement. It is ultimately outperformed by the APT algorithm, which is easier to implement and parallelize in hardware. APT achieves significantly lower residual energies with identical computational resources, showing that relying solely on κ_f as a performance metric can be misleading, as different algorithms may exhibit distinct scaling behaviors at different stages of optimization, which are crucial for real-world applications.

IV. Comparison with Adaptive Parallel Tempering

We now compare DT-SQA with the powerful replica-based adaptive parallel tempering (APT) algorithm, widely considered as the state-of-the-art for solving spin-glass problems [43–49]. APT also utilizes replicas of the problem graph, but these run in parallel at different temperatures, with adjacent replicas periodically swapped based on the Metropolis criterion:

$$p_{\text{swap}} = \min(1, \exp(\Delta\beta\Delta E)), \quad (4)$$

Fig. 3. Scaling improvement of the APT algorithm with ICM: (a) Iso-Trotter replica comparison of DT-SQA and adaptive parallel tempering (APT) with isoenergetic cluster moves (ICM) on CPUs for the same logical instances as in Fig. 2, using 50 independent runs per instance. A sweep-to-swap ratio of 1 minimizes residual energy for APT. While DT-SQA approaches a flat plateau, APT shows a sharp bending towards lower residual energy. (b) Finite-size scaling analysis for various cube sizes (L) reveals a universal collapse of the data. The later bending appears to be universal for APT, as shown by averaging over 200 independent runs per instance. For each size, 4 ICM replicas are used, with the total number of replicas varying due to the adaptive nature of the pre-processing algorithm. A similar collapse for APT without ICM is provided in the Supplementary Fig. S19. Error bars represent the 95% bootstrap confidence interval of the mean across spin-glass instances.

where $\Delta E = E_{i+1} - E_i$ is the energy difference, and $\Delta\beta = \beta_{i+1} - \beta_i$ is the difference in inverse temperatures between replicas i and $i+1$ (with $\beta_i < \beta_{i+1}$). This mechanism enables high-temperature replicas to explore the energy landscape broadly while low-temperature replicas preserve optimal states. The adaptive variant further optimizes the algorithm by preprocessing the problem graph to equalize swap probabilities across replicas, avoiding bottlenecks [49, 50].

Fig. 3(a) compares DT-SQA and APT for the same problem as in Fig. 2. Adaptive preprocessing produces approximately 33 temperature replicas per instance (see Section VIII). To further enhance APT, we incorporate isoenergetic cluster moves (ICM) [36–38], which, as we demonstrate later, play a crucial role. ICM are non-local Monte Carlo swaps added on top of the standard APT algorithm. A swap attempt follows each network sweep, maintaining a sweep-to-swap ratio of 1. Using 4 replicas per temperature for ICM, the APT algorithm used in this work operates with a total of 132 replicas. As shown in Supplementary Fig. S16, we found that this sweep-to-swap ratio produces the smallest residual energy for a fixed MCS budget.

Optimization of APT parameters (detailed in Supplementary Section IV) reveals that the initial slope of the optimized APT with ICM corresponds to $\kappa_f = 0.53$, slightly lower than DT-SQA with a similar number of replicas ($\kappa_f = 0.647$ at $R = 132$). However, APT achieves better absolute residual errors lower residual energy for a given MCS budget. Although DT-SQA initially shows a better slope, it plateaus at higher MCS, as shown in Fig. 3(a). This trend is observed across various cube sizes L and Trotter replicas R (see Supplementary Fig. S3) and is consistent with previous findings [41]. In contrast, APT with ICM shows two distinct scaling regimes: an initial gentler slope followed by a steeper one (see Supplementary Figs. S14(a) and S15). Notably, the APT algorithm without ICM does not exhibit this steeper bending, even with the

same number of replicas (see Supplementary Fig. S14(a) and Supplementary Fig. S19). The presence of this bending suggests the potential for algorithms that incorporate non-local and non-equilibrium moves [51] to further enhance the performance of probabilistic approaches in solving hard optimization problems. As before, a similar performance characteristic is observed for embedded instances (see Supplementary Fig. S17).

This steeper slope is also observed for other cube sizes L (see Supplementary Fig. S18(a)) and appears to be a universal feature of APT with supplemented by ICM. Finite-size scaling analysis confirms that the residual-energy curves for different sizes collapse onto a single universal curve (Fig. 3(b)). However, at very low residual energies near the ground state, we observe another transition to a gentler slope (not visible in Fig. 3(b)). This transition occurs at residual energies that are very close to the uncertainty limit of the ground energy estimations used in our analysis (see Supplementary Fig. S18(b)). As such, it is difficult to reliably confirm the existence of this feature. Whether similar steeper slopes at longer annealing times can be observed in quantum annealers, in particular for problems spanning an entire processor, remains an open question, as current hardware faces challenges such as limited coherence times. On the other hand, the robust universal collapse shown in Fig. 3(b) allows us to extrapolate the time required to reach a target residual energy for cubes of any size.

V. Architecture-based scaling improvement and massive parallelism

Beyond scaling improvements, a critical metric for optimization problems is achieving the lowest possible residual energy within a given amount of time. Here, we evaluate the relative performance of CPU, FPGA (see Supplementary Section V for the details of FPGA

Fig. 4. Architectural improvement with probabilistic computers in hardware: (a) Algorithmic complexity of the 3D spin glass problem as a function of the number of p-bits, showing the minimum number of Monte Carlo sweeps (MCS) required to reach a target threshold residual energy, $\rho_{E_0}^f = 0.007$, using APT + ICM, independent of hardware platform. For each lattice size (L), 300 logical instances with 50 independent runs are reported. Swaps are probabilistically performed pairwise between adjacent β replicas, alternating between even and odd pairs. Error bars (95% confidence) are small and often invisible. (b) Measured average MCS time per replica for CPU and FPGA. CPU shows $\mathcal{O}(n)$ scaling, while FPGA achieves $\mathcal{O}(1)$ scaling due to the massively parallel p-computer architecture. sMTJ projections assume experimentally demonstrated nanosecond p-bits [52, 53]. (c) Total estimated time to reach $\rho_{E_0}^f = 0.007$, combining (a) and (b). Time for ICM and swaps are negligible compared to sampling times (see Methods section) and are excluded from this estimation. Also, it is assumed that FPGA can run all the replicas in parallel (as long as they fit on a single chip [19]; see Supplementary Section III). We assume the same for CPU as well and do not include the replica factor when estimating these times. FPGA maintains $\mathcal{O}(n)$ improvement over CPU, while ‘prefactor improvement’ for sMTJ-based p-computers refers to the additional constant speed-up expected when the same architecture is implemented with fast, on-chip sMTJ p-bits (approximately 1 ns intrinsic flip time), relative to the measured FPGA sweep time. sMTJ projections assume a single chip with 1 million sMTJs, achieving 1 million flips per nanosecond ns (the thick orange line assumes 10 and 50 replicas for the upper and lower bounds, respectively). Prefactor: This also includes improvements stemming from additional parallelization, meaning that if the problem size is smaller than the chip’s capacity, multiple independent runs or problem instances can be processed simultaneously on the same chip, scaling as $10^6 / (\text{spins per replica} \times \text{APT replicas})$.

implementation), and sMTJ-based implementations of p-computers, highlighting architectural advantages. One key feature of the p-computer architecture we adopt here is the ability to probabilistically update all spins in the system in constant time. This differs from the sequential or partially parallel updates typically used in software implementations. In sparse networks/problems, such as the nearest-neighbor 3D spin-glass instances considered in this work, spin-glasses, planted Ising benchmarks [30, 54] or circuit SAT problems with sparse connectivity [4], p-computer architectures leverage physically parallel nodes to simultaneously update large independent sets [4]. Fig. 4(a) shows the number of Monte Carlo sweeps (MCS) per replica: one MCS involves one update attempt per spin for all spins in the network needed to reach a target residual energy threshold of $\rho_{E_0}^f = 0.007$, an arbitrarily chosen optimization goal given a computational budget. Using the APT + ICM algorithm, we show the required number of MCS as a function of the lattice size L . Fig. 4(b) compares the time required for one sweep per replica across three architectures: CPU, FPGA-based p-computers, and sMTJ-based p-computers. While sweep times increase with problem size on CPUs, they remain constant, $\mathcal{O}(1)$, for FPGA- and sMTJ-based p-computers, exploiting massive parallelism until resource limits are reached. For the problem considered, $n/2$ or $n/4$ spins (depending on whether logical or embedded instances are used) can be updated simultaneously, resulting in an $\mathcal{O}(n)$ performance improvement over CPUs. The current FPGA implementation yields up to 185 flips/ns approaching performance of the state-of-the-art GPUs and FPGA-based

simulators [31, 55], with further possible improvements using specialized ASICs and nanodevices. Note however, that the p-computers we propose here can support arbitrary sparse graph topologies beyond the regular and more easily parallelizable topologies shown in [31, 55]. Fig. 4(c) combines results from (a) and (b), showing total runtime as the product of sweep count and average sweep time. The analysis confirms an $\mathcal{O}(n)$ scaling advantage for p-computers over CPUs. Furthermore, sMTJ-based devices could achieve an additional 1 to 3 orders of magnitude improvement in prefactors, assuming nanosecond fluctuations in a 1-million p-bit MRAM chip, which are feasible.

The proposed architecture is also highly energy efficient, consuming 2 to 5 orders of magnitude less energy per flip compared to the state-of-the-art GPUs and TPUs used for probabilistic tasks. Our FPGA implementation consumes 9.168 W, which corresponds to 5×10^{-2} nJ/flip. sMTJ-based devices, with 1 million p-bits integrated on a single MRAM chip, reduce this further to 2×10^{-5} nJ/flip assuming 20 W power consumption [56]. In comparison, a NVIDIA Tesla V100 GPU consumes 21.99 nJ/flip, and a Google TPU v3 requires 7.77 nJ/flip to solve probabilistic problems on simpler graphs [16].

There is a trade-off between reconfigurable and application-specific hardware. FPGAs offer full reconfigurability, ideal for algorithmic exploration across diverse problem structures, but at a significant performance and energy cost. For the 3D spin glasses studied here, the fixed nearest-neighbor topology is an excellent match for a custom ASIC. Connectivity can be hard-wired while

programmability for different instances is retained by reloading different weights. For problems with arbitrary sparse topologies, however, achieving reconfigurability on static ASICs is an open problem and may require different approaches, such as higher-order problem formulations or master graph approaches [19, 57].

VI. Physical design feasibility of single p-computing chips

We now evaluate the feasibility of ~~implementing a physical a custom Application-Specific Integrated Circuit (ASIC) designed for replica-based algorithms on sparse, structured problems. A monolithic chip that can house a large number of replicas, required for DT-SQA~~ all replicas on-die would eliminate the off-chip communication overhead that constrains current FPGA implementations. To make realistic projections for a full-scale ASIC, our analysis is grounded in a rigorous, bottom-up physical design flow using a 7 nm process. Our findings on chip capacity for the DT-SQA algorithm apply equally to the APT algorithm which requires fewer replicas to match and surpass DT-SQA and quantum annealer performance, as we discuss better performing APT algorithm, which requires significantly fewer replicas, as shown in Section IV.

The details of the p-computer architecture are discussed in Supplementary Section III. As shown in Supplementary Fig. S11 and Supplementary Table S1, a full place-and-route analysis was performed using the ASAP7 7 nm open-source process design kit (PDK) [58] for up to 5 replicas of the $15 \times 15 \times 12$ logical instances. The analysis revealed an approximately linear growth (with a slope of 1.05) in chip area requirements.

Based on this scaling, we project that approximately 7.66 million p-bits – corresponding to 2850 replicas can fit on a single chip using 7 nm technology. This translates to a chip area of $28.61 \times 28.61 \text{ mm}^2$, which is well within the capabilities of current fabrication technology. Furthermore, multiple such chips can be interconnected to support even larger numbers of p-bits as needed. With advances in fabrication technology and the adoption of nanodevice-based p-bits, the number of p-bits per chip can be significantly increased, enabling even greater scalability.

VII. Outlook

This paper demonstrates that probabilistic computing with p-bits provides a practical and scalable approach to solving 3D Ising spin glass problems. Using the DT-SQA algorithm, we showed how leveraging a large numbers of replicas greatly improves scaling exponents, well-explained by extreme value theory. We further explored powerful algorithms like APT supported with non-local moves, significantly improving scaling and time-to-solution. Finite-size scaling analysis revealed a universal collapse of residual energy curves for APT, emphasizing the generality of our results. This makes APT particularly well-suited

for large-scale optimization tasks when implemented on dedicated probabilistic hardware, as demonstrated by our FPGA-based implementation, achieving high performance through hardware acceleration.

~~We would like to emphasize that as hardware accelerators the existence of classically hard to simulate quantum dynamics during an annealing schedule is neither necessary nor sufficient for observing speedup over certain state-of-the-art solvers for combinatorial optimization problems. For example, quantum annealers can enable probabilistic asynchronous updates with significant speedup against local synchronous updates without having non-trivial quantum fluctuations to be present or to be computationally relevant.~~

Advances in fabrication technology now allow large-scale replica systems, delivering orders-of-magnitude speedups compared to software methods. Projections for monolithic nanodevice-based p-computers highlight ~~their potential to surpass current solvers~~ a path toward performance competitive with and potentially exceeding current solvers, all while operating at room temperature while avoiding challenges like decoherence and embedding overheads faced by quantum annealers. and without the specific hardware challenges of qubit decoherence or fixed-connectivity embedding. Co-designed together, powerful algorithms, scalable architectures, and emerging hardware provide a clear pathway for solving hard optimization problems at unprecedented scales. Beyond combinatorial optimization, probabilistic computers hold promise for diverse applications including training and inference in energy-based models and Bayesian learning and in general for sampling over discrete spaces where the performance of traditional solvers have saturated.

VIII. Methods

p-computing overview

p-computing relies on an interacting network of p-bits σ_i , which generate two-valued outputs ($\sigma_i \in \{-1, +1\}$) and are governed by two key equations [59]:

$$I_i = \sum_j J_{ij} \sigma_j + h_i \quad (5)$$

$$\sigma_i = \text{sgn}(\tanh(\beta I_i) - r_{[-1,1]}) \quad (6)$$

Here, J , h , and β represent the interconnection matrix, bias vector, and inverse temperature, respectively. $r_{[-1,1]}$ is a random number uniformly distributed in the range $[-1, 1]$. Equations (5) and (6) collectively approximate the Boltzmann distribution:

$$p(\{\sigma_i\}) = \frac{1}{Z} \exp(-\beta E(\{\sigma_i\})) \quad (7)$$

$$E(\{\sigma_i\}) = - \sum_{i < j} J_{ij} \sigma_i \sigma_j - \sum_i h_i \sigma_i \quad (8)$$

where $p(\{\sigma_i\})$ represents the probability and $E(\{\sigma_i\})$ represents the energy of the state $\{\sigma_i\}$, with Z as the partition function.

Instances

For comparison, we use the instances from Ref. [32] for ~~$L=8, 10,$ and 15~~ $L = 8, 10,$ and instances of size $15 \times 15 \times 12$. These instances have open boundaries along the x and y directions and periodic boundaries along the z direction. For sizes greater than ~~$L=9$~~ $L = 9$, some spins are missing due to embedding constraints of the quantum annealer. Consequently, the total number of qubits for ~~$L=10$~~ $L = 10$, for example, is 999, instead of the expected ~~$L^3=1000$~~ $L^3 = 1000$. We also use the putative ground energies reported in Ref. [32] for these sizes. However, for a few instances, we found lower ground state energies than those reported and therefore used the improved ground energies in our analysis.

For ~~$L=12$~~ $L = 12$ and 16, we generate instances using the codes provided in Ref. [32], ensuring that these instances do not suffer from missing spins. For these instances, the putative ground state energies are obtained by running APT with ICM algorithm up to 10^7 sweeps (in each sweep all replicas are updated once), in a single run with a sweep-to-swap ratio of 10, choosing the minimum energy found along the whole simulation, and following a similar fitting and limiting procedure discussed in [32]. Our corresponding approximate error estimate per site for these sizes, is 2.5×10^{-4} (attributed to the increased problem size and the use of a single run) as indicated in Supplementary Fig. S18. This is comparable to the estimated error in the mean ground state energy per site, 4×10^{-5} , as reported in Ref. [32] for ~~instances of lattice~~ size $15 \times 15 \times 12$. The residual energy ranges used in this work to draw our conclusions are well above the range of these errors, or otherwise carefully discussed.

Graph Coloring of 3D Cubic Spin Glass Instances

If two p-bits in a network are not connected, they can be updated in parallel [4]. Graph coloring assigns colors to the network such that any two connected p-bits are given different colors, while p-bits that are not connected can share the same color. This enables massive parallelism for sparse graphs even if they are irregular, allowing a network of p-bits to be updated in constant time, regardless of network size.

A perfect 3D lattice is bipartite and easily 2-colorable. However, the D-Wave instances have missing spins and complex embeddings (due to hardware constraints), which necessitates graph coloring. In this work, graph coloring is performed using DSATUR [60], a heuristic graph coloring algorithm with polynomial-time complexity. Since the underlying graph is identical for all problem instances of a given size, we perform graph coloring for one representative instance of each size as a preprocessing step. These problem instances typically require 2 to 4 colors, depending on their

connectivity. In DT-SQA, replicas are connected and periodic boundary condition is applied. As a result, networks with odd numbers of replicas require an extra color.

Annealing Schedule of DT-SQA

Supplementary Section I details the description of the DT-SQA algorithm. Annealing is performed by gradually changing the transverse field (Γ_x) from a high value to 0. Change in Γ_x is reflected in the coupling strength J_\perp (see Supplementary Eq. (S.3)), which couples the spins of two neighboring replicas.

In our implementation, we use a slightly modified form for J_\perp :

$$J_\perp(t) = -\frac{1}{\beta} \ln \tanh \left(\frac{\beta \Gamma'_x(t)}{R} \right) \quad (9)$$

and anneal $\Gamma'_x(t)$ linearly, from 3.0 to 0. This modification represents a transformation between Γ'_x and Γ_x and does not alter the underlying physics. We also set $\beta/R = 0.5$ in all our simulations.

APT details

For the APT algorithm, we start with a preprocessing step to compute the inverse temperature (β) schedule and determine the required number of replicas. We perform the preprocessing individually for each of the 300 instances, even though schedules and number of replicas obtained are similar (see Supplementary Section IV A). Specific details of the preprocessing algorithm we adopted can be found in [19, 51] and Supplementary Algorithm S2. For our simulations, we set the initial inverse temperature to $\beta_0 = 0.5$ and the temperature update factor to $\alpha = 1.25$. We calculate the average energy variance across 100 parallel chains, where the variance for each chain is computed from the last 1000 sweeps of a 10000-sweep run before updating the temperature schedule. This process is repeated until the average energy variance drops below $\min(J_{ij})/2$. For the 300 instances with ~~$L=15$~~ ~~lattice~~ ~~size~~ $15 \times 15 \times 12$, the number of replicas ranged from 32 to 34.

After determining the β schedule, each instance is simulated using the parallel tempering algorithm, both with and without the isoenergetic cluster moves (ICM) [36–38]. We employ 4 ICM replicas per temperature. During simulation, each replica undergoes a fixed number of sweeps before a swap is attempted. A swap attempt involves performing an isoenergetic cluster move for each of the two randomly chosen ICM replica pairs at each temperature. This is followed by a swap attempt between neighboring replicas, determined as follows: for an odd-numbered swap attempt, pairs (1, 2), (3, 4), ... are swapped; for an even-numbered swap attempt, pairs (2, 3), (4, 5), ... are swapped. Within a given β , ICM replicas are labeled a, b, c, d , and swaps between neighboring β values occur only between replicas with the

same label. The algorithm is detailed in Supplementary Algorithm S2.

APT collapse

The collapse of APT with ICM residual energies was obtained using the open-source library `autoScale.py` [61]. The parameter b fluctuates slightly around 3.0; we use $b = 3.0$, as it intuitively reflects the fact that the residual energy behaves extensively, scaling with the system size, L^3 .

FPGA implementation of APT with ICM algorithm
We implement the physics-inspired massively parallel p-computer architecture [4] on an FPGA (CPU details)

All CPU-based simulations were run on a 10-core Intel Core i9-10900 processor (2.80 GHz) with 64 GB RAM, using MATLAB R2023b on a 64-bit Windows 10 machine. p-bits were updated sequentially using Gibbs sampling. All computations used MATLAB's default double-precision arithmetic and Mersenne Twister pseudorandom number generator (PRNG).

FPGA details

We mapped the physics-inspired, massively parallel p-computer architecture of Ref. [4] onto a Xilinx Alveo U250 Data-Center-Accelerator-Card) data-center accelerator card using graph coloring ~~for~~ to maximize parallelism on the sparse instances. ~~We~~ All arithmetic is fixed-point: DT-SQA uses $s\{6\}\{3\}$ precision (1 sign, 6 integer, 3 fractional bits) while APT + ICM uses the higher $s\{6\}\{6\}$ precision. Custom RTLs were developed to implement the algorithm based on the p-computing architecture and synthesized, placed and routed with Xilinx Vivado/Vitis tool chain. Further details are provided in Supplementary Sections III (DT-SQA) and V (APT).

FPGA implementation of APT with ICM algorithm

At this time, we can accommodate only 1 replica in a single FPGA for large scales, such as ~~$L = 15$~~ instances of size $15 \times 15 \times 12$. To implement APT for this size, 32 to 34 replicas are required, while APT with ICM requires 128 to 136 replicas. We address this limitation by employing time-division multiplexing (TDM), allowing the same hardware to be reused for multiple replicas. At the start of each run, the weights of all replicas (scaled by β) are loaded into the BRAM. During each sweep, based on the replica index that will be sampled next, the weights corresponding to that replica are dynamically fetched from the BRAM. Then after the sampling is done, the state of p-bits of the current replica are also stored in the BRAM. This process is repeated until all replicas are sampled once. Then from MATLAB, we

read all the p-bits states, (perform ICM whenever applicable) compute energies and perform the swaps. For the subsequent swap attempts, replicas are reinitialized either from the state saved in the previous swap or from the new biases (to restore certain p-bits to their original states before hardware was reused) generated by the APT swap/ICM. After initialization, the biases are released, and the p-bits resume their usual MCS at their respective β .

Measurement of flips per second and time per MCS

To measure the time per MCS in the FPGA accurately, we implemented precise counters within the FPGA to track the number of flip attempts made by each p-bit during a fixed time interval. This reference interval is determined by a predefined counter running at 125 MHz, which counts up to 50000. All counters are simultaneously enabled by a global signal from MATLAB and stop when the reference counter completes its count. This corresponds to an elapsed time of $50000 / (125 \times 10^6)$ seconds (= 400 μ s). The total flip attempts during this period are summed across all p-bit counters to compute the total flips per second (fps).

Since one MCS involves updating all p-bits in a single replica, the time per MCS is calculated using the total measured flips and the elapsed time. For each network size L , we performed 100 measurements and reported the average time per MCS. FPGA measurements are instance-independent. We reported MCS times for a single replica, given our detailed feasibility analysis that shows all relevant replicas sizes we considered can fit on a single chip. This full integration would eliminate the overhead for time-division multiplexing. It should be noted that, at present, the overhead from swapping and ICM moves dominates the MCS time because it is performed off-chip on the CPU. This is not a fundamental limitation: the overhead can be computed directly on the FPGA rather than off-chip on the CPU. Using standard hardware design flows, such as those enabled by high-level synthesis (HLS), these computations can be seamlessly implemented on-chip, thus making their contribution negligible compared to the MCS time. Consequently, it was excluded from the FPGA measurements. We emphasize that none of the overheads that are omitted here affect our scaling exponent measurements where prefactors in time per MCS does not affect slopes in power laws.

For CPU measurements, MATLAB's built-in `tic` and `toc` functions were used to measure the time taken to perform 100 MCS across 10 instances with 10 runs each. The average time per MCS is reported for a single replica of each network size L . Swap times were similarly excluded from CPU measurements to ensure a fair comparison with FPGA performance. For both FPGA and CPU measurements, error bars represent 95% confidence intervals and were computed using bootstrapping with replacement.

DATA AVAILABILITY

The data that support the findings of this study are available from the corresponding author upon reasonable request.

CODE AVAILABILITY

Simplified MATLAB implementations of the SQA and APT + ICM algorithms and the instances used in this study are openly available at <https://github.com/OPUSLab/3DSpinGlassWithPbits.git>. Pseudocode for both routines is provided in Supplementary Algorithms S1 and S2.

Acknowledgments

SC, KYC, and NAA acknowledge support from the Office of Naval Research (ONR) Young Investigator Program grant, the National Science Foundation (NSF) CAREER Award under grant number CCF 2106260, the Army Research Laboratory under grant number W911NF-24-1-0228, the Semiconductor Research Corporation (SRC) grant, and the ONR-MURI grant N000142312708. Use was made of computational facilities purchased with funds from the National Science Foundation (CNS-1725797) and administered by the Center for Scientific Computing (CSC). The CSC is supported by the California NanoSystems Institute and the Materials Research Science and Engineering Center (MRSEC; NSF DMR 2308708) at UC Santa Barbara. AG, ER, and GF acknowledge the support of project number 101070287 — SWAN-on-chip — HORIZON-CL4-2021-DIGITAL-EMERGING-01; the MUR-PNRR project SAMOTHRACE (ECS00000022), funded by European Union (NextGeneration EU); the projects PRIN 2020LWPKH7 – The Italian factory of micromagnetic modeling and spintronics and PRIN 20225YF2S4 – Magneto-Mechanical Accelerometers, Gyroscopes and Computing based on nanoscale magnetic tunnel junctions (MMAGYC), funded by the Italian Ministry of University and Research. PAL, NAA and MM were supported in part under NSF CCF (grant 1918549). PAL was also supported in part through the NASA Academic Mission Services (contract NNA16BD14C) under SAA2-403506, as well as the Intelligent Systems Research and Development-3 (ISRDS-3) Contract 80ARC020D0010 under SAA2-403688. JHM acknowledges funding from the VIDI project no. 223.157 (CHASEMAG) and KIC project no. 22016 which are (partly) financed by the Dutch Research Council (NWO), as well as support from the European Union Horizon 2020 and innovation program under the European Research Council ERC Grant Agreement No. 856538 (3D-MAGiC) and the Horizon Europe project no. 101070290 (NIMFEIA). FRT received financial support from the “National Centre for HPC, Big Data and Quantum Computing - HPC”, Project CN-00000013, CUP B83C22002940006, NRP Mission 4

Component 2 Investment 1.5, funded by the European Union - NextGenerationEU. TS and AR acknowledge support from NSF FuSe2 Award 2425218, Carnegie Mellon University Dean’s Fellowship and Tan Endowed Graduate Fellowship in Electrical and Computer Engineering, Carnegie Mellon University. TS and AR also acknowledge Tong Wu for discussions.

IX. Author Contributions

SC, MM and KYC conceived the study. KYC supervised the study. SC, AG, ER, NAA performed different parts of the simulations. NAA and SC performed FPGA experiments. AR and TS performed the physical design simulations for chip design. MMR, FRT, MM analyzed the scaling data with critical feedback. SC wrote the initial draft of the manuscript with inputs from PAL, JHM, MMR, FRT, MC, LST, GF, MM, KYC. All authors contributed to improving the draft and participated in designing the experiments, analyzing the results, and editing the manuscript.

X. Competing Interests

The authors declare no other competing interests.

References

- [1] Richard P. Feynman. Simulating physics with computers. *International Journal of Theoretical Physics*, 21(6):467–488, 06 1982. ISSN 1572-9575.
- [2] Logan G Wright, Tatsuhiro Onodera, Martin M Stein, Tianyu Wang, Darren T Schachter, Zoey Hu, and Peter L McMahon. Deep physical neural networks trained with backpropagation. *Nature*, 601(7894):549–555, 2022.
- [3] Maxwell Aifer, Kaelan Donatella, Max Hunter Gordon, Samuel Duffield, Thomas Ahle, Daniel Simpson, Gavin Crooks, and Patrick J Coles. Thermodynamic linear algebra. *nj Unconventional Computing*, 1(1):13, 2024.
- [4] Navid Anjum Aadit, Andrea Grimaldi, Mario Carpentieri, Luke Theogarajan, John M Martinis, Giovanni Finocchio, and Kerem Y Camsari. Massively parallel probabilistic computing with sparse Ising machines. *Nature Electronics*, 5(7):460–468, 2022.
- [5] Masoud Mohseni, Artur Scherer, K. Grace Johnson, Oded Wertheim, Matthew Otten, Navid Anjum Aadit, Yuri Alexeev, Kirk M. Bresniker, Kerem Y. Camsari, Barbara Chapman, Soumitra Chatterjee, Gebremedhin A. Dagnaw, Aniello Esposito, Farah Fahim, Marco Fiorentino, Archit Gajjar, Abdullah Khalid, Xiangzhou Kong, Bohdan Kulchytsky, Elica Kyoseva, Ruoyu Li, P. Aaron Lott, Igor L. Markov, Robert F. McDermott, Giacomo Pedretti, Pooja Rao, Eleanor Rieffel, Allyson Silva, John Sorebo, Panagiotis Spentzouris, Ziv Steiner, Boyan Torosov, Davide Venturelli, Robert J. Visser, Zak Webb, Xin Zhan, Yonatan Cohen, Pooya Ronagh, Alan Ho, Raymond G. Beausoleil, and John M. Martinis. How to build a quantum supercomputer: Scaling from hundreds to millions of qubits. *arXiv preprint arXiv:2411.10406*, 2024.

- [6] Peter W Shor. Algorithms for quantum computation: discrete logarithms and factoring. In *Proceedings 35th annual symposium on foundations of computer science*, pages 124–134. IEEE, 1994.
- [7] Frank Arute, Kunal Arya, Ryan Babbush, Dave Bacon, Joseph C Bardin, Rami Barends, Rupak Biswas, Sergio Boixo, Fernando GSL Brandao, David A Buell, et al. Quantum supremacy using a programmable superconducting processor. *Nature*, 574(7779):505–510, 2019.
- [8] Seth Lloyd, Masoud Mohseni, and Patrick Rebentrost. Quantum principal component analysis. *Nature Physics*, 10(9):631–633, July 2014. ISSN 1745-2481.
- [9] Hsin-Yuan Huang, Michael Broughton, Jordan Cotler, Sitan Chen, Jerry Li, Masoud Mohseni, Hartmut Neven, Ryan Babbush, Richard Kueng, John Preskill, and Jarrod R. McClean. Quantum advantage in learning from experiments. *Science*, 376(6598):1182–1186, 2022.
- [10] Shuvro Chowdhury, Kerem Y Camsari, and Supriyo Datta. Emulating Quantum Circuits with Generalized Ising Machines. *IEEE Access*, 2023.
- [11] Matthias Troyer and Uwe-Jens Wiese. Computational Complexity and Fundamental Limitations to Fermionic Quantum Monte Carlo Simulations. *Physical review letters*, 94(17):170201, 2005.
- [12] Shuvro Chowdhury, Kerem Y Camsari, and Supriyo Datta. Accelerated quantum Monte Carlo with probabilistic computers. *Communications Physics*, 6(1):85, 2023.
- [13] Naoya Onizawa and Takahiro Hanyu. Gpu-accelerated simulated annealing based on p-bits with real-world device-variability modeling. *Scientific Reports*, 15(1):6118, Feb 2025. ISSN 2045-2322.
- [14] Benjamin Block, Peter Virnau, and Tobias Preis. Multi-GPU accelerated multi-spin Monte Carlo simulations of the 2D Ising model. *Computer Physics Communications*, 181(9):1549–1556, 2010.
- [15] Tobias Preis, Peter Virnau, Wolfgang Paul, and Johannes J Schneider. GPU accelerated Monte Carlo simulation of the 2D and 3D Ising model. *Journal of Computational Physics*, 228(12):4468–4477, 2009.
- [16] Kun Yang, Yi-Fan Chen, Georgios Roumpos, Chris Colby, and John Anderson. High performance Monte Carlo simulation of Ising model on TPU clusters. In *Proceedings of the International Conference for High Performance Computing, Networking, Storage and Analysis*, pages 1–15, 2019.
- [17] Ye Fang, Sheng Feng, Ka-Ming Tam, Zhifeng Yun, Juana Moreno, Jagannathan Ramanujam, and Mark Jarrell. Parallel tempering simulation of the three-dimensional Edwards–Anderson model with compact asynchronous multispin coding on GPU. *Computer Physics Communications*, 185(10):2467–2478, 2014.
- [18] Shaila Niazi, Shuvro Chowdhury, Navid Anjum Aadit, Masoud Mohseni, Yao Qin, and Kerem Y. Camsari. Training deep Boltzmann networks with sparse Ising machines. *Nature Electronics*, 7(7):610–619, Jul 2024. ISSN 2520-1131.
- [19] Srijan Nikhar, Sidharth Kannan, Navid Anjum Aadit, Shuvro Chowdhury, and Kerem Y Camsari. All-to-all reconfigurability with sparse and higher-order Ising machines. *Nature Communications*, 15(1):8977, 2024.
- [20] Navid Anjum Aadit, Masoud Mohseni, and Kerem Y Camsari. Accelerating Adaptive Parallel Tempering with FPGA-based p-bits. In *2023 IEEE Symposium on VLSI Technology and Circuits (VLSI Technology and Circuits)*, pages 1–2. IEEE, 2023.
- [21] Navid Anjum Aadit, Andrea Grimaldi, Mario Carpentieri, Luke Theogarajan, Giovanni Finocchio, and Kerem Y Camsari. Computing with invertible logic: Combinatorial optimization with probabilistic bits. In *2021 IEEE International Electron Devices Meeting (IEDM)*, pages 40–3. IEEE, 2021.
- [22] Navid Anjum Aadit, Andrea Grimaldi, Giovanni Finocchio, and Kerem Y Camsari. Physics-inspired ising computing with ring oscillator activated p-bits. In *2022 IEEE 22nd International Conference on Nanotechnology (NANO)*, pages 393–396. IEEE, 2022.
- [23] Nihal Sanjay Singh, Keito Kobayashi, Qixuan Cao, Kemal Selcuk, Tianrui Hu, Shaila Niazi, Navid Anjum Aadit, Shun Kanai, Hideo Ohno, Shunsuke Fukami, et al. CMOS plus stochastic nanomagnets enabling heterogeneous computers for probabilistic inference and learning. *Nature Communications*, 15(1):2685, 2024.
- [24] Andrea Grimaldi, Kemal Selcuk, Navid Anjum Aadit, Keito Kobayashi, Qixuan Cao, Shuvro Chowdhury, Giovanni Finocchio, Shun Kanai, Hideo Ohno, Shunsuke Fukami, et al. Experimental evaluation of simulated quantum annealing with MTJ-augmented p-bits. In *2022 International Electron Devices Meeting (IEDM)*, pages 22–4. IEEE, 2022.
- [25] Nihal Sanjay Singh, Shaila Niazi, Shuvro Chowdhury, Kemal Selcuk, Haruna Kaneko, Keito Kobayashi, Shun Kanai, Hideo Ohno, Shunsuke Fukami, and Kerem Y Camsari. Hardware Demonstration of Feedforward Stochastic Neural Networks with Fast MTJ-based p-bits. In *2023 International Electron Devices Meeting (IEDM)*, pages 1–4. IEEE, 2023.
- [26] William A Borders, Ahmed Z Pervaiz, Shunsuke Fukami, Kerem Y Camsari, Hideo Ohno, and Supriyo Datta. Integer factorization using stochastic magnetic tunnel junctions. *Nature*, 573(7774):390–393, 2019.
- [27] Andrew D King, Jack Raymond, Trevor Lanting, Sergei V Isakov, Masoud Mohseni, Gabriel Poulin-Lamarre, Sara Ejtemaee, William Bernoudy, Isil Ozfidan, Anatoly Yu Smirnov, et al. Scaling advantage over path-integral Monte Carlo in quantum simulation of geometrically frustrated magnets. *Nature communications*, 12(1):1113, 2021.
- [28] Tameem Albash and Daniel A Lidar. Demonstration of a scaling advantage for a quantum annealer over simulated annealing. *Physical Review X*, 8(3):031016, 2018.
- [29] Vasil S Denchev, Sergio Boixo, Sergei V Isakov, Nan Ding, Ryan Babbush, Vadim Smelyanskiy, John Martinis, and Hartmut Neven. What is the computational value of finite-range tunneling? *Physical Review X*, 6(3):031015, 2016.
- [30] Itay Hen, Joshua Job, Tameem Albash, Troels F Rønnow, Matthias Troyer, and Daniel A Lidar. Probing for quantum speedup in spin-glass problems with planted solutions. *Physical Review A*, 92(4):042325, 2015.
- [31] Massimo Bernaschi, Isidoro González-Adalid Pemartín, Víctor Martín-Mayor, and Giorgio Parisi. The quantum transition of the two-dimensional Ising spin glass. *Nature*, Jul 2024. ISSN 1476-4687.
- [32] Andrew D. King, Jack Raymond, Trevor Lanting, Richard Harris, Alex Zucca, Fabio Altomare, Andrew J. Berkley, Kelly Boothby, Sara Ejtemaee, Colin Enderud, Emile Hoskinson, Shuiyuan Huang, Eric Ladizinsky, Allison J. R. MacDonald, Gaelen Marsden, Reza Molavi, Travis Oh, Gabriel Poulin-Lamarre, Mauricio Reis, Chris Rich, Yuki Sato, Nicholas Tsai, Mark Volkmann, Jed D. Whittaker, Jason Yao, Anders W. Sandvik, and Mohammad H. Amin. Quantum critical dynamics in a 5,000-qubit programmable spin glass. *Nature*, 617(7959):61–66, April 2023. ISSN 1476-4687.

- [33] Zheng Zhu, Andrew J Ochoa, and Helmut G Katzgraber. Efficient cluster algorithm for spin glasses in any space dimension. *Physical review letters*, 115(7):077201, 2015.
- [34] Changjun Fan, Mutian Shen, Zohar Nussinov, Zhong Liu, Yizhou Sun, and Yang-Yu Liu. Searching for spin glass ground states through deep reinforcement learning. *Nature communications*, 14(1):725, 2023.
- [35] Kyle Lee, Shuvro Chowdhury, and Kerem Y. Camsari. Noise-augmented chaotic ising machines for combinatorial optimization and sampling. *Communications Physics*, 8(1):35, Jan 2025. ISSN 2399-3650.
- [36] J. Houdayer. A cluster Monte Carlo algorithm for 2-dimensional spin glasses. *The European Physical Journal B*, 22(4):479–484, August 2001. ISSN 1434-6028.
- [37] Zheng Zhu, Andrew J. Ochoa, and Helmut G. Katzgraber. Efficient Cluster Algorithm for Spin Glasses in Any Space Dimension. *Physical Review Letters*, 115(7):077201, August 2015. ISSN 1079-7114.
- [38] James King, Masoud Mohseni, William Bernoudy, Alexandre Fréchet, Hossein Sadeghi, Sergei V Isakov, Hartmut Neven, and Mohammad H Amin. Quantum-assisted genetic algorithm. *arXiv preprint arXiv:1907.00707*, 2019.
- [39] Masuo Suzuki. Relationship between d-Dimensional Quantal Spin Systems and (d+1)-Dimensional Ising Systems Equivalence, Critical Exponents and Systematic Approximants of the Partition Function and Spin Correlations. *Progress of Theoretical Physics*, 56:1454–1469, 1976.
- [40] Kerem Y. Camsari, Shuvro Chowdhury, and Supriyo Datta. Scalable Emulation of Sign-Problem-Free Hamiltonians with Room-Temperature p -bits. *Phys. Rev. Applied*, 12:034061, 09 2019.
- [41] Bettina Heim, Troels F. Rønnow, Sergei V. Isakov, and Matthias Troyer. Quantum versus classical annealing of Ising spin glasses. *Science*, 348(6231):215–217, 2015.
- [42] Giuseppe E. Santoro, Roman Martoňák, Erio Tosatti, and Roberto Car. Theory of Quantum Annealing of an Ising Spin Glass. *Science*, 295(5564):2427–2430, 2002.
- [43] A Billoire, L A Fernandez, A Maiorano, E Marinari, V Martin-Mayor, J Moreno-Gordo, G Parisi, F Ricci-Tersenghi, and J J Ruiz-Lorenzo. Dynamic variational study of chaos: spin glasses in three dimensions. *Journal of Statistical Mechanics: Theory and Experiment*, 2018(3):033302, mar 2018.
- [44] T Papakonstantinou and A Malakis. Parallel tempering and 3D spin glass models. *Journal of Physics: Conference Series*, 487(1):012010, mar 2014.
- [45] David J Earl and Michael W Deem. Parallel tempering: Theory, applications, and new perspectives. *Physical Chemistry Chemical Physics*, 7(23):3910–3916, 2005.
- [46] Robert H Swendsen and Jian-Sheng Wang. Replica Monte Carlo simulation of spin-glasses. *Physical review letters*, 57(21):2607, 1986.
- [47] Koji Hukushima and Koji Nemoto. Exchange Monte Carlo method and application to spin glass simulations. *Journal of the Physical Society of Japan*, 65(6):1604–1608, 1996.
- [48] Andrea Grimaldi, Luis Sánchez-Tejerina, Navid Anjum Aadit, Stefano Chiappini, Mario Carpentieri, Kerem Camsari, and Giovanni Finocchio. Spintronics-compatible approach to solving maximum-satisfiability problems with probabilistic computing, invertible logic, and parallel tempering. *Phys. Rev. Appl.*, 17:024052, Feb 2022.
- [49] S.V. Isakov, I.N. Zintchenko, T.F. Rønnow, and M. Troyer. Optimised simulated annealing for Ising spin glasses. *Computer Physics Communications*, 192:265–271, 2015. ISSN 0010-4655.
- [50] Helmut G Katzgraber, Simon Trebst, David A Huse, and Matthias Troyer. Feedback-optimized parallel tempering monte carlo. *Journal of Statistical Mechanics: Theory and Experiment*, 2006(03):P03018, 2006.
- [51] Masoud Mohseni, Daniel Eppens, Johan Strumpfer, Raffaele Marino, Vasil Denchev, Alan K. Ho, Sergei V. Isakov, Sergio Boixo, Federico Ricci-Tersenghi, and Hartmut Neven. Nonequilibrium Monte Carlo for Unfreezing Variables in Hard Combinatorial Optimization. *arXiv*, nov 2021.
- [52] Keisuke Hayakawa, Shun Kanai, Takuya Funatsu, Junta Igarashi, Butsurin Jinnai, WA Borders, H Ohno, and S Fukami. Nanosecond random telegraph noise in in-plane magnetic tunnel junctions. *Physical review letters*, 126(11):117202, 2021.
- [53] Lucile Soumah, Louise Desplat, Nhat-Tan Phan, Ahmed Sidi El Valli, Advait Madhavan, Florian Disdier, Stéphane Auffret, Ricardo Sousa, Ursula Ebels, and Philippe Talatchian. Entropy-Assisted Nanosecond stochastic operation in perpendicular superparamagnetic tunnel junctions. *arXiv preprint arXiv:2402.03452*, 2024.
- [54] Firas Hamze, Darryl C Jacob, Andrew J Ochoa, Dilina Perera, Wenlong Wang, and Helmut G Katzgraber. From near to eternity: spin-glass planting, tiling puzzles, and constraint-satisfaction problems. *Physical Review E*, 97(4):043303, 2018.
- [55] M. Baity-Jesi, R.A. Baños, A. Cruz, L.A. Fernandez, J.M. Gil-Narvion, A. Gordillo-Guerrero, D. Iñiguez, A. Maiorano, F. Mantovani, E. Marinari, V. Martin-Mayor, J. Monforte-Garcia, A. Muñoz Sudupe, D. Navarro, G. Parisi, S. Perez-Gaviro, M. Pivanti, F. Ricci-Tersenghi, J.J. Ruiz-Lorenzo, S.F. Schifano, B. Seoane, A. Tarancon, R. Tripiccione, and D. Yllanes. Janus II: A new generation application-driven computer for spin-system simulations. *Computer Physics Communications*, 185(2):550–559, 2014. ISSN 0010-4655.
- [56] Brian Sutton, Rafatul Faria, Lakshmi Anirudh Ghantasala, Risi Jaiswal, Kerem Yunus Camsari, and Supriyo Datta. Autonomous probabilistic coprocessing with petaflips per second. *IEEE Access*, 8:157238–157252, 2020.
- [57] Devrath Iyer and Sara Achour. Efficient optimization with encoded ising models. In *2025 IEEE International Symposium on High Performance Computer Architecture (HPCA)*, pages 85–98. IEEE, 2025.
- [58] Lawrence T. Clark, Vinay Vashishtha, Lucian Shifren, Aditya Gujja, Saurabh Sinha, Brian Cline, Chandarasekaran Ramamurthy, and Greg Yeric. ASAP7: A 7-nm finFET predictive process design kit. *Microelectronics Journal*, 53: 105–115, 2016. ISSN 1879-2391.
- [59] Kerem Yunus Camsari, Rafatul Faria, Brian M Sutton, and Supriyo Datta. Stochastic p -bits for invertible logic. *Physical Review X*, 7(3):031014, 2017.
- [60] Daniel Brélaz. New methods to color the vertices of a graph. *Communications of the ACM*, 22(4):251–256, apr 1979.
- [61] O. Melchert. autoscale.py - a program for automatic finite-size scaling analyses: A user’s guide. *arXiv preprint arXiv:0910.5403*, 2009.
- [62] H. Rieger and N. Kawashima. Application of a continuous time cluster algorithm to the two-dimensional random quantum Ising ferromagnet. *The European Physical Journal B - Condensed Matter and Complex Systems*, 9(2):233–236, May 1999.
- [63] Shuvro Chowdhury, Andrea Grimaldi, Navid Anjum Aadit, Shaila Niazi, Masoud Mohseni, Shun Kanai, Hideo Ohno, Shunsuke Fukami, Luke Theogarajan, Giovanni Finocchio, Supriyo Datta, and Kerem Y. Camsari. A Full-Stack View of Probabilistic Computing With p -Bits: Devices, Architectures, and Algorithms. *IEEE Journal on Exploratory Solid-State*

Computational Devices and Circuits, 9(1):1–11, 2023.

- [64] M. Mahmudul Hasan Sajeeb, Navid Anjum Aadit, Shuvro Chowdhury, Tong Wu, Cesely Smith, Dhruv Chinmay, Atharva Raut, Kerem Y. Camsari, Corentin Delacour, and Tathagata Srimani. Scalable connectivity for ising machines: Dense to sparse. *Phys. Rev. Appl.*, 24:014005, Jul 2025.
- [65] Troels F. Rønnow, Zhihui Wang, Joshua Job, Sergio Boixo, Sergei V. Isakov, David Wecker, John M. Martinis, Daniel A. Lidar, and Matthias Troyer. Defining and detecting quantum speedup. *Science*, 345(6195):420–424, July 2014.
- [66] Andrew D. King, Sei Suzuki, Jack Raymond, Alex Zucca, Trevor Lanting, Fabio Altomare, Andrew J. Berkley, Sara Ejtemaee, Emile Hoskinson, Shuiyuan Huang, Eric Ladizinsky, Allison J. R. MacDonald, Gaelen Marsden, Travis Oh, Gabriel Poulin-Lamarre, Mauricio Reis, Chris Rich, Yuki Sato, Jed D. Whittaker, Jason Yao, Richard Harris, Daniel A. Lidar, Hidetoshi Nishimori, and Mohammad H. Amin. Coherent quantum annealing in a programmable 2,000 qubit Ising chain. *Nature Physics*, 18(11):1324–1328, November 2022. ISSN 1745-2481. Number: 11 Publisher: Nature Publishing Group.
- [67] Alex Carsello, James Thomas, Ankita Nayak, Po-Han Chen, Mark Horowitz, Priyanka Raina, and Christopher Torng. mflowgen: a modular flow generator and ecosystem for community-driven physical design: invited. In *Proceedings of the 59th ACM/IEEE Design Automation Conference, DAC '22*, page 1339–1342, New York, NY, USA, 2022. Association for Computing Machinery. ISBN 9781450391429.

Supplementary Information

Pushing the Boundary of Quantum Advantage in Hard Combinatorial Optimization with Probabilistic Computers

Shuvro Chowdhury, Navid Anjum Aadit, Andrea Grimaldi, Eleonora Raimondo, Atharva Raut, P. Aaron Lott, Johan H. Mentink, Marek M. Rams, Federico Ricci-Tersenghi, Massimo Chiappini, Luke S. Theogarajan, Tathagata Srimani, Giovanni Finocchio, Masoud Mohseni and Kerem Y. Camsari

I. Discrete-time Simulated Quantum Annealing (DT-SQA) algorithm

The DT-SQA algorithm used in Fig. 2 of the main text relies on the Suzuki-Trotter approximation, where the partition function, Z_Q of a quantum Hamiltonian,

$$H_Q = - \sum_{i < j} J_{ij} \sigma_i^z \sigma_j^z - \Gamma_x \sum_i \sigma_i^x \quad (\text{S.1})$$

with σ_i^α ($\alpha \in \{x, y, z\}$) being Pauli spin matrix at site i , is approximated by the partition function, Z_C of a classical Hamiltonian, H_C as

$$Z_Q = \text{tr} [\exp(-\beta H_Q)] = \lim_{R \rightarrow \infty} Z_C = \lim_{R \rightarrow \infty} \exp(-\beta H_C) \quad (\text{S.2})$$

where H_C , corresponding to H_Q , is defined as

$$H_C = - \sum_{k=1}^R \left[\sum_{i < j} J_{\parallel, ij} \sigma_{i,k} \sigma_{j,k} + \sum_i J_{\perp, ik} \sigma_{i,k} \sigma_{i,k+1} \right] \text{ with } J_{\parallel, ij} = \frac{J_{ij}}{R} \text{ and } J_{\perp} = -\frac{1}{2\beta} \ln \left[\tanh \left(\frac{\beta \Gamma_x}{R} \right) \right]. \quad (\text{S.3})$$

where $\sigma_{i,j}$ (without any superscript) denotes Ising spin ($\sigma_{i,j} \in \{-1, +1\}$) at site i of replica j . Thus, a d -dimensional quantum Hamiltonian can be theoretically mapped to a $(d+1)$ -dimensional classical Hamiltonian, where the extra dimension corresponds to the replica dimension. When R approaches infinity, the partition function corresponding to the quantum Hamiltonian is exactly reproduced by the partition function of the mapped classical counterpart. However, for practical purposes or when DT-SQA is used as a classical algorithm (for example, in optimization), a finite number of replicas typically in the range of 10 to 100—may be preferred [40–42]. A pseudocode outlining our implementation of the DT-SQA algorithm is presented in Algorithm S1. As described in the Methods section, graph coloring is used to partition the spin system into independent sets that can be updated in parallel. Each color corresponds to one of these independent sets. Algorithm S1 implements this update scheme by looping over colors.

There is also a continuous-time version of the DT-SQA algorithm (CT-SQA) [62]. However, CT-SQA lacks the straightforward hardware implementation offered by DT-SQA. Although both algorithms emulate the equilibrium statistics of the quantum system, differences arise when comparing their transient dynamics to those of a quantum annealer.

A. Residual energy as a function of total Monte Carlo effort

In the main text, we analyze the scaling behavior of the residual energy as a function of Monte Carlo sweeps for three distinct Discrete-Time Simulated Quantum Annealing (DT-SQA) systems, each characterized by a different number of Trotter slices. We define a Monte Carlo sweep as the process of updating every spin in the system exactly once. Consequently, larger systems naturally involve more computational effort, as they necessitate a greater number of probabilistic spin flips per sweep.

This definition aligns intuitively with our hardware-centric context. We envision the different systems, each comprising R replicas, as separate black-box computational units or stand-alone chips. As previously detailed in Refs. [4, 19, 63, 64], our graph-colored architecture is designed to mimic a truly asynchronous analog probabilistic annealer (one example being a stochastic Magnetic Tunnel Junction based probabilistic computer [26]) and as such it enables updates across the entire network in constant time.

This uniform update time primarily depends on the topology of the network rather than system size. Specifically, if a given graph has degree scaling as $\mathcal{O}(k)$, where k denotes the number of neighbors per node (as in the case of 3D spin glasses considered in this work), the clock frequency that sets the update frequency is mainly determined by the time required to compute the local

field as given by Eq. (5), a calculation independent of the overall network size, as we demonstrate in Supplementary Fig. S11 and Supplementary Table S1.

Our use of Monte Carlo sweeps for systems of different sizes is different from how Monte Carlo effort is defined in CPU-centric comparisons [65]. As carefully discussed in Ref. [65], an analog quantum annealer acquires an $\mathcal{O}(N)$ speedup (similar to our asynchronous architecture) due to its linearly growing resources with problem size and this parallel speedup must be separated from intrinsically quantum speedups. Given that our systems also benefit from the same speedup, our definitional choice for MCS is appropriate.

Nevertheless, it is crucial to note that our scaling argument against quantum annealing does not hinge upon any MCS definition, since the residual energy comparisons are made on a fixed size, hence N does not vary. As demonstrated in Fig. S1, the residual energy exhibits a clear power-law dependence on annealing time or Monte Carlo sweeps. Thus, multiplying the annealing duration by a constant factor (in this case, the number of replicas R) does not alter the fundamental slope of the scaling:

$$\rho_E^f(Rt_a) \propto (Rt_a)^{-\kappa_f} = (\text{constant}) t_a^{-\kappa_f}. \quad (\text{S.4})$$

B. Performance of DT-SQA algorithm on embedded instances

In Fig. 2 of the main text, we presented scaling results for logical instances using the DT-SQA algorithm. Fig. S2 shows the scaling results from DT-SQA experiments on embedded instances. As in logical instances, the slope increases gradually with the number of replicas for the embedded instances, however, achieving the same slope as the quantum annealer requires more replicas compared to logical instances. Also, the embedded instances as provided require four colors for graph coloring (due to the embedding requirements). An even number of replicas are used so that the replicated networks of the embedded instances can also be colored with four colors.

C. Cube size and Trotter replica dependence of DT-SQA

Fig. S3 illustrates the dependence of the final residual energy on annealing time (t_a) for different cube sizes (L) of logical instances. The slopes (absolute value) of the plots decrease as L increases, but all plots eventually reach a flat plateau region. This behavior is consistently observed across different numbers of Trotter replicas (R) as shown in the figure. The residual energy at which the plateau occurs decreases with an increasing number of Trotter replicas, also observed in [41].

Algorithm S1: Discrete-time simulated quantum annealing algorithm with p-computers

Input: Weights, biases, number of replicas, number of sweeps, colormap, annealing schedule, temperature

Output: State corresponding to minimum energy, m_{opt}

- 1 **Function** p-computer (*weights, biases, colormap, temp.*):
 - 2 **for** each color in the colormap **do**
 - 3 **for** each p-bit in the color **do**
 - 4 solve Eq. (5) and Eq. (6).
 - 5 Divide weights by number of replicas.
 - 6 Generate all replicas as indicated in number of replicas.
 - 7 Insert transverse couplings between the replicas.
 - 8 Perform graph coloring for the replicated network given the colormap of a single replica.
 - 9 Initialize all spins in all replicas to random states.
 - 10 **for** each sweep until number of sweeps is reached **do**
 - 11 Get the transverse field value from the annealing schedule.
 - 12 Update the transverse coupling using the transverse field.
 - 13 Sample p-bit states from p-computer.
 - 14 Compute energy of each replica.
 - 15 **return** p-bit states for the replica with the minimum energy.
-

Fig. S1. Slope of the residual energy, ρ_E^f as a function of total Monte Carlo effort: The residual energy is re-plotted against the total Monte Carlo effort for the logical instances of size $15 \times 15 \times 12$. Here total Monte Carlo effort is defined as the total number of attempted spin flips during the whole annealing process. We emphasize that doing so only changes the prefactor and does not change the slope of the plots because of the power-law nature since multiplying time with any constant factor does not change the slope. We also emphasize that the Total Monte Carlo effort is a fixed-size CPU-centric measure, as discussed in Ref. [65], not appropriate in our context, as we discuss in the text.

Fig. S2. Replica scaling of the DT-SQA on embedded instances: (a) Residual energy, ρ_E^f , plotted as a function of the annealing time, t_a , for embedded instances of size $15 \times 15 \times 12 \times 2$ (the latter 2 represents the number of physical qubits used to represent a logical spin), for three distinct values of R . In these instances, each lattice point consists of a pair of spins strongly coupled by a ferromagnetic interaction with an absolute strength of 2. (b) The DT-SQA curve with $R = 2850$ in (a) is compared to the slope obtained from a quantum annealer, showing nearly identical slope as reported in Ref. [32]. (c) The measured slopes (κ_f) from DT-SQA simulations are plotted against the number of replicas, R . The red dashed line serves as a visual guide, while the gray dotted-dashed line represents the slope derived from the QA. The slope for the quantum annealer can be matched by using more than 2850 replicas.

II. Analysis of DT-SQA scaling using extreme value theory

In this section, we explain the increase in DT-SQA slopes with increasing Trotter replicas using extreme value theory (EVT). Recall that in our DT-SQA simulations, we have R interconnected Trotter replicas and we select the replica with the best (minimum) energy. The interconnection of Trotter replicas complicates the direct application of EVT. Therefore, we first describe the conventional EVT, followed by the modified EVT for interconnected Trotter replicas.

Conventional EVT: A straightforward application of EVT can be demonstrated with the following experiment: we run P independent copies of DT-SQA simulations, each consisting of R Trotter replicas. From each of these P independent copies, we select the minimum energy replica and then choose the minimum among these P minimum energies. Since these P minimum energies are independent and identically distributed, their distribution is approximately a Gaussian distribution, particularly at low MCS as shown in Supplementary Fig. S4. At high MCS, the distributions become skewed to the left, due to the hard constraint imposed by the ground state which serves as a lower bound.

For the derivations that follow, we assume a Gaussian distribution for the energies. Let E_1, E_2, \dots, E_P represent the energies of P independent and identically distributed (i.i.d.) Gaussian random variables, each characterized by a mean μ and standard

Fig. S3. Cube size and replica dependence of DT-SQA: Residual energy (ρ_E^f) is plotted as a function of annealing time (t_a) for various cube sizes (L) and three different number of Trotter replicas (R). Each data point represents an average over 300 problem instances, with each instance averaged over 50 independent runs. Error bars represent the 95% bootstrap confidence interval of the mean across 300 spin-glass instances.

Fig. S4. Gaussian distribution of minimum energy values: The distribution of minimum energies for a randomly chosen spin-glass instance is shown at various MCS values, based on 5000 independent runs. In each run, DT-SQA is performed with 32 Trotter replicas for annealing at a fixed MCS, and the best energy replica is selected at the end of the annealing process. The distributions closely follow a Gaussian shape for smaller MCS values, as indicated by the dashed lines representing Gaussian fits. At higher MCS values, the distributions become increasingly skewed to the left due to the hard constraint imposed by the ground state energy.

deviation σ . Our objective is to derive the expected value of the minimum of these P runs, $\mathbb{E}(\min(E_1, E_2, \dots, E_P))$ which plays a crucial role in improving the residual energy and, consequently the slope in DT-SQA simulations. The probability density function (PDF) of a Gaussian random variable E_i is given by:

$$f(x) = \frac{1}{\sqrt{2\pi}\sigma} \exp\left(-\frac{(x-\mu)^2}{2\sigma^2}\right) \quad (\text{S.5})$$

The cumulative distribution function (CDF) of a random variable E represents the probability that E takes a value less than or equal to a specific value x . Mathematically, the CDF, $F(x)$, is defined as:

$$F(x) = \Pr(E \leq x) \quad (\text{S.6})$$

For a continuous random variable, the CDF can be obtained by integrating its probability density function (PDF) $f(x)$:

$$F(x) = \int_{-\infty}^x f(x) dx \quad (\text{S.7})$$

The CDF satisfies the property that

$$\lim_{x \rightarrow \infty} F(x) = 1 \quad (\text{S.8})$$

which ensures that the total probability adds to 1. For a Gaussian random variable, the cumulative distribution function (CDF) is given by:

$$F(x) = \frac{1}{\sqrt{2\pi}\sigma} \int_{-\infty}^x \exp\left(-\frac{(x-\mu)^2}{2\sigma^2}\right) dx = \frac{1}{2} \left[1 + \operatorname{erf}\left(\frac{x-\mu}{\sqrt{2}\sigma}\right)\right] \quad (\text{S.9})$$

where $\operatorname{erf}(\cdot)$ is the error function.

The CDF allows us to compute probabilities over intervals and is particularly useful to find the probability that the minimum of P variables falls below a given threshold. To calculate the probability that the minimum of these P random variables E_i is less than or equal to a certain threshold x , i.e., $\Pr(\min(E_1, E_2, \dots, E_P) \leq x)$, we start by noting that for a single random variable E_i , $\Pr(\min(E_i) \leq x) = \Pr(E_i \leq x)$ and is given by its CDF. For two variables, E_1 and E_2 , the probability that their minimum is less than or equal to x can be computed using their complement probabilities, similar to the Bernoulli case. Specifically,

$$\Pr(\min(E_1, E_2) \leq x) = 1 - \Pr(\min(E_1, E_2) > x) \quad (\text{S.10})$$

But $\min(E_1, E_2) > x$ implies that both $E_1, E_2 > x$ and therefore,

$$\Pr(\min(E_1, E_2) \leq x) = 1 - [\Pr(E_1 > x) \Pr(E_2 > x)] \quad (\text{S.11})$$

Using the property $\Pr(E_i > x) = 1 - \Pr(E_i \leq x) = 1 - F(x)$, this can be rewritten as:

$$\Pr(\min(E_1, E_2) \leq x) = 1 - \{[1 - \Pr(E_1 \leq x)][1 - \Pr(E_2 \leq x)]\} = 1 - [1 - F(x)]^2 \quad (\text{S.12})$$

Generalizing this result to P independent random variables, we obtain:

$$\Pr(\min(E_1, E_2, \dots, E_P) \leq x) = F_P(x) = 1 - [1 - F(x)]^P = 1 - 2^{-P} \left[\operatorname{erfc}\left(\frac{x-\mu}{\sqrt{2}\sigma}\right)\right]^P \quad (\text{S.13})$$

where $\operatorname{erfc}(\cdot) = 1 - \operatorname{erf}(\cdot)$ is the complementary error function.

Finally, the expected value of the minimum of these P random variables can be found by calculating the mean of the PDF associated with this CDF. Specifically, the expected value is derived as:

$$\mathbb{E}(\min(E_1, E_2, \dots, E_P)) = \int_{-\infty}^{+\infty} x \frac{dF_P(x)}{dx} dx = \frac{2^{0.5-P} P}{\sqrt{\pi}\sigma} \int_{-\infty}^{+\infty} x \exp\left[-(x-\mu)^2/2\sigma^2\right] \left[\operatorname{erfc}\left(\frac{x-\mu}{\sqrt{2}\sigma}\right)\right]^{P-1} dx \quad (\text{S.14})$$

While Eq. (S.14) formally expresses the expected value of the minimum energy, it lacks a simple closed-form solution for $P > 5$ due to the complexity of integrating terms involving the complementary error function. However, we can obtain an excellent approximation by focusing on the median instead of the mean, leveraging the fact that for symmetric distributions like the Gaussian, the mean and median are close in value.

So, we proceed to solve for the median x_p such that $F_P(x_p) = 0.5$:

$$F_P(x_p) = 0.5 = 1 - 2^{-P} \left[\operatorname{erfc}\left(\frac{x_p - \mu}{\sqrt{2}\sigma}\right)\right]^P \quad (\text{S.15})$$

which leads to the following solution:

$$x_p = \mu + \sqrt{2}\sigma \operatorname{erfc}^{-1}\left(2^{\frac{P-1}{P}}\right). \quad (\text{S.16})$$

which is an exact expression that finds the median. However, this expression also does not reveal the relationship between x_p and P explicitly. In order to get a more revealing expression, we find an asymptotic expansion of x_p which leads to:

$$x_p \approx \mu - \sigma \sqrt{\ln \frac{2}{\pi(\ln 4)^2} + \frac{\ln 2}{P} + 2 \ln P - \ln \left[\ln \frac{2}{\pi(\ln 4)^2} + \frac{\ln 2}{P} + 2 \ln P \right]} \quad (\text{S.17})$$

when $P \rightarrow \infty$,

$$x_p \approx \mu - \sigma \sqrt{2 \ln P} \quad (\text{S.18})$$

which is a textbook result from EVT. Thus, as P increases, the minimum energy decreases as $\sqrt{\ln P}$, which reflects how increasing the number of replicas enhances the system's ability to find lower-energy states.

The functional form of the final residual energy, ρ_E^f can be obtained from the Supplementary Eq. (S.16) and the definition of ρ_E^f from the main text:

$$\rho_E^f(t) = a(t) + \sqrt{2} b(t) \operatorname{erfc}^{-1}\left(2 \frac{P-1}{P}\right). \quad (\text{S.19})$$

where $a(t)$ and $b(t)$ are two time-dependent parameters which are related to the average mean and average standard deviation of distributions of residual energies from individual runs at a fixed MCS. To validate our theory, we conduct P independent DT-SQA simulations, each with 32 Trotter replicas, after which we select the best solution across all P runs. The results of this experiment are shown in Fig. S5(a-c). We observe that with approximately $P \approx 50$ repetitions, the slope achieved matches the slope reported in [66].

Fig. S5. Enhanced scaling using independent runs of Trotter replicas in DT-SQA: (a) Residual energy (ρ_E^f) as a function of annealing time (t_a) is shown for P independent DT-SQA simulations, each with 32 Trotter replicas. The best solution from these P experiments is selected. R_T is defined as the total number of Trotter replicas, and is equal to $R_T = 32P$. Results are shown for ($P = 1, R = 32$) and ($P = 50, R = 32$). Error bars denote the 95% bootstrap confidence interval of the mean across spin-glass instances. (b) Comparison of DT-SQA results with $R_T = 1600$ to those of the quantum annealer (QA). With $P \approx 50$, DT-SQA achieves a slope comparable to the slope reported for QA in [66]. (c) Slope (κ_f) as a function of R_T . The red dashed line represents predictions from extreme value theory (EVT), while the gray dotted-dashed line corresponds to the slope from QA. Unlike the original DT-SQA algorithm, where all Trotter replicas are interconnected, the EVT-based approach achieves better scaling with fewer Trotter replicas. Error bars denote 95% confidence interval of fitting. The total number of replicas R_T should be compared with the total number of replicas R in Fig. 2 in main text. Fig. 2 uses only $P = 1$ which implies $R_T = R$, for simplicity we did not use R_T there.

For each MCS value, the mean (μ) and standard deviation (σ) for each instance are obtained by fitting the distribution of the sampled best energies from multiple DT-SQA simulations (each simulation contains 32 Trotter replicas, and only the best energy replica is selected). The averages of these means and standard deviations are then computed across all 300 instances annealed at the same MCS. These averages are converted into residual energies using Eq. (2) in the main text. Residual energy predictions are derived from these mean and standard deviation values using Supplementary Eq. (S.19), as shown in Fig. S5. We find excellent agreement between experimental results and predictions from conventional EVT as shown in Supplementary Fig. S5(c).

Modified EVT: The extreme value theory (EVT) predictions for Fig. 2(c) are more involved and not exact. Two technical challenges arise: (1) all Trotter replicas are interconnected, so the block size is not known a priori, unlike the conventional EVT approach and (2) the block size may depend on the MCS values through Γ_x and J_\perp . The latter stems from the fact that the correlation length along the replica direction varies with MCS, as shown in Fig. S6. Despite these challenges, we apply EVT for an approximate understanding. We formulate a self-consistent approach as follows: we start by guessing a block size and partitioning the total number of Trotter replicas accordingly. For each partition, the best energy among the Trotter replicas within that partition is determined. As before, the mean and standard deviation of these best energy values are computed for each instance at a given MCS and averaged over all 300 instances and multiple runs per instance. Using these average mean and standard deviation values, we predict the residual energy with Supplementary Eq. (S.19). This predicted residual energy is then compared with the actual residual energy computed using Eq. (2) in the main text. The procedure is repeated until a block size is found where the prediction and actual residual energies closely match, as shown in Supplementary Fig. S7. For low MCS values, the block sizes determined from this approach approximately align with the correlation lengths obtained independently

from simulations (see Supplementary Fig. S6). This alignment provides a justification for our modified EVT approach. At higher MCS values, however, the distributions deviate from a Gaussian shape, becoming increasingly skewed to the left. Consequently, deviations from Supplementary Eq. (S.19) become more pronounced.

Comparison between conventional and modified EVT: Despite the seeming similarities, the independent (conventional) and interconnected (modified) EVT approaches are not equivalent. Supplementary Fig. S8 shows that the conventional approach deviates from the power-law behavior, encountering an early flat plateau around 1000 MCS. This behavior aligns well with the correlation trends shown in Supplementary Fig. S6, where at around 1000 MCS the replica-to-replica correlation length exceeds the system size ($R = 32$) for the conventional EVT.

Fig. S6. Decay of correlation length along the replica direction: The average correlation ($C_r = (1/N) \sum_N \sigma_{i,0} \sigma_{i,r}$, $C_r = (1/n) \sum_n \sigma_{i,0} \sigma_{i,r}$) between the replicas as a function of the distance (r) from the first replica ($r = 0$) is shown for the DT-SQA algorithm, with various total number of Trotter replicas R and annealing times (t_a , in MCS units). We measure C_r from the first replica, however, due to the periodic boundary conditions along the replica direction, C_r is measured to be invariant across all replicas. The correlation length along the replica direction shows a strong dependence on the annealing times (t_a) with longer annealing times leading to broader correlation peaks. However, the dependence on the number of Trotter replicas (R) is minimal, as indicated by the nearly overlapping dashed and solid lines for different R values.

Fig. S7. Comparison of predictions from the modified EVT with experimentally obtained residual energies: (a) The block size extraction procedure based on extreme value theory (EVT) is illustrated for $t_a = 100$ MCS. Various block sizes are assumed, and the corresponding residual energies are predicted. The block size that yields a prediction closest to the experimental residual energy is selected. (b) The procedure in (a) is repeated for different MCS values (such as $t_a = 50, 100$ and 200 MCS). The chosen block sizes for each MCS value are shown alongside the number of blocks (P). The predictions from the modified EVT align closely with the experimental residual energies. These results are in approximate agreement with the correlation lengths shown in Supplementary Fig. S6.

Fig. S8. Comparison of two DT-SQA approaches used in this work: For logical instances of size $15 \times 15 \times 12$, EVT based approach of DT-SQA ($R = 32$, $P = 50$) is compared against the conventional DT-SQA approach with $R = 2850$, $P = 1$. Both approaches share the same initial slope, which exceeds the quantum annealer’s slope. However, the EVT-based approach reaches a flat plateau at a relatively higher residual energy emphasizing that the two approaches are not identical (see text).

III. Feasibility analysis of DT-SQA

As shown in Fig. 2 of the main text, a large number of replicas ($R = 2850$) is required for the DT-SQA algorithm to match the scaling exponent of the quantum annealer for 3D spin glass problems. We now evaluate the feasibility of having 2850 replicas on a single chip. This way, a hardware implementation of DT-SQA would physically house and update all replicas in parallel, unlike software-based simulations where replicas are updated sequentially on a CPU. The feasibility analysis we present here is equally applicable to the APT algorithm, which achieves performance on par with or exceeding DT-SQA and the quantum annealer while requiring fewer replicas (132 replicas as discussed in the main text) and further improvement in performance is expected with 2850 replicas. To assess the feasibility, we conduct a detailed physical design analysis. For our analysis, we use the open-source mflowgen physical design flow [67] with the open-source ASAP7 7 nm process design kit (PDK) [58]. Custom RTLs are developed to implement the algorithm based on the p-computing architecture shown in Supplementary Fig. S9, incorporating multi-phase clocking to manage timing across the design. Synthesis is carried out using Cadence Genus, followed by floorplanning, power distribution network (PDN) generation, clock tree synthesis (CTS), and final place-and-route (P&R) using Cadence Innovus. Fig. S10 provides examples of designs after placement and routing, with the corresponding design metrics summarized in Table S1. The routed designs are verified to meet timing constraints under a 3-phase clock (to allow provision for using odd number of replicas), with each phase operating at a frequency of 100 MHz. The final designs, completed after placement and routing, demonstrate that: (1) an ASIC implementation is feasible, and (2) the area required for this ASIC scales linearly with problem sizes.

Fig. S11 shows the area scaling of placed and routed designs for different instances of the DT-SQA algorithm. The area scaling follows a linear trend, with the largest design—featuring 13435 p-bits—occupying approximately 1 mm^2 . Extrapolating this scaling to modern chip dimensions, a chip measuring $28.61 \text{ mm} \times 28.61 \text{ mm}$ could accommodate approximately 7.66 million p-bits, corresponding to 2850 replicas of size $15 \times 15 \times 12$, achieving a scaling similar to that of a quantum annealer.

IV. APT Algorithm with Isoenergetic Cluster Move (ICM)

Parallel tempering (PT) and its variants are a standard choice for solving challenging optimization problems such as the 3D spin glass. Hence, we also evaluate the performance of PT. Like DT-SQA, PT is also a replica-based algorithm. For our analysis, we employ an adaptive version of PT (APT) that includes a preprocessing step to determine the temperature schedule and optimize the number of replicas required. To further enhance the performance of APT, we incorporate isoenergetic cluster moves (ICM). A pseudocode detailing the adaptive parallel tempering algorithm, including the temperature schedule preprocessing and ICM, is presented in Algorithm S2. Additional details regarding the exact parameters used can be found in the Methods section of the main text.

TABLE S1. **Details of the physical flow designs:** Detailed information about the physical flow design performed in this study are listed. The design results are based on ASAP7 - a 7 nm finFET predictive process design kit (PDK) with mflowgen. L represents the dimension of the cube and R denotes the number of replicas used.

Number of p-bits	27	64	125	2687	5374	8061	10748	13435	7.66×10^6
Cube dimension, L	3	4	5	15	15	15	15	15	15
Number of replicas, R	1	1	1	1	2	3	4	5	2850
Flow status	routed	routed	routed	routed	routed	routed	routed	routed	projected
Number of standard cells	11303	27653	54685	1.385×10^6	2.978×10^6	4.886×10^6	6.562×10^6	8.199×10^6	4.7×10^9
Timing met (per phase) MHz	100	100	100	100	100	100	100	100	100
Area (mm ²)	0.001581	0.003844	0.007594	0.184317	0.382561	0.634809	0.851805	1.065527	818.34

A. Temperature profile and swap acceptance rate in APT

We apply the preprocessing algorithm individually to each problem instance as detailed in the Methods section. The temperature profiles for all 300 instances of size $15 \times 15 \times 12$ are shown in Fig. S12. The profiles are highly consistent across instances, with only slight deviations in the low-temperature (high β) region. In principle, a temperature profile generated for a randomly selected instance could be applied to all instances without significantly impacting performance. However, in this work, we optimize the temperature profiles for each instance.

Additionally, Fig. S12 includes the standard deviation of the sampled Monte Carlo energies at each iteration (corresponding to each β). The standard deviation decreases monotonically as β increases, confirming that the replica temperatures are chosen appropriately.

Fig. S13 shows the typical swap acceptance rate in the APT algorithm for logical instances of size $15 \times 15 \times 12$. For optimal performance, it is generally recommended that the acceptance rate remains approximately constant. As shown, the acceptance rate stays roughly constant at around 40%, except for the last few replicas. However, since the algorithm selects the replica with the minimum energy at each swap, the lower acceptance rate of the last few replicas does not impact the overall performance of the algorithm.

Fig. S9. p-computing architecture used in the feasibility analysis of the DT-SQA algorithm: (a) The pseudo-asynchronous architecture features a multiplier-accumulator (MAC) unit that implements Eq. (5). Each p-bit unit consists of a linear feedback shift register (LFSR)-based pseudorandom number generator (PRNG), a lookup table for the activation function (tanh), and a comparator to generate a binary output. (b) The clocking unit on the Alveo U250 data center accelerator card generates three phase-shifted clocks (0° , 120° , and 240°) at 45 MHz from a 300 MHz system clock using an on-board mixed-mode clock manager (MMCM). These are used to trigger the PRNGs inside the colored p-bit blocks. (c) The memory unit interfaces with the CPU via Peripheral Component Interconnect Express (PCI) Express. Data transfer between MATLAB and the FPGA is managed through Advanced eXtensible Interface (AXI) interfaces, with BRAMs (block RAMs) allocated for weights, biases, and binary p-bit outputs. Weights and biases have fixed point $s\{6\}\{3\}$ precision where ‘s’ denotes the sign bit and the first and second curly braces represent the integer and fractional parts.

Fig. S10. Physical design flow results for the DT-SQA algorithm at various scales: The feasibility of ASIC implementation for the DT-SQA algorithm is evaluated by running the mflowgen physical design flow with the ASAP7 7 nm PDK on custom RTL designs. Results are shown for different combinations of cube size (L) and Trotter replicas (R).

Fig. S11. Scaling of chip area with the number of p-bits for the DT-SQA algorithm: The growth in chip area is studied as more p-bits are integrated into the chip, following a full place-and-route design process. The observed trend exhibits near-linear growth, with a slope of 1.05. The largest routed designs, based on the 7 nm ASAP7 PDK, are shown in red, while the projected scaling is represented in blue. Extrapolating to a modern chip size of 28.61 × 28.61 mm², it is estimated that such a chip could accommodate approximately 7.66 million p-bits. This corresponds to 2850 replicas of size 15 × 15 × 12, achieving scaling comparable to a quantum annealer with $\kappa_f > 0.785$.

Algorithm S2: Adaptive parallel tempering with p-computers

Input: Weights, biases, number of swaps, sweeps per swap, colormap, step rate α , initial inverse temperature β_0 , energy variance tolerance, number of chains, sweeps per chain

Output: State corresponding to minimum energy, m_{opt}

```

1 Function p-computer (weights, biases, colormap, temp.):
2   for each color in the colormap do
3     for each p-bit in the color do
4       Solve Eq. (5) and Eq. (6).
5 Function ICMop (replica1, replica2):
6   Find the overlap vector between replica 1 and replica 2.
7   Randomly pick one cluster where overlap is  $-1$ .
8   if size of the cluster is greater than half of the total number of spins then
9     Randomly chose one of the two replicas.
10    Flip all the spins of the chosen replica.
11  else
12    Flip all the spins inside the chosen cluster of replica 1 and replica 2.
13  return replica 1, replica 2
14  $t \leftarrow 0, \beta_t \leftarrow \beta_0$ 
15 Initialize all parallel chains to random states.
16 while energy variance is greater than tolerance do
17   for each chain in parallel do
18     for each sweep do
19       Sample p-bit states from p-computer.
20       Compute energy of the chain.
21     Compute energy variance for the chain.
22     Save the p-bit states.
23   Compute mean energy variance of chains,  $\sigma_E$ .
24   Set next step inverse temperature:  $\beta_{t+1} \leftarrow \beta_t + \frac{\alpha}{\sigma_E}, t \leftarrow t + 1$ .
25 Initialize all replicas to random states
26 for each swap attempt do
27   if it is an even numbered swap attempt then
28     Choose (even, odd) sequential pairs.
29   else
30     Choose (odd, even) sequential pairs.
31   for each replica in parallel do
32     for each sweep do
33       Sample p-bit states from p-computer.
34       Compute energy of the replica.
35       Randomly partition ICM replicas into pairs.
36       for each ICM replica pair do
37         Perform ICMop on the replicas in the pair.
38   for each sequential pair of replicas do
39     Propose a swap.
40     if accepted then
41       Swap the p-bit states of all corresponding iso-temperature replicas between two replicas.
42 return p-bit states for the replica with the minimum energy.

```

B. Improvement in the performance of APT algorithm with ICM

Here, we justify the inclusion of ICM and compare the performance of APT with and without ICM, as shown in Fig. S14(a). Despite the additional computational overhead introduced by ICM, it offers several advantages: (1) It achieves lower residual energy compared to APT without ICM for a fixed MCS budget, with the difference becoming more pronounced at larger t_a values. (2) It delivers an improved slope compared to APT without ICM. Adding more ICM replicas improves the performance of the algorithm as shown in Fig. S14(b).

Fig. S12. Inverse temperature profile of the APT algorithm: The inverse temperature (β) profiles generated by the preprocessing algorithm for each of the 300 instances are shown. At high temperatures (low β), the profiles across instances are nearly identical, with slight deviations appearing at low temperatures (high β). The figure also shows the standard deviation of sampled energies (σ_E) at each temperature, which decreases monotonically as β increases. The preprocessing begins with $\beta = 0.5$, below which the standard deviation saturates to a constant value. The preprocessing step terminates when the average standard deviation of energy drops below the minimum coupling value, $\min(J_{i,j})$.

Fig. S13. Swap acceptance rate of the APT algorithm: The swap acceptance rate obtained from the APT with ICM algorithm for a randomly chosen instance using the preprocessed temperature profiles is shown. With $\alpha = 1.25$, the average swap acceptance rate across all nearest-neighbor replica pairs is approximately 40%. The results are averaged over 10000 swaps and six independent runs. 4 ICM replicas are used, but their averages are reported as a single replica.

A logical question that arises is whether the bending observed in APT with ICM is primarily due to the increased number of replicas or the inclusion of ICM. Supplementary Fig. S15 addresses this by comparing APT with and without ICM. The impact of ICM is evident from the clear separation between the cases: APT using ICM (blue circles and green squares) show lower residual energy compared to those without ICM (purple triangles), even for the same number of replicas. Additionally, performance improvements are observed as the number of replicas increases, further contributing to the bending. This emphasizes the critical role of non-local moves in classical/probabilistic algorithms, which can significantly enhance performance. A natural next step could be to use non-equilibrium Monte Carlo algorithms aimed to improve the APT algorithm [51], but we do not attempt this here.

C. APT with ICM as a function of sweep to swap ratio

Next, we evaluate the performance of APT with ICM as a function of the sweep-to-swap ratio, which defines the number of sweeps performed before each swap attempt. Our findings indicate that the sweep-to-swap ratio significantly impacts the algorithm performance. While a lower sweep-to-swap ratio increases the number of swaps, it consistently results in better residual energy for a fixed MCS budget, as shown in Fig. S16 for three different sweep-to-swap ratios. Replica energies are calculated at each swap attempt and not saved for the entire annealing time. The sweep-to-swap ratio of 1 gives the best residual energy for an MCS budget even though a more typical sweep-to-swap ratio of 10 achieves close performance with a similar bending behavior, therefore this sweep-to-swap choice does not critically change our results. We carefully verified that computing replica energy at each sweep does not affect the conclusion, confirming that the superior performance at a sweep-to-swap ratio of 1 is not simply a result of increased computational effort. The additional computational cost from more frequent swaps can be mitigated using dedicated hardware, as described in the Methods section of the main text.

D. Performance of APT with ICM on the embedded instances

Fig. S17 compares the performance of APT with ICM on the embedded instances. The performance is very similar to the logical instances showing the transition from a gentler to steeper slope.

Fig. S14. Residual energy (ρ_E^f) from APT with and without ICM: (a) The residual energy as a function of annealing time (t_a) is shown for the APT algorithm, both with and without isoenergetic cluster moves (ICM) and for cube size $L = 8$ (512 spins). The results demonstrate that incorporating ICM improves performance, yielding lower residual energy compared to APT without ICM, particularly at longer annealing times. In this analysis, one Monte Carlo sweep is performed for each replica before a sweep-to-swap ratio of 1 swap is attempted and 4 ICM replicas are used. (b) The impact of varying the number of ICM replicas is illustrated, with performance improving as the number of ICM replicas increases.

Fig. S15. Effect of ICM on residual energy (ρ_E^f) in APT with ICM algorithm: The performance of APT with ICM is compared with varying ICM per MCS to investigate the role of ICM. A Houdayer move [36] is an isoenergetic cluster update used in spin-glass Monte Carlo simulations. It operates by identifying a cluster of antiparallel spins between two replicas and swapping them to enhance mixing and improve sampling efficiency. With 4 replicas for ICM, two replica pairs are available for ICM—one ICM-Houdayer move on each pair. The purple triangles represent APT without any ICM. The blue circles use one ICM on a randomly chosen pair, and the green squares use two ICMs. These three plots use the same number of replicas for a fair comparison. For comparison, we also show APT with 2 ICM replicas (red diamonds) which has only one replica pair for ICM.

E. Slope of the residual energy as a function of system size of the logical instances

In Supplementary Fig. S18, we show the final residual energy as a function of annealing time for four different system sizes, for the APT with ICM algorithm. We perform a scaling analysis: the annealing time is rescaled as $t_a L^{-\mu}$, and the residual energy is rescaled as $L^b \rho_E^f$. This rescaling collapses the data onto a single universal curve, indicating that the system behavior follows a universal finite-size scaling law. The $L = 15$ data is excluded from the collapse analysis because these instances do not form a perfect cube ($15 \times 15 \times 12$). At very low residual energies we also observe another transition to a gentler slope, probably due to the uncertainty in the determination of the ground energy (see Methods section of the main text). We define an arbitrarily chosen target residual energy, ρ_{E0}^f , to approximate the optimization performance. For larger system sizes, achieving a given residual energy target requires progressively longer annealing times. The observed universal collapse confirms that the annealing time needed to reach any target residual energy can be predicted approximately for any system size. Supplementary

Fig. S16. Residual energy (ρ_E^f) with APT + ICM as a function of sweep-to-swap ratio: The performance of the APT with ICM algorithm is evaluated for various sweep-to-swap ratios (defined as the number of Monte Carlo sweeps performed for each replica before a swap is attempted). The residual energy is plotted as a function of annealing time (t_a) for sweep-to-swap ratios of 1, 10, and 516. The results show that a lower sweep-to-swap ratio (sweep/swap = 1) yields the best performance, achieving lower residual energy and a steeper slope, indicating faster convergence towards solutions. In this analysis, 4 ICM replicas are used for ICM.

Fig. S17. Residual energy (ρ_E^f) with APT + ICM for embedded instances: The performance of the APT with ICM algorithm is shown for embedded instances. The residual energy is plotted as a function of annealing time (t_a) for sweep-to-swap ratios of 1 and 4 ICM replicas compared with logical instances of the same size ($L = 15$). The embedded instances show similar characteristics to those of logical instances.

Fig. S18. Residual energy vs. annealing time t_a plots as a function of system size L of the logical instances for APT with ICM: (a) Residual energies plotted against the annealing time for a few system sizes L . All data points are averaged over 100 initial conditions per each of the 300 instances provided/generated. (b) The collapse of the residual energy versus annealing time is shown, using the finite size scaling method. Setting $\mu = 6.34$, and $b = 3$ provides a gentle collapse of the data onto a single universal curve although the collapse breaks down at very low residual energies near the ground states of the instances, probably due to the uncertainty in the determination of the ground energy. The $L = 15$ data is excluded from this collapse because it does not correspond to an exact cube.

Fig. S19 shows similar plots to those in Supplementary Fig. S18, but for the APT algorithm without ICM. To ensure a fair comparison, the number of replicas is kept consistent with that used in the APT with ICM algorithm. Unlike the case with ICM, the collapse here does not exhibit a second bending.

V. FPGA implementation of Adaptive Parallel Tempering

To evaluate hardware acceleration, we implemented the APT with ICM algorithm on a moderately sized FPGA capable of supporting approximately 5000 p-bits. For system size $L = 15$, the FPGA results (Supplementary Fig. S20) closely match CPU simulations, verifying the correctness of the implementation. The details of the implementation are provided in the Methods section. At this scale, implementation of APT without ICM requires 32 to 34 replicas, while APT with ICM requires 128 to 136

Fig. S19. Residual energy vs. annealing time t_a plots as a function of system size L of the logical instances for APT without ICM: (a) Residual energies plotted against the annealing time for a few system sizes L . All data points are averaged over 100 initial conditions per each of the 150 instances provided/generated. (b) The collapse of the residual energy versus annealing time is shown, using the finite size scaling method. Setting $\mu = 6.866$, and $b = 3$ provides a gentle collapse of the early part of the data onto a single universal curve. The later part of the data does not show a second bending like the universal curve for the APT with ICM algorithm.

replicas. To overcome FPGA resource constraints, we employ time-division multiplexing (TDM), allowing the same hardware to be reused for multiple replicas. However, this introduces communication overhead, primarily from saving and reloading p-bit states and performing off-chip energy calculations. As a result, FPGA experiments are limited to 10 instances, 10 runs, and a maximum of 1000 MCS. We emphasize that this is not a fundamental limitation. As our hardware feasibility analysis with custom integrated circuits shows, larger FPGAs or ASICs could eliminate the need for TDM. In addition, energy calculations required for the PT swaps can be performed on chip, reducing overhead and improving scalability. Currently, our FPGA setup can accommodate only one replica of size $15 \times 15 \times 12$ (2687 p-bits) or $16 \times 16 \times 16$ (4096 p-bits). The architecture utilizes graph coloring to maximize parallelism, achieving one sweep per replica in 22.22 ns (45 MHz clock; see Methods Section), corresponding to 120 and 185 flips per ns for the respective problem sizes. Performance can be further improved by increasing the number of on-chip p-bits or interconnecting multiple chips. Our implementation achieves a flips per ns metric 50 to 75 times higher than the 2.5 flips per ns reported for optimized simulated annealing on CPUs [32].

Fig. S20. Verification of APT with ICM algorithm implemented on FPGA: Adaptive parallel tempering (APT) with isoenergetic cluster moves (ICM) on CPUs for the problem size $15 \times 15 \times 12$, averaging over 300 instances and using 50 initial conditions per instance. A sweep-to-swap ratio of 1 minimizes residual energy for APT. Also shown FPGA implementation of APT with ICM, running 10 instances with 10 initial conditions each, closely matching the CPU result. Deviations between the FPGA and CPU may be due to the fixed point weight precision used in the FPGA ($s\{6\}\{6\}$, where ‘s’ denotes the sign bit and the first and second curly braces represent the integer and fractional part of the weights, respectively) compared to float64 in CPU and possibly due to the pseudorandom number generator differences (LFSR in the FPGA and Mersenne Twister in the CPU), though this difference typically does not play a significant role. Note that this precision is higher than what we used for DT-SQA in our feasibility analysis, due to the sensitivity of the APT algorithm to the weight precision. Error bars denote 95% bootstrapped confidence interval of mean over instances.

Review of the manuscript entitled "**Pushing the Boundary of Quantum Advantage in Hard Combinatorial Optimization with Probabilistic Computers**" authored by Shuvro Chowdhury, Navid Anjum Aadit, Andrea Grimaldi, Eleonora Raimondo, Atharva Raut, P. Aaron Lott, Johan H. Mentink, Marek M. Rams, Federico Ricci-Tersenghi, Massimo Chiappini, Luke S. Theogarajan, Tathagata Srimani, Giovanni Finocchio, Masoud Mohseni and Kerem Y. Camsari. (Manuscript #: **NCOMMS-25-33449-T**)

This manuscript investigates the performance of probabilistic computers on hard combinatorial optimization problems, comparing the DT-SQA and APT algorithms with state-of-the-art quantum annealers. The authors argue that, when implemented on massively parallel architectures such as FPGA or projected sMTJ chips, these classical algorithms can match or even exceed the residual energy scaling observed in quantum annealing. While the approach is promising and includes novel architectural insights and new analytical perspectives, further clarification is needed regarding the originality relative to prior work, the justification for using residual energy as a cross-platform benchmark, and the scalability of the methods to larger problem sizes. I believe the manuscript can be significantly strengthened by addressing the following major points.

Major Comments

1. **Clarification of novelty relative to prior work (Lee et al., Communications Physics, 2025)**

While this manuscript presents a valuable comparison by incorporating QMC and quantum annealing (QA) scaling behaviors, its conceptual overlap with the authors' prior work requires further clarification. In particular, the scaling behavior of residual energy and the interpretation of deviations from Kibble–Zurek mechanism (KZM) predictions—framed through extreme value theory (EVT), coarsening dynamics, and the use of asynchronous probabilistic samplers—appear closely related to the results presented in Figure 4 of the earlier publication. Although new terminology (e.g., isoenergetic cluster moves) and theoretical framing are introduced, the overall narrative feels incremental. A more explicit and detailed discussion of what constitutes new physical insights, experimental regimes, or theoretical contributions in this manuscript would be needed to warrant the publication of this work.

2. **Support for claims about probabilistic computing and its advantages**

The manuscript makes strong claims about the competitiveness or even superiority of probabilistic computing hardware relative to QMC and QA approaches. While the reviewer

agrees that such architectures may offer practical advantages in specific applications, the argument would benefit from further support. Specifically, the transition in Section 2—from a formal scaling framework (via KZM) to using residual energy as a single empirical benchmark—should be more clearly justified. For example, explaining how residual energy captures relevant non-equilibrium dynamics beyond what KZM predicts, and under what assumptions it remains a valid comparison metric across such heterogeneous platforms, would strengthen the theoretical foundation for this claim.

3. **Generalization of scaling behavior to larger problem size in APT + ICM algorithm.**

While the authors demonstrate compelling scaling collapse across $L = 8$ to 16 using the APT + ICM algorithm, it remains unclear whether this performance persists for larger problem sizes beyond the tested regime. Given that real-world combinatorial optimization applications often involve far larger systems, it would be valuable to discuss how the proposed methods scale in practice beyond $L = 16$. Could the authors provide additional justification—either through extrapolation, simulation evidence, or architectural considerations—that the observed scaling behavior is robust for larger instances?

Minor Comments

1) **Awkward phrasing in the main text**

'We would like to emphasize that as hardware accelerators the existence of classically hard to simulate quantum dynamics during an annealing schedule is neither necessary nor sufficient for observing speedup over certain state-of-the-art solvers for combinatorial optimization problems. → This sentence does not make a good sense.

2) **Some typos in the manuscript**

- On page 4, '(Supplementary Fig. S7(b))' → '(Supplementary Fig. S7(b))'
- In Fig. S9 on page 21, 'Interconect' → 'Interconnect'
- In Fig. S17 on page 26, 'The embedded instances show similar characteristics to those of logical instanes.' → 'The embedded instances show similar characteristics to those of logical instances.'